# Hebbian priming of human motor learning

**Jonas Rud Bjørndal** [1] ✉, **Mikkel Malling Beck**[1,2], **Lasse Jespersen** [1],
**Lasse Christiansen**[2,3] **& Jesper Lundbye-Jensen** [1] ✉

Motor learning relies on experience-dependent plasticity in relevant neural circuits. In four experiments, we provide initial evidence and a double-blinded, sham-controlled replication (Experiment I-II) demonstrating that motor learning involving ballistic index finger movements is improved by preceding paired corticospinal-motoneuronal stimulation (PCMS), a human model for exogenous induction of spike-timing-dependent plasticity. Behavioral effects of PCMS targeting corticomotoneuronal (CM) synapses are order- and timing-specific and partially bidirectional (Experiment III). PCMS with a 2 ms inter-arrival interval at CM-synapses enhances learning and increases corticospinal excitability compared to control protocols. Unpaired stimulations did not increase corticospinal excitability (Experiment IV). Our findings demonstrate that non-invasively induced plasticity interacts positively with experience-dependent plasticity to promote motor learning. The effects of PCMS on motor learning approximate Hebbian learning rules, while the effects on corticospinal excitability demonstrate timing-specificity but not bidirectionality. These findings offer a mechanistic rationale to enhance motor practice effects by priming sensorimotor training with individualized PCMS.

Across organisms, motor learning is governed by experience-dependent plasticity in relevant neural circuits along the neuroaxis dependent on the task demands[1]. Such plastic changes are timing-dependent[2], and rely at the synaptic level on repeated and closely timed activation between pre- and postsynaptic neurons, a mechanism often explained through Hebbian plasticity[3]. These intrinsic processes of learning may be modulated extrinsically by non-invasive neuromodulation techniques. Paired corticospinal-motoneuronal stimulation (PCMS) is an example of such a technique used in humans[4]. By repeatedly pairing descending corticospinal volleys elicited by transcranial magnetic stimulation (TMS) of the primary motor cortex (M1) with an antidromic volley in the motor axons triggered by electrical stimulation of peripheral nerves timed to arrive at the corticomotoneuronal (CM) synapses in close temporal proximity, it is possible to induce spike-timing-dependent, bidirectional changes in CM transmission[2,5]. PCMS-induced increases in CM transmission have been demonstrated to transiently improve motor function in patients with spinal cord injury[6,7]. However, it remains unknown whether PCMS-

protocols can promote motor learning and how PCMS protocols interact with mechanisms of experience-dependent plasticity.

In a series of four experiments, we investigated the effects of PCMS-induced plasticity on subsequent ballistic motor learning and corticospinal excitability in humans. We hypothesized that PCMS could prime mechanisms of subsequent motor learning when directed to the neural circuitry underpinning the motor behavior. Ballistic performance requires a high voluntary output and, thereby, a high firing rate in as many motor units as possible. Increased efficiency of CM-synapses may lead to a more effective input to the motor neurons, resulting in increased peak acceleration. Indeed, changes in neural transmission at the spinal cord level may contribute in particular to improvements during ballistic motor learning[8].

In Experiment I, we individualized PCMS so that the descending corticospinal volleys elicited by TMS of the M1 hand area arrived at the corticomotoneuronal pre-synapse 2 ms before a peripherally triggered antidromic volley in the motor axons arrived at the post-synapse. This protocol (referred to as the 'PCMS + ') has consistently been shown to

[1]Movement & Neuroscience, Department of Nutrition, Exercise and Sports (NEXS), University of Copenhagen, Nørre Allé 51, Copenhagen N, Denmark. [2]Danish Research Centre for Magnetic Resonance, Centre for Functional and Diagnostic Imaging and Research, Copenhagen University Hospital Amager and Hvidovre, Kettegård Allé 30, Hvidovre, Denmark. [3]Department of Neuroscience, Faculty of Health and Medical Sciences, University of Copenhagen, Blegdamsvej 3B, Copenhagen N, Denmark. ✉e-mail: jrb@nexs.ku.dk; jlundbye@nexs.ku.dk

increase corticospinal excitability in humans[4,6,9]. In Experiment II, we expanded the relevance by replicating the findings from Experiment I in a double-blinded, sham-controlled experiment. Additionally, we investigated the robustness of the behavioral effects in a retention test conducted seven days after motor practice. In Experiment III, we investigated whether the effects of PCMS on ballistic learning were specific to stimulus timing and order. Specifically, whether only paired stimulations timed to facilitate corticomotoneuronal transmission increased learning compared to paired stimulations timed to depress corticomotoneuronal synaptic transmission or leave it unaltered. We additionally investigated the effects of the different PCMS protocols and ballistic motor learning on corticospinal excitability. In Experiment IV, we further scrutinized the effects of PCMS on corticospinal excitability. Specifically, we explored whether the effects of PCMS on corticospinal excitability required pairing of peripheral and cortical stimulations by assessing the effects of both paired and unpaired repetitive TMS and PNS on corticospinal excitability.

We show that non-invasive paired neuromodulation targeting the spinal cord at the level of corticomotoneuronal synapses interacts with subsequent experience-dependent neuroplasticity and leads to improved ballistic motor learning and coinciding increases in muscle activation and corticospinal excitability. The positive priming effects of PCMS+ on spinal motor learning persist seven days after motor practice. This observation highlights the behavioral relevance of the findings. The observed priming effects of PCMS on ballistic motor learning approximate Hebbian learning rules, demonstrating the importance of close temporal proximity and order of spike timing. The effects of PCMS on corticospinal excitability demonstrate timing-specificity but not bidirectionality.

## Results

### Experiment 1: Hebbian priming of ballistic motor learning

In Experiment I ($N = 26$), we found that priming ballistic motor learning with 100 paired stimuli prior to motor practice resulted in superior learning, evidenced by a better ballistic performance at the end of practice compared to controls who only performed motor practice and received no paired stimuli ('Rest') (Fig. 1a, b). This was supported statistically by the linear mixed effects model (LMM) showing a significant GROUP x TIME interaction ($F_{(3,4038)} = 52$, $p < 0.001$) on peak index finger acceleration. Compared to rest, the PCMS+ group performed better at the second practice block (B2) (PCMS + : 139.8% ± 1.8 vs. Rest 115.7% ± 1.8, $p = 0.01$) and third practice block (B3) (PCMS + : 148.2% ± 1.8 vs. Rest 120.5% ± 1.8, $p < 0.01$) (Fig. 1B). This was corroborated by comparisons of changes in performance from the beginning of practice (B1) to the end of practice (B3) (PCMS + : 22.28% ± 1.0 vs. Rest: 10.93% ± 1.0, $p < 0.001$, Fig. 1c). Performance did not differ between groups at baseline in raw values (PCMS + : M = 1.9 g, SD = 0.44 vs. Rest: M = 2.14, SD = 0.46 $p = 0.727$), suggesting that the differences that emerged during practice were not due to general differences in task proficiency but likely could be ascribed to PCMS+ interacting with effects of motor practice.

For both groups, we also investigated changes in corticospinal excitability through motor evoked potentials (MEP) elicited by TMS of the M1 recorded from the first dorsal interosseous (FDI) muscle at baseline, post-stimulation and post-practice (Fig. 1d). We observed no between-group difference at baseline in MEP amplitudes expressed as the ratio to $M_{max}$ amplitude (PCMS + = 8% ± 6 vs. 'Rest' 6% ± 5, $p = 0.26$) (Fig. 1e). PCMS+ led to increased MEP amplitudes as compared to the resting condition (Fig. 1e), evident as a significant GROUP and TIME interaction ($F_{(2,1766)} = 10.9$, $p < 0.001$). Post-hoc comparisons revealed that PCMS+ induced larger increases in MEP amplitudes from baseline to post-stimulation compared to 'Rest' (PCMS + = 103.5% ± 11.3 vs. Rest = 26.4% ± 12.4, $p < 0.01$).

We cannot rule out that the observed between-group difference in average performance in the first part of block 1 of motor practice could

be due to a placebo effect since the design of Experiment I was not blinded nor placebo or sham-controlled. To resolve these questions, we introduced a sham protocol that mimicked the perceptual experience of PCMS+ in the double-blinded Experiment II.

### Experiment II: Double-blinded evidence of lasting behavioral benefits from priming PCMS

The behavioral findings from Experiment I were replicated in a double-blinded, sham-controlled study in twenty individuals participating in Experiment II. The PCMS protocol was similar to the one used in Experiment I whereas sham stimulation consisted of PNS delivered just above the perceptual threshold, and the TMS coil turned upside down. A LMM from Experiment II showed a significant interaction effect of GROUP and TIME ($F_{(4,3882)} = 14.6$, $p < 0.001$). Performance in raw values did not differ between groups at baseline (PCMS + : M = 2.2 g, SD = 0.7 vs. Sham: M = 2.3 g, SD = 0.4 $p = 0.67$). Compared to Sham, the PCMS+ group performed better in the third block of practice (B3) (PCMS + : 128.6% ± 1.4 vs. Sham 119.3% ± 1.4, $p < 0.01$) (Fig. 2B), but not in practice block 1 (B1) or block 2 (B2). Similar to the results in Experiment I, there was no difference at the beginning of practice (B1), indicating that the effects of PCMS+, in fact, interacted with motor practice. Indeed, a significant difference was observed between groups when comparing changes in performance from the beginning of practice (B1) to the end of practice (B3) (PCMS + : 17.4% ± 0.8 vs. Sham: 11.8% ± 0.8, $p < 0.001$, Fig. 2c).

In addition, a 7-day follow-up test was included in Experiment II to assess the effects of PCMS+ compared to SHAM (Fig. 2c) on long-term retention following motor learning. Between-group comparison showed that the PCMS+ group performed significantly better than the SHAM group throughout 50 practice trials at day 7 relative to baseline (PCMS + : 131.4% ± 4.4 vs. SHAM: 119.8% ± 4.4, $p < 0.01$).

The assessment of corticospinal excitability revealed a pattern in part aligned with Experiment 1 and previous results[6,10]: The LMM revealed a significant GROUP and TIME interaction ($F_{(2,1457)} = 5.9$, $p < 0.01$). The two groups were not entirely matched in terms of MEP amplitude normalized to $M_{max}$ at baseline (PCMS = 9% ± 5 vs. SHAM = 5% ± 2, $p = 0.03$) (Fig. 2d, e). Despite larger MEP amplitudes in PCMS+ compared to SHAM at baseline, post-hoc comparisons revealed that PCMS induced relatively larger increases in MEP amplitudes from baseline to post stimulation (PCMS = 58% ± 8 vs. SHAM = 25% ± 8, $p < 0.01$). Analyzes of $M_{max}$ and F-wave amplitudes demonstrated no significant main effects across experiments (see Supplementary Fig. 3 for Experiment I, II & III), suggesting that α-motoneuronal excitability and peripheral recording conditions were stable, and that the normalization to $M_{max}$ was unlikely to explain the observed group differences in corticospinal excitability.

Next, we investigated practice-induced changes in FDI muscle activity during the ballistic motor task and whether changes in peak acceleration were correlated with changes in produced electromyographic (EMG) activity. As effects of PCMS on motor performance were comparable in Experiment I and II we chose to pool acceleration and EMG data for these experiments ($N = 46$ participants). The EMG analyzes demonstrated a significant main effect of TIME for both EMG amplitudes and rate of EMG rise indicating that EMG increased with motor practice across groups. Importantly, we also found a significant GROUP and TIME interactions for both EMG amplitudes and rate of EMG rise. Pairwise comparisons showed that PCMS led to larger increases in EMG during ballistic motor practice compared to controls. Finally, we found a significant positive correlation between relative change in acceleration during ballistic motor task performance and the relative change in rate of EMG rise (see Supplementary Notes and Supplementary Fig. 2 for details). Collectively, this indicates that increased performance in the ballistic motor task was associated with an increased activation of the FDI muscle, which was targeted by the intervention.

## Experiment III: Timing-dependent and network-specific effects of PCMS

Cellular STDP is characterized by after-effects, which depend on both the temporal proximity and order of pre- and postsynaptic spiking and the effects can be bidirectional[11]. To investigate if similar rules of order and timing-specificity apply to the network mediating the priming

effects on ballistic motor learning shown in Experiment I and II, we conducted a within-subject cross-over experiment. In Experiment III, we investigated whether the effects of PCMS on motor learning were dependent on temporal proximity and order of spike timing at the level of the CM-synapses. Moreover, the experiment investigated whether PCMS can lead to bidirectional priming effects in humans and

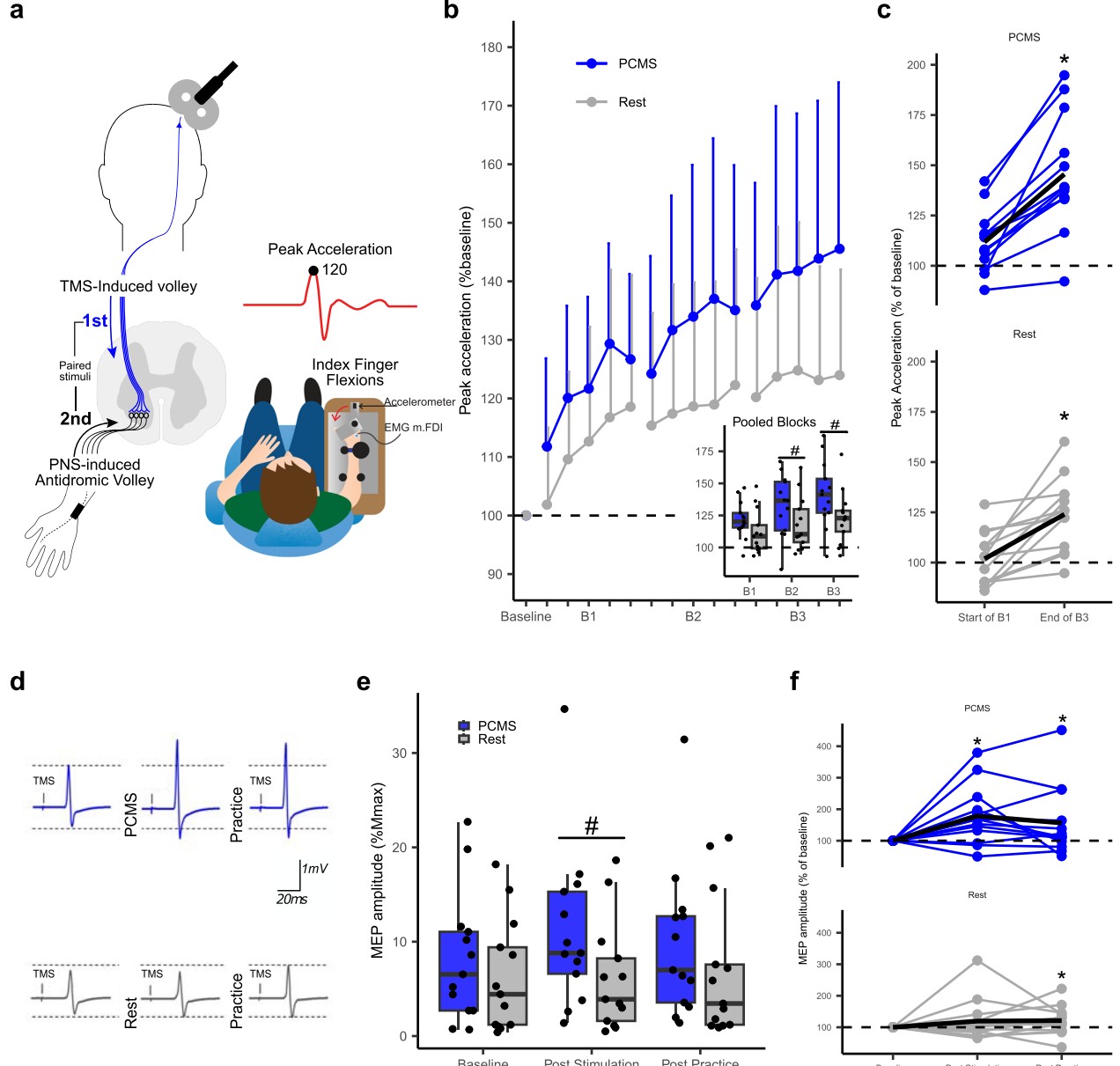

**Fig. 1 | Experiment I: Effects of paired corticomotoneuronal stimulations (PCMS) and rest on ballistic motor learning and corticospinal excitability.**
**a** Between-group design, 26 participants were randomized to two groups (n = 13 in each group), PCMS+ (blue) or Rest (gray), followed by ballistic motor practice. **b** Learning curves; each data point represents the group mean of 10 trials of the ballistic index finger flexion task relative to individual mean at baseline, error bars indicate standard deviation. Pooled data from entire practice blocks are shown in the inserted boxplot showing the median as the midline, the box bounds the 25th and 75th quartiles, and the whiskers bound the minimum and maximum values excluding outliers (defined as values > 1.5xinterquartile range), (same boxplot definition is used in (**e**)), single data points represent individual means. Linear Mixed Effect Model (LMM) evaluated between-group differences in the effect of PCMS+ on peak acceleration (GROUPxTIME interaction: $F_{(3,4038)} = 52$, $p < 0.001$). Pairwise comparisons showed the superior effect of PCMS+ compared to Rest on performance at B2 (#, $p = 0.01$) and B3 (#, $p < 0.01$). **c** Individual data, the mean of

the 10 first trials of practice, compared to the mean of the last 10 trials of practice (black line is group mean). Pairwise comparisons showed higher change in performance during practice in PCMS compared to Rest (*, $p < 0.001$). **d** Example of single subject data displaying raw Motor evoked potentials (MEP) traces after PCMS + or Rest, each trace is the mean of 20 MEPs, and were used to quantify corticospinal excitability (**e**) MEPs in percentage of $M_{max}$ amplitude was obtained from all 26 participants. LMM evaluated between-group differences in the effect of PCMS + on MEP (GROUPxTIME interaction: ($F_{(2,1766)} = 10.9$, $p < 0.001$). Pairwise comparisons showed that PCMS compared to rest led to acute increases in MEP measured at post stimulation (#, $p < 0.01$). **f** Individual responses (MEP % baseline) to PCMS+ or sham conditions (black line represents the mean MEP across participants). The asterisk (*) indicates significant within-protocol comparisons $p < 0.05$ between time points relative to baseline. For all models, pairwise comparisons were two-sided hypothesis tests and were adjusted by the Holm-sidak method. Source data are provided as a SourceData file.

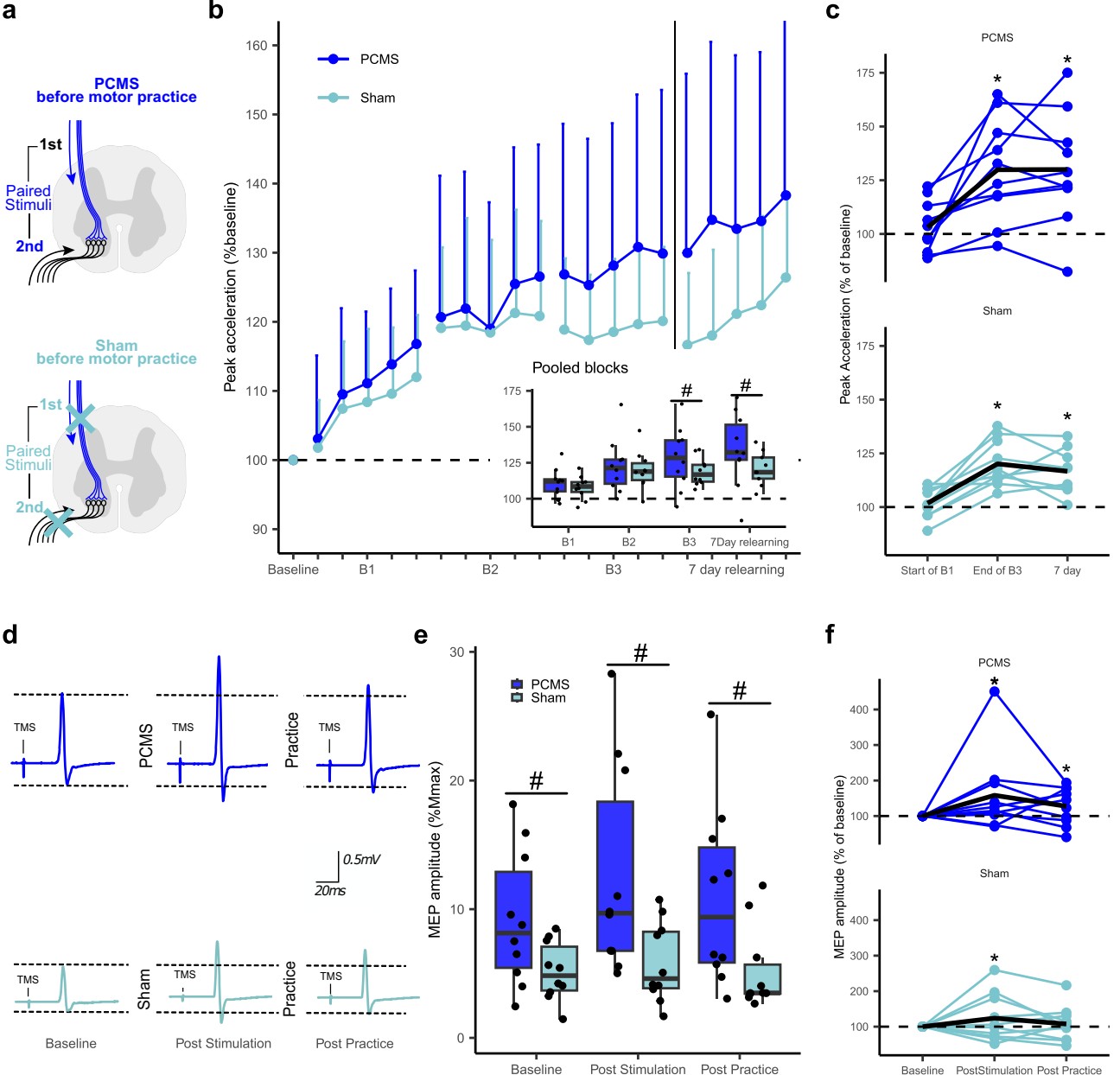

**Fig. 2 | Experiment II: Effects of paired corticomotoneuronal stimulations (PCMS) and sham protocols on ballistic motor learning and corticospinal excitability. a** Double-blinded, sham-controlled design, 20 participants were randomly assigned to either, PCMS+ (blue) or Sham (light blue) (*n* = 10 in each group). **b** Learning curves; each data point represents the group mean of 10 trials of the ballistic index finger flexion task relative to individual mean at baseline, error bars indicate standard deviation. Pooled data from entire practice blocks are shown in the inserted boxplot showing the median as the midline, the box bounds the 25th and 75th quartiles and the whiskers bound the minimum and maximum values excluding outliers (defined as values > 1.5×interquartile range), (same boxplot definition is used in (**e**)), single data points represent individual means. Linear Mixed Effect Model (LMM) evaluated between-group differences in the effect of PCMS+ on peak acceleration (GROUPxTIME interaction: F$_{(4,3882)}$ = 14.6, *p* < 0.001). Pairwise comparisons showed superior effect of PCMS+ compared to sham on performance at B3 (#, *p* < 0.01) and day 7 (*p* < 0.01). **c** Individual data showing the

mean of the first and last ten trials of practice along with the first ten trials seven days later (black line represents group mean). Pairwise comparisons showed higher change in performance during practice in PCMS compared to Sham (*, *p* < 0.001). **d** Example of single subject data displaying raw Motor evoked potentials (MEP) traces after PCMS+ or Sham, each trace is the mean of 20 MEPs. **e** MEPs in percentage of M$_{max}$ amplitude was obtained from all 20 participants. LMM evaluated between-group differences in the effect of PCMS+ on MEP (GROUPxTIME interaction: (F$_{(2,1457)}$ = 5.9, *p* < 0.01). Pairwise comparisons showed that PCMS had higher MEPs compared to sham at baseline (*p* = 0.03) and the other time points (#*p*-values < 0.05). **f** Individual response (MEP % baseline) to PCMS+ or sham conditions (black line represents the mean MEP across participants). The asterisk (*) indicates significant within-protocol comparisons *p* < 0.05 between time-points relative to baseline. For all models, pairwise comparisons were two-sided hypothesis tests and were adjusted by Holm-sidak method. Source data are provided as a SourceData file.

hence be capable of promoting or conversely decreasing ballistic learning depending on spike-timing. We compared the effects of PCMS + to a protocol previously shown to reduce effective transmission at the level of the CM synapse ('PCMS-', interarrival interval of 15ms[4]) and a coupled control protocol where the peripheral stimulation preceded

the cortical stimulation with 100 ms, i.e., well outside the window of spinal interactions (PCMS$_{coupled-control}$) (Fig. 3a).

The LMM revealed a significant interaction effect of PROTOCOL and TIME (F$_{(6,8482)}$ = 12.2, *p* < 0.001) on peak acceleration. In agreement with Experiment I and II, PCMS+ facilitated motor learning as

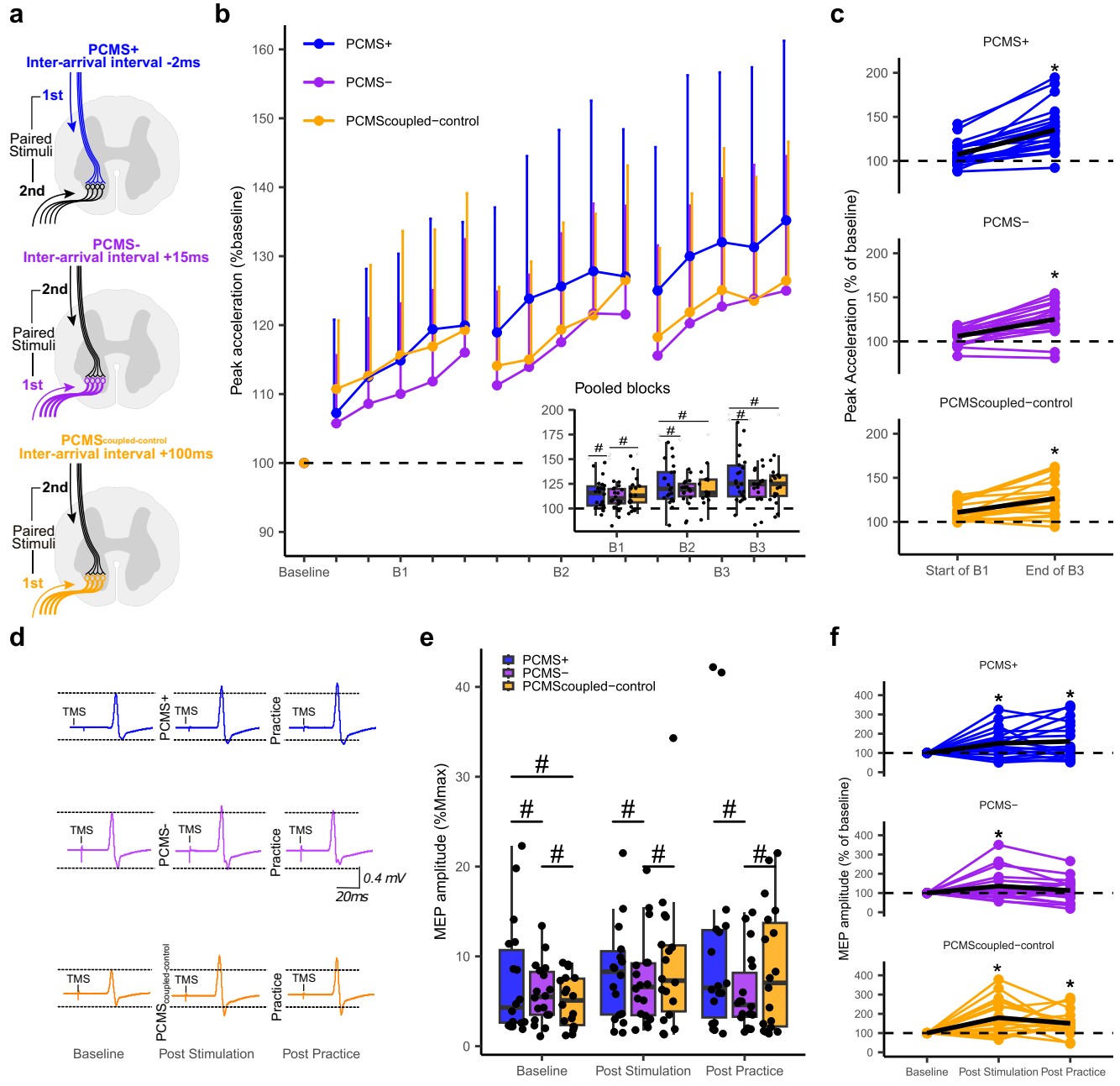

**Fig. 3 | Experiment III: Effects of different PCMS protocols on ballistic motor learning and corticospinal excitability. a** Within-subject design, 18 participants completed three sessions with different PCMS protocols timed according to the arrival interval at the CM-synapse: −2ms (blue, PCMS + ), +15 ms (purple, PCMS-), +100 ms (orange, PCMS_coupled-control), negative inter-arrival interval denotes TMS-volley arriving at the spinal level before the PNS-volley. **b** Learning curves; each data point represents the group mean of 10 trials of the ballistic index finger flexion task relative to individual mean at baseline, error bars indicate standard deviation. Pooled data from the practice blocks are shown in the inserted boxplot showing the median as the midline, the box bounds the 25th and 75th quartiles and the whiskers bound the minimum and maximum values excluding outliers (defined as >1.5xin-terquartile range), (same boxplot definition in (**e**)), single data points represent individual means. Linear Mixed Effect Model (LMM) evaluated between-group differences in the effect of protocol on peak acceleration (PROTOCOLxTIME interaction: F_{(6,8482)} =12.2, p < 0.001). Pairwise comparisons showed superior effect of PCMS+ compared PCMS_coupled-control at B2 and B3 (#p < 0.05). PCMS- had lowed

lower performance in B1 compared to PCMS+ (p < 0.01) and PCMS_coupled-control (p < 0.01). **c** Individual data showing the mean of the first and last ten trials of practice (black line represents group mean). Pairwise comparisons showed higher change in performance during practice in PCMS+ compared to PCMS_coupled-control (p < 0.001) and compared to PCMS- (p < 0.001). **d** Example of single subject data (first test day for different individuals), raw MEP mean traces (n = 20 trials) after each protocol. **e** MEPs in percentage of M_max amplitude was obtained from all participants (n = 18 in each protocol). LMM evaluated between-group differences in the effect of protocol on MEP (PROTOCOLxTIME interaction: (F_{(4,4143)} = 10.95, p < 0.001). Pairwise comparisons showed that PCMS+ had higher MEPs at baseline compared to PCMS- (p < 0.001) and PCMS_coupled-control (p = 0.001). **f** Individual response (MEP % baseline) to the different protocols (black line is group mean). The asterisk (*) indicates significant within-protocol comparisons p < 0.05 between time-points relative to baseline. For all models, pairwise comparisons were two-sided hypothesis tests and adjusted by Holm-sidak method. Source data are provided as a SourceData file.

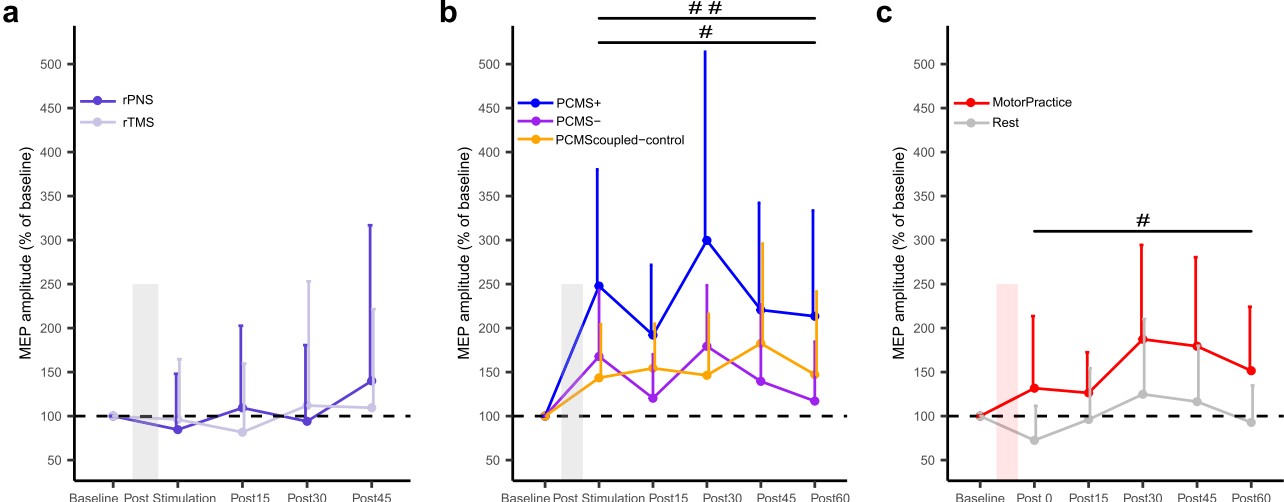

**Fig. 4 | Experiment IV: Control Experiments investigating the effects of unpaired rPNS or rTMS, and the specificity of PCMS effects on corticospinal excitability. a** Within-subject design (n = 10), with participants completing both rPNS (purple) or rTMS (light-purple) separated by a week. To test the effects of unpaired low frequency stimulations on corticospinal excitability. Time course of MEP amplitude in percentage of baseline. Each data point represents the group mean of 20 MEPs for each participant relative to mean at baseline, error bars indicate standard deviation. Linear mixed model (LMM) revealed no significant effect of TIME ($F_{(3,1331)}$=1.01, p = 0.38). **b** Within-subject design (n = 8), with participants completing three test days, to test the effects of three PCMS stimulation protocols on corticospinal excitability (1: PCMS + , blue; 2: PCMS-, purple, and 3: PCMS$_{coupled-control}$, orange). Note similar PCMS protocols were used as in Experiment III, but now with an expanded time course, and without effects of motor practice. Time course of MEP amplitude in percentage of baseline. Each data point represents the group mean of 20 MEPs obtained for each participant relative to mean at baseline, error bars indicate standard deviation. LMM revealed a significant interaction effect of PROTOCOLxTIME ($F_{(8,1694)}$ = 2.05, p = 0.03). Pairwise comparisons showed larger MEPs after PCMS+ across the entire post stimulation period compared to both PCMS- and PCMS$_{coupled-control}$ (##, p < 0.001), and smaller MEPs after PCMS- across the entire post stimulation period compared to PCMS$_{coupled-control}$. (#, p < 0.01). Pairwise comparisons were two-sided hypothesis tests and adjusted by Holm-sidak method. **c** The same 8 participants from (b) completed two other test days, in which the effect of motor practice without prior PCMS was compared against a non-practicing group (Rest) that also did not receive prior PCMS. Time course of MEP amplitude in percentage of baseline, each data point represents the group mean of 20 MEPs for each participant relative to individual mean at baseline, error bars indicate standard deviation. LMM revealed a significant effect of GROUP (#, $F_{(1,1056)}$ = 64.7, p < 0.001). Source data are provided as a Source Data file.

shown by differences in performance between this protocol and PCMS$_{coupled-control}$ in Block 2 and 3 but not Block 1 (Fig. 3b). Comparison between protocols revealed greater improvements from B1 to B3 in ballistic performance in PCMS+ as compared to the other two protocols (PCMS + : 15.9% ± 0.8 vs. PCMS-: 10.8% ± 0.8, p < 0.001; and vs. PCMS$_{coupled-control}$ 8.3% ± 0.8, p < 0.001) (Fig. 3b, c). Additionally, we found PCMS- to have a lower performance in Block 1 compared to the two other protocols (PCMS-: 110.5% ± 1.4 vs. PCMS + : 114.4% ± 1.4, p < 0.01 and PCMS- vs. PCMS$_{coupled-control}$: 114.3% ± 1.4, p < 0.01, and no difference between PCMS+ and PCMS$_{coupled-control}$ p = 0.69, see Fig. 3b).

We found facilitating effects of PCMS on corticospinal excitability as seen from increases in MEP amplitudes (Fig. 3d, e). However, the effects of PCMS on corticospinal excitability differed between protocols. This was supported statistically by a LMM that showed a significant PROTOCOL and TIME interaction ($F_{(4,4143)}$ = 10.95, p < 0.001). Post-hoc comparisons revealed that PCMS+ had larger MEPs (relative to $M_{max}$) at baseline compared to PCMS- (p < 0.001) and PCMS$_{coupled-control}$ (p < 0.001). Despite this baseline difference, PCMS+ induced relatively larger increases in MEP amplitudes from baseline to post stimulation compared to PCMS- (PCMS + :51.3% ± 5.9; vs. PCMS-: +35.3% ± 5.9, p = 0.03), but relatively lower compared to PCMS$_{coupled-control}$ (75.1% ± 6.0, p < 0.01). Pairwise comparisons revealed that PCMS+ and PCMS$_{coupled-control}$ had significantly larger MEPs than PCMS- at post-practice (p-values < 0.01) (Fig. 3d, e). PCMS- did not display a significant within-group difference from baseline to post practice (p = 0.08), as opposed to both PCMS+ (p < 0.001) and PCMS$_{coupled-control}$ (p < 0.001). Notably, the significant baseline difference with larger MEPs in PCMS- compared to PCMS$_{coupled-control}$ reversed during the experiment, with PCMS-

showing lower MEPs at post stimulation (p < 0.01) and post practice (p < 0.01) compared to PCMS$_{coupled-control}$. Analyzes of $M_{max}$ and F-wave amplitudes demonstrated no significant main effects (see Supplementary Fig. 3).

## Experiment IV: Independent effects of stimulation protocols and motor practice on corticospinal excitability

To replicate the effects of PCMS on corticospinal excitability from Experiment III, and to investigate whether the observed effects of PCMS on corticospinal excitability observed in Experiment III were contingent on pairing of peripheral and central stimulations, we tested the effects of unpaired rTMS and rPNS on corticospinal excitability in 10 naïve participants. MEP amplitudes were recorded before the intervention and at 0, 15, 30 and 45 min after the intervention. The results depicted in Fig. 4a showed that unpaired cortical or peripheral nerve stimulations did not induce changes in corticospinal excitability. Neither of the stimulation protocols led to significant changes in normalized MEP amplitudes, evident from the non-significant main effect of Time ($F_{(3,1331)}$ = 1.01, p = 0.38) (see Fig. 4a).

In the second part of Experiment IV, we tested the effects of the three PCMS protocols used in Experiment III without motor practice and the effects of motor practice vs. rest per se on corticospinal excitability. Eight naïve participants completed the PCMS protocols used in Experiment III (PCMS + , PCMS- and PCMS$_{coupled-control}$), a motor practice session (without any PCMS) and a rest session (no PCMS + no motor practice). Session order was randomized and counterbalanced, and sessions were spaced by one week. Repeated measures of corticospinal excitability were obtained before the intervention and at 0, 15, 30, 45 and 60 min after the interventions (PCMS or motor practice). In line with the results presented in Experiment III,

increases in MEP amplitudes were observed following all three PCMS protocols (Fig. 4b). However, the extent of these increases varied among the different PCMS protocols (Fig. 4b). This was supported statistically by the LMM showing a significant interaction effect of Protocol and Time ($F_{(8,1694)} = 2.05$, $p = 0.03$). Post-hoc comparisons showed larger MEP amplitudes across the entire time course after PCMS+ compared to PCMS- ($85\% \pm 10.9$, $p < 0.001$) and PCMS$_{coupled-control}$ ($+49.5 \pm 11.0$, $p < 0.001$). Furthermore, PCMS- displayed smaller MEPs compared to PCMS$_{coupled-control}$ ($-36.3 \pm 11.0$, $p < 0.01$). These results replicate our findings from Experiment III by demonstrating that PCMS can indeed lead to increases in corticospinal excitability irrespective of spike-timing intervals, but also show that the effect on corticospinal excitability is greater in magnitude for the PCMS+ protocol (Fig. 4b).

The final part of Experiment IV investigated the effects of motor practice alone vs. rest on corticospinal excitability. The results demonstrated that ballistic motor practice increased corticospinal excitability compared to rest (Fig. 4c). Across the time period after motor practice, the motor practice group had significantly greater MEP amplitudes (%baseline) compared to the rest condition, evident from the significant main effect of Group ($F_{(1,1056)} = 64.7$, $p < 0.001$).

Collectively, Experiment IV demonstrates that unpaired stimulations do not lead to changes in corticospinal excitability, showing that the facilitation of corticospinal excitability in the present study was coupling-dependent and required paired stimulations. All PCMS protocols increased corticospinal excitability, with the largest effect observed after PCMS + . Finally, Experiment IV demonstrates that ballistic motor learning is accompanied by increases in corticospinal excitability.

## Discussion

This study shows that non-invasive PCMS interacts with subsequent experience-dependent neuroplasticity and leads to improved ballistic motor learning with accompanying increases in muscle activation and corticospinal excitability (Experiment I and II). The improved performance that ensued from the effect of PCMS on ballistic motor learning persisted seven days after motor practice (Experiment II). Finally, this study observed that priming effects on motor learning approximate with Hebbian learning rules (Experiment III): PCMS with a short inter-arrival interval of pre-synaptic activation just before post-synaptic activation at the level of the CM-synapse improved motor learning compared to PCMS protocols with longer timing intervals and opposite order of spike-timing.

In Experiment I, we provide evidence of an interaction between stimulation-induced and practice-dependent neuroplasticity, meaning that priming with PCMS before motor practice leads to improved ballistic learning and coinciding increases in corticospinal excitability compared to rest. The results of Experiment II replicated the findings from Experiment I in a double-blinded sham-controlled design, demonstrating a robust beneficial effect of Hebbian priming on ballistic motor learning.

During the sham, PNS was delivered just above the perceptual threshold, and the TMS coil was turned upside down. Thereby, skin and scalp sensations, along with an acoustic experience, were maintained, but PNS did not elicit an antidromic volley in the axons of the motoneurons, and TMS did not elicit descending volleys. Differences in the effects of PCMS+ and sham on ballistic motor learning can consequently be assumed to reflect the intended pairing of descending and ascending stimulation effects at the corticomotoneuronal level and not peripheral confounds (Fig. 2b). In addition to replicating the acute positive effect on ballistic motor learning, Experiment II additionally demonstrated that the positive effects of PCMS persisted 7 days later, i.e., a lasting effect on retention following combined PCMS and motor practice. This suggests that PCMS+ results in long lasting effects on motor learning that may have relevance in

neurorehabilitation. Previous studies did not find an effect of PCMS on maximal voluntary contraction force[12,13], but a recent study found a positive effect of PCMS on subsequent force control motor practice in electrophysiological responders[14]. Other studies have found positive effects of PCMS alone on motor functions in individuals with and without spinal cord injury[6,7,15]. However, the priming effect of PCMS on motor learning demonstrated in the current study remains to be investigated in patient groups.

In the present study, we demonstrate a positive priming effect of PCMS compared to rest and sham on ballistic motor learning. Without any directional movement constraints, peak acceleration in a specific direction can be increased by optimizing movement direction (likely a cortical phenomenon) as well as acceleration per se by improving fast, coordinated activation of agonist spinal motoneurons. In line with previous experiments, we confined ballistic index finger movements to one plane to emphasize the spinal contribution to learning[8]. Consequently, the observed increase in peak acceleration with training in the present study can be assumed to largely reflect improved efficiency of direct activation of spinal motoneurons. This is supported by analyses of EMG data obtained during ballistic motor practice demonstrating increases in both EMG amplitudes and rate of EMG rise during motor practice. These increases were greatest after priming with PCMS. Furthermore, the change in FDI rate of EMG rise was related to change in performance during ballistic motor learning. In primates, the motoneurons innervating hand muscles are, to a large extent, excited through corticospinal projections with monosynaptic connections from the cortical hand area of the precentral gyrus[16]. The direct corticomotoneuronal projections are also the prime candidate signaling pathway for TMS-evoked excitation of motoneurons (see e.g., review[17]) and our results replicate previous findings that PCMS targeting the spinal cord increases the motor response to TMS over M1 (see review[18]).

Experiment III confirmed the findings from the two first experiments in that facilitatory PCMS targeting the CM synapse primed subsequent ballistic motor learning and increased corticospinal excitability. The results from Experiment III additionally demonstrated that the positive effects of PCMS on motor learning were specific to protocols targeting the CM synapses within a critical window of STDP induction and did not simply depend on arbitrary pairing of cortical and peripheral stimulation (PCMS+ vs. PCMS$_{coupled-control}$). The results also demonstrate that PCMS effects on motor learning are to some extent bidirectional: adjusting the inter-arrival interval to one previously demonstrated to induce negative after-effects on corticomotoneuronal transmission impaired early learning (PCMS-) evident from the lower performance compared to the PCMS$_{coupled-control}$ in the first block of motor practice. It should be noted that this negative effect of PCMS- on learning was only observed during the early stage of learning, suggesting a smaller or less consistent effect of PCMS- that was readily overruled by the subsequent improvements induced by practice. The smaller effect of PCMS- may also in part be influenced by effect sizes being diluted by the within-subject design and the ensuing carry-over effects between experimental days. Nevertheless, the results of Experiment III, provide proof-of-principle that the priming effects of PCMS on ballistic learning approximate Hebbian learning rules, in that they may be contingent on temporal proximity of paired stimulations and order of spike-timing.

In the present study, we have expanded the line of evidence that PCMS protocols with an inter-arrival interval (IAI) at the CM synapse of 2 ms (pre-synapse before post-synapse) acutely facilitate corticospinal excitability[4,6,10,19–21]. Importantly, we did not find changes in M$_{max}$ or F-wave amplitudes in any of the three experiments in the present study, indicating that the protocols did not influence motoneuronal excitability or peripheral recording conditions (see Supplementary Notes and Supplementary Fig. 3 for details). Compelling evidence suggests that the increased corticospinal excitability following PCMS+

(IAI of 2 ms) is due to enhanced transmission at the spinal level and more specifically at the corticomotoneuronal (CM) synapses. Previous studies have reported increased size of cervicomedullary motor evoked potentials (CMEPs)[4,22]. Based on the narrow spike width in post-stimulus time histograms of single-motor unit firing after cervicome-dullary electrical stimulation, the spinal motoneurons are thought to be excited predominantly through monosynaptic connections[23]. In support, PCMS has not been demonstrated to increase F-wave per-sistence or amplitude, which provides indirect measures of intrinsic α-motoneuronal excitability, rendering the CM-synapse a likely site for the facilitation of corticospinal excitability[6]. Previous studies have demonstrated that the efficiency of CM-synapses can be modulated non-invasively using stimulation techniques[24]. Previous work also suggests that improvements in ballistic motor performance depend on plastic changes at the corticomotoneuronal synapses evident from increases in CMEP amplitude after practice[8] along with limited changes in F-waves[25]. Taken together, this evidence suggests that improving ballistic practice is contingent on experience-dependent plasticity at CM synapses that are also the target of the PCMS+ protocol.

In Experiment III, we also compared the effects of PCMS+ to two other PCMS protocols designed to leave transmission at the CM synapse unchanged (PCMS$_{coupled-control}$) and designed to reduce effective CM transmission (PCMS-). We found again a positive priming effects of PCMS+ and as acknowledged a smaller and shortlived negative effect of PCMS- on ballistic motor learning. Interestingly, we found increases in MEP amplitudes in all three protocols. These effects on corticospinal excitability were replicated in another sample in Experiment IV. Importantly, our results suggest that increasing corti-cospinal excitability does not improve motor learning per se; only when the targeted plasticity is directed at the neural circuitry that underpins the specific motor behavior – i.e., PCMS+ aimed at the CM level – then subsequent ballistic motor learning is enhanced. This shows that the relationship between changes in corticospinal excit-ability and motor performance is not straightforward, and this is a commonly observed phenomenon in the literature[26–28]. Nevertheless, the observation that PCMS$_{coupled-control}$ led to increased MEP ampli-tudes provides evidence that repeated pairing of high-intensity PNS (130% M$_{max}$) and TMS (150% rMT) at this interval increases corticosp-inal excitability. In conventional paired associative stimulation (PAS)[29], paired stimulations are timed with an interstimulus interval (ISI) of ~25 ms to target the sensorimotor cortical circuitry mediating short latency afferent inhibition. However, it is less well-studied how per-ipheral nerve stimulation with an ISI of ~107 ms (CM IAI of 100 ms) before TMS modulates the size of the MEP[30]. The one study investi-gating this ISI found an inhibition of the MEP[31], ascribing the effect to networks associated with long latency afferent inhibition (LAI) (see e.g., Turco et al.[30] for discussion). It should be acknowledged that the peripheral stimulation at supramaximal intensities may have elicited sensory afferent activation, which at longer latencies may have influ-enced cortical processing and potentially influenced the TMS pulse at the cortical level. Although caution is warranted when concluding on the mechanisms underlying the effects of PCMS$_{coupled-control}$ (IAI 100 ms) on corticospinal excitability, we suggest changes in the cor-tical sensorimotor network mediating LAI to account for the observed facilitation.

Furthermore, we did not find a suppressive effect of PCMS- on corticospinal excitability. Instead, a significant facilitation of MEP amplitudes was observed after PCMS-. At first glance, this result is at odds with previous reports, but in contrast to Taylor & Martin (2009)[4], we evaluated corticospinal excitability with MEPs evoked by cortical TMS and not CMEPs, a measure of subcortical excitability. One study found that inhibitory PCMS was accompanied by suppressed corti-cospinal excitability, but only for individuals in whom, the afferent input reduced MEP size during the PCMS protocol[7]. The lack of a suppressive effect of inhibitory PCMS on corticospinal excitability in

the present study could thus be related to electrophysiological non-responders. We did however, not see any link between responders/non-responders and motor performance (this is elaborated further in the Supplementary Discussion and Supplementary Fig. 4). Although PCMS- increased rather than decreased corticospinal excitability, this does not exclude the possibility of a suppressive effect at the CM level. Interestingly, when targeting FDI, the average interstimulus interval (ISI) between the delivery of PNS and TMS needed to ensure the inter-arrival interval of 15 ms at CM level was ~22.5 ms. This closely resembles the ISI used in facilitatory PAS, demonstrated to induce STDP-like plasticity at the cortical level, leading to increased MEP amplitudes[29]. This means that although PCMS- was designed to reduce effective transmission at the CM-level, it is possible that the ISI needed to induce such timing-specific interactions also led to co-occuring changes in cortical circuits. It is well-known that the MEP size reflects excitability along the corticomotor pathway as well as upstream of M1[32]. In this light, the MEP amplitude after PCMS- likely reflects the sum of cortical and spinal plasticity, including potential opposing effects. However, as we did not evaluate changes in muscle response to electrical cervico-medullary stimulation, we are limited to speculating on the loci of PCMS induced changes in excitability.

Another alternative is that the observed after-effects on corti-cospinal excitability were not caused by the coupling during paired stimulations per se, but rather by individual effects of PNS or TMS alone. To investigate this possibility, we conducted a set of control experiments presented in Experiment IV. These control experiments showed that neither unpaired high intensity TMS or unpaired high-intensity PNS at a frequency of 0.1 Hz resulted in changes in MEP amplitudes (see Fig. 4). These findings are consistent with results from previous studies[22,33]. However, It has previously been reported that high-intensity rTMS at 0.1 Hz can lead to increased MEP amplitudes in the adductor pollicis muscle, with effects lasting up to 35 min[34]. Dif-ferences in stimulation parameters and the specific muscle targeted could potentially account for these conflicting findings. Nevertheless, our control experiment demonstrated that the stimulation parameters employed for TMS and PNS in the present study did not increase MEP amplitudes when administered alone. This emphasizes that the observed changes in corticospinal excitability were specifically medi-ated by the paired stimulations, rather than their individual effects. Experiment IV additionally demonstrated in another independent sample of individuals, that all PCMS protocols increased corticospinal excitability compared to rest. Notably, the PCMS+ led to larger increases in corticospinal excitability compared to PCMS- and PCMS$_{coupled-control}$. As was the case in Experiment III, the PCMS- pro-tocol did not result in a suppression of MEP amplitudes. The results demonstrate again – but now for changes in corticospinal excitability – that facilitatory effects required paired stimulations and were modu-lated by specific timing aimed at the CM synapse. This finding is rele-vant, since it demonstrates the specificity of the effects related to pre- and postsynaptic activation at the CM level and the temporal proxi-mity of this activation. Collectively, the findings from Experiment III and IV show a facilitation of corticospinal excitability measured via motor evoked potentials following PCMS. This finding demonstrates that while the behavioral results approximate with Hebbian principles and learning rules, the effects of PCMS on corticospinal excitability also demonstrate the necessity of paired stimulations and an impor-tant role of timing-specificity but not bidirectionality. This latter result may be explained by the multilevel effects of PCMS- as discussed above.

In summary, this study demonstrates that non-invasive paired neuromodulation targeting the spinal cord at the level of CM synapses interacts with subsequent experience-dependent neuroplasticity leading to improved ballistic motor learning that persists after seven days. The behavioral effects of PCMS on ballistic learning approximate with Hebbian learning rules. Previous studies have demonstrated the

clinical potential of PCMS in the recovery of dexterous hand function after spinal cord injury[6] and stroke[35]. Our findings suggest that promising results of PCMS in improving dexterity in spinal cord injury may generalize to movements with other control policies through interactions between PCMS-induced and practice-dependent plasticity. The fact that the improvements observed after practice persisted seven days later suggests that they are long-lasting and behaviorally relevant. Whether repeated sessions of priming practice of brisk movements with PCMS+ result in cumulative effects is yet to be investigated. Notwithstanding, our results support PCMS as an add-on therapy in neurorehabilitation, benefitting both manipulative and vigorous movements.

## Methods

### Participants

All the experimental procedures were approved by the local ethics committee for the Greater Copenhagen area (protocol H-17019671) and the study was performed in accordance with the declaration of Helsinki. Participants were compensated economically for the time spent on their involvement in the study. The study included sixty-six young (32 males; age 20–30 y/o − 25 ± 2 mean ± standard deviation (SD)) adults who all volunteered and provided consent after thorough information of the study procedures. The sex of participants was determined based on self-report. Participants were defined as able-bodied based on a standardized general eligibility questionnaire, with no history of neurological, psychiatric, or medical diseases and no intake of medication. All participants were right-handed (except one who had no preference), according to the Edinburgh Handedness Inventory (Laterality quotient: 88.1 ± 25.9 mean ± SD)[36].

### Experimental design

Four separate experiments were conducted: Experiment I, a between-group design with twenty-six participants (14 males/12 females, mean age 24.9 ± 2.2) randomized to two groups PCMS+Practice and Rest+Practice. Experiment II, a double-blinded between-group design, with twenty participants (10 males/10 females, mean age 25.5 ± 2.3) randomized to PCMS+Practice and Sham+Practice. Experiment III, a within-subject repeated measures design, where eighteen participants (9 males/9 females, mean age 25.3 ± 2.3) (incl. first 6 from PCMS group from experiment I) took part in three test days in a randomized and counterbalanced order with one week between sessions (Supplementary Fig. 1). In Experiment IV we investigated effects of paired and unpaired stimulation protocols and effects of motor practice on corticospinal excitability.

In Experiment II, Investigator A was aware of allocation (whether PCMS or Sham was administered), while Investigator B was blinded to the allocation. During the intervention, Investigator B physically left the laboratory during the PCMS/Sham protocol. Investigator B returned to the laboratory after the PCMS/sham intervention and oversaw the motor practice session. Importantly, before commencing motor practice, Investigator A (who was aware of the group allocation) left the laboratory. Effectively, this means that Investigator A could not influence how participants performed in the motor practice session. Group allocation was not revealed until all data points were collected and analyzed. This also means that Investigator B performed the data analysis blinded. With this setup, Investigator A, who managed the TMS coil (position and orientation) during PCMS or Sham, was unable to affect motor performance and the subsequent data analysis.

We chose a within-subject design for Experiment III, to try to limit inter-individual variability between groups. Based on previous literature[37,38] we rationalized that our participants would not reach a plateau in motor performance after just three motor practice sessions (with 150 trials in each) spaced with one week. A test session consisted of one of three different PCMS protocols using TMS and PNS, followed by ballistic motor training. In all experiments, participants were

asked to fill in a sleep diary with their night's sleep prior to the experiment. Repeated measures of motor performance (peak acceleration), corticospinal excitability (MEPs), and neuromuscular excitability ($M_{max}$) were recorded at baseline, post-PCMS, and post-ballistic training on each test day. The procedure on test days was identical except for the electrophysiological intervention (i.e., paired stimulations or none) with different settings of the inter-arrival interval (IAI) in the PCMS protocols. This experimental design allowed us to investigate the acute effects of different PCMS protocols on ballistic motor performance and on the performance gains during the practice of a ballistic motor task. Furthermore, it allowed us to investigate the effects of different PCMS protocols and ballistic motor practices on corticospinal excitability.

Experiment IV was performed to control for the effects of unpaired stimulation protocols, the role of stimulus timing in paired protocols, and the effects of motor practice on corticospinal excitability. In Experiment IV, ten participants (4male/6female, mean age 25.5 ± 2.1) from Experiment III accepted an invitation to engage in two experimental sessions investigating effects of unpaired rTMS or PNS. The protocol did not involve motor practice, and the order was counterbalanced between participants. In Experiment IV, eight new participants (5male/3female, mean age 25.7 ± 2.1) engaged in five experimental sessions investigating the effects of PCMS protocols and effects of ballistic motor practice vs rest. In all experiments, measures of corticospinal excitability were obtained at baseline and again 0, 15, 30, 45, and 60 minutes after the intervention.

### Ballistic motor performance and learning

To assess ballistic motor learning, participants practiced a task requiring rapid accelerations via flexions of their right index finger. Participants were seated in a height-adjustable chair in front of a computer screen (1920×1200 res, Dell U2415) showing a white screen with a green vertical midline. Their right arm was elbow flexed approximately 90° degrees resting on the table while grabbing a custom build handle (Fig. 1a). Their index finger was placed in a metal splint placed perpendicular to the axis of rotation of the handle, with an accelerometer mounted on top of the metal splint. The handle allowed flexions of the index finger but restricted movements in other planes. The accelerometer signal was amplified and filtered (low pass 20 Hz) and sampled at 1 kHz on a computer with a USB6008 DAQ Board (National Instruments, Inc.). A trial consisted of a rapid finger flexion within a 1 second window, with a new trial every 4th second (software created for the purpose in MATLAB R2012b, MathWorks inc.). In a trial, participants were paced by a horizontal trace running from left to right on the white computer screen. When near the green midline, the participants performed the movement. Instruction of the task consisted of a visual presentation that underlined that the movement should be performed approximately around the green midline, meaning that accuracy and precise timing at the green midline were not important. Participants were instructed to perform each movement as fast as possible and to keep improving their score during the practice blocks. A goniometer was attached to the handle above the metacarpophalangeal joint to measure the position of the movements, and to make sure that participants got back to the stretched starting position after each trial. Participants were allowed five familiarization trials before the baseline test on the first test day. Motor performance was quantified as peak acceleration and was measured in 10 trials at each time point (baseline, post-PCMS and post practice). At baseline and post-tests, no augmented feedback was provided on motor performance. During practice, participants performed three blocks of 50 trials with a 2-minute break between blocks. Augmented feedback was provided during the three practice blocks following each trial as a score normalized to the score measured at post-PCMS. The augmented feedback was presented as knowledge of results for 2 s before the next trial began. Additionally, verbal encouragement was

provided during practice at least every 10th trial to ensure that participants were motivated.

## Electrophysiological recordings

Electromyography (EMG) was recorded from m. FDI on participants' right hand through surface electrodes (Ag-AgCl, 1 cm diameter) applied on the skin after preparation with medical sandpaper. The electrodes were placed in a muscle-belly-tendon montage, with the active electrode on the muscle belly. A zinc plate was used as a ground electrode placed at the base of the hand. The EMG signals were amplified (x200), filtered (band-pass 5 Hz to 1 kHz), and sampled at 2 kHz on a computer for offline analysis (Cambridge Electronic Design 1401 with Signal software v6.05). Line noise (50 Hz) was removed with a Hum Bug noise eliminator (Digitimer). During the ballistic motor task, EMG signals were amplified (x200), filtered (band-pass 5 Hz to 1 kHz), and sampled at 2 kHz on a computer for offline analysis (Cambridge Electronic Design 1401 with Spike2 software v7.10).

## Transcranial magnetic stimulation

We assessed corticospinal excitability as the peak-to-peak EMG amplitudes of TMS-evoked muscle responses, called motor-evoked potentials (MEPs). Monophasic single-pulse TMS was applied to the contralateral M1 to the dominant hand via a figure-of-eight TMS coil (Magstim®D70² connected to a Magstim²⁰⁰). The hotspot for each participant in each experiment was localized via a mini-mapping procedure by determining the site of stimulation that provided large and robust MEP responses in the FDI. The coil was placed with the center oriented parallel to the scalp over the hotspot of FDI representation with the handle of the coil pointing backward at an angle of 45° to the sagittal and horizontal axis (Supplementary Information Fig. 1a). This induces a posterior-anterior current direction in the targeted cortex. The resting motor threshold (rMT) was defined as the stimulus intensity needed to elicit recognizable MEPs with an amplitude above 0.05 mV in 5/10 consecutive stimulations[39]. Muscle relaxation was monitored through concurrent EMG recordings. Twenty TMS stimulations (120% of rMT) were delivered at each time point (2xbaseline, post-PCMS, and post-motor practice) to measure MEP amplitudes. A neuro-navigation system (Brainsight 2, Rogue Research, Montreal, Canada) was used to ensure stable positioning of the coil throughout the experiments. MEP latencies during voluntary contraction (10% MVC) were recorded to calculate the central conduction time used to inform the interstimulus intervals in the paired stimulation protocols.

## Electrical peripheral nerve stimulation

Electrical stimulation with high voltage electrical current (200 μs pulse duration, DS7A; Digitimer) was delivered to the ulnar nerve at the wrist (Bar Stimulating Electrode, Digitimer) to measure the maximal compound muscle action potential ($M_{max}$) and F-waves. The latencies of $M_{max}$ and F-waves were used to calculate the peripheral conduction time used to individualize the paired stimulation protocols (Supplementary Fig. 1b–d). F-wave latency was defined as the F-wave with the earliest onset[10]. At baseline, post stimulation and post practice $M_{max}$ amplitudes, and F-wave amplitudes were measured to indirectly quantify α-motoneuron excitability[40]. We we delivered 60 stimulations at supramaximal (130% of $M_{max}$) intensity delivered at 1 Hz[41].

## Paired corticomotoneuronal stimulation protocols

PCMS protocols were individualized based on recordings of $M_{max}$, Fwave and MEP latencies from each participant on each test day. Interstimulus intervals were based on calculations of individual peripheral and central conduction times. The PCMS protocols consisted of 100 pairs of TMS (intensity at 150%rMT) and PNS (intensity at 130% of $M_{max}$ stimulation intensity) with a frequency of 0.1 Hz. The stimulation protocols targeted the corticospinal-motoneuronal synapses with different inter-arrival intervals between the descending TMS

volley and the antidromic stimulation from PNS (see Supplementary Methods and Supplementary Fig. 1 for details).

## Unpaired low-frequency stimulation protocols

The unpaired stimulation protocols applied in Experiment IV were identical to the paired stimulation protocols. The protocol however involved only TMS stimulations or peripheral nerve stimulations.

## Data analysis

Visual inspection of all signals during and after the experiments ensured the exclusion of M-waves and MEPs that contained pre-stimulus EMG activity 100 ms before stimulation[38]. Post hoc analysis included the removal of outliers defined as mean±2 standard deviation for the given measurement. MEPs were normalized to the respective $M_{max}$ amplitude to allow between-session comparisons. F-wave recordings were filtered offline using a $2^{nd}$ order Bessel high-pass filter (200 Hz) to allow accurate assessment of onset latency, amplitude and persistence (% of F-waves with amplitude > 0.045 mV)[42]. Peak acceleration scores were normalized to the baseline value obtained in the same experimental session to allow comparisons between participants. Data from the motor practice sessions of a total of 150 trials of peak acceleration was analyzed both at block level (50 trials in each block) and binned in trials of 10 to analyze motor performance within a practice session. FDI EMG data obtained during the ballistic task was analyzed post hoc for Experiments I and II to investigate if EMG during ballistic contractions changed with motor practice. Rectified EMG data was filtered using a moving root-mean-square filter with a time constant of 50 ms (similar to Aagaard et al.[43]). We calculated EMG root-mean-square ($EMG_{RMS}$) amplitude in a time window from EMG onset to 70 ms after EMG onset. Additionally, rate of EMG rise was calculated from EMG onset to 30 ms. Both measures were normalized to the individual $M_{max}$ amplitude, to allow comparison between sessions (see Supplementary Fig. 2, for EMG results).

## Statistical analysis

All statistical analyses were performed using R (R Core Team, 2022[44], version 4.1.3), and data visualization was done using the R-Package *ggplot2*[45] (v3.3.5). Linear mixed effect models were fitted to data for all dependent variables (peak acceleration, EMG, MEP-, $M_{max}$- and F-wave amplitudes) using the *lme4* R-package (v1.1–28)[46]. For Experiment I and II, linear mixed effect models with the fixed factors GROUP (2 levels: Rest/Sham and PCMS) and TIME (Experiment I: 4 levels: Baseline, Block1-3, Experiment II: 5 levels: Baseline, Block1-3, Day7) was fitted to the acceleration and EMG data with an interaction term (GROUP and TIME). In Experiment III, linear mixed effect models with the fixed factors PROTOCOL (3 levels: PCMS + ; PCMS-; $PCMS_{coupled-control}$) and TIME (4 levels: Baseline, Block1-3) were fitted to the acceleration and EMG data with an interaction term (GROUP and TIME). In Experiment III and IV, potential order effects were accommodated statistically by adding 'DAY' to the model with an additive term. Random intercepts were fitted for each subject to account for the repeated measures design. We also tested in a new model the relative change from the first 10 trials of Block 1 to the last 10 trials of Block 3 Figs. 1–3C). For all four experiments, we made similar models for the electrophysiological data (MEP-, $M_{max}$- and F-wave amplitudes), only changing the levels of the TIME factor (3 levels: Baseline, Post Stimulation, Post motor practice) and in Experiment IV (6 levels: Baseline, Post Stimulation, Post15-60). Assumptions of normality and homogeneity of variance of residuals were inspected by quantile-quantile plots and residual plots. To evaluate significance of main effects or interactions, we used the R-package *lmerTest (v3.1-3)*[47] that computes P-values from mixed effect models via the Satterthwaite's degrees of freedom method. If main effects or interactions were significant, we proceeded to pairwise comparisons using the *multcomp* R-package (v1.4-18)[48]. Namely, we computed contrasts to test specific hypotheses,

e.g., how ballistic performance was affected after the paired stimulations (during practice). These contrasts are presented as model estimates with standard errors (SE). Mean (M) and standard deviation (SD) are shown when raw data is presented. For all post-hoc comparisons, the Holm-Sidak method was used to adjust for multiple statistical comparisons. Following Experiments I and II, we used Pearson correlation to explore whether practice-induced changes in EMG measures were correlated with practice-induced changes in peak acceleration. For all statistical analyses the significance level was set at $p < 0.05$.

### Reporting summary

Further information on research design is available in the Nature Portfolio Reporting Summary linked to this article.

## Data availability

The raw data are protected and are not available due to data privacy laws. The processed source data generated in this study have been deposited in the Figshare database with the following https://doi.org/10.6084/m9.figshare.23689119. The complete dataset used in the study is available upon request to J.R.B. or J.L.J. Source data are provided with this paper.

## Code availability

Matlab scripts for the custom-made non-commercialized behavioral task are uploaded to Github[49], see link https://doi.org/10.5281/zenodo.11203521. R-scripts to reproduce figures are uploaded to Github: https://github.com/JONAS-RUD-BJORNDAL/HebbianPrimingMotorLearning.

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

## Acknowledgements
The authors would like to thank the participants for their time and patience. We are grateful for the financial support from Nordea-fonden (Grant no. 02–2019-00045). Lasse Christiansen holds a postdoc grant from the Lundbeck Foundation (Grant no. R322–2019–2406). Mikkel Malling Beck holds a postdoc grant from the Capital Region of Copenhagen (Region H) and was funded by a grant from Innovation Fund Denmark (Innovation Fund Denmark, grant no. 9068-00025B) and the Danish Ministry of Culture (Grant no. FKP.2018-0070).

## Author contributions
Conceptualization, J.L.J., M.M.B., L.C. and J.R.B.; Methodology, J.L.J., M.M.B. and J.R.B.; Formal Analysis, J.R.B. and M.M.B.; Investigation, J.R.B., L.J. and M.M.B.; Writing – Original Draft, J.R.B.; Writing – Review & Editing, J.R.B., M.M.B., L.C., L.J. and J.L.J.; Visualization, J.R.B.; Supervision, M.M.B. and J.L.J. Project Administration, J.L.J.; Funding Acquisition, J.L.J.

## Competing interests
The authors declare no competing interests.
