## [Peer Review File · Nature Communications]

REVIEWER COMMENTS

Reviewer #1 (Remarks to the Author):

This manuscript reports an interesting series of studies that support the idea that priming the human motor pathway through a paired-stimulation plasticity protocol (PCMS) can influence later activity-dependent changes in motor output. The studies provide good evidence that a non-invasive protocol designed to elicit facilitatory spike-timing-dependent plasticity at corticospinal-motoneuronal synapses can increase the rate of improvement during subsequent practice of a fast movement. The evidence for a reduction of the rate of improvement with PCMS designed to depress synaptic efficacy is less compelling.

Previously, PCMS applied in people with incomplete spinal cord injury has been shown to improve motor learning of a task requiring dexterity (nine-hole peg test; Bunday & Perez 2012) and PCMS in able-bodied people can immediately alter motor output in submaximal tasks (Taylor & Martin 2009; Bunday & Perez 2012). The current manuscript is novel in showing effects of PCMS on learning of rapid movements and also that the performance benefits from practice after facilitatory PCMS last for at least 7 days.

As the practised movement engages other muscles besides those targeted with the paired-stimulation plasticity protocol, reporting of changes in first dorsal interosseous EMG that are consistent with the changes in finger acceleration would strengthen the argument that the protocols interact to alter motor learning with regard to this muscle in particular.

Major comments

1. The manuscript lacks an Introduction. Therefore, understanding the rationale requires considerable prior familiarity with paired corticospinal motoneuronal stimulation, activity-dependent plasticity, spike-timing-dependent plasticity, control of human movement and human neurophysiology.
2. The choice of index finger flexion rather than abduction complicates the interpretation of these studies. The long finger flexors are prime movers for index flexion but PCMS with peripheral nerve stimulation delivered at the wrist was designed to act on the ulnar innervated intrinsic muscles of the hand. Specifically, it was aimed at first dorsal interosseous (FDI). FDI does contribute to index flexion, as does the first palmar interosseous muscle which is also ulnar innervated and should also be influenced by the PCMS. However, the relative contribution of these ulnar-innervated intrinsic muscles versus the other muscles contributing to index flexion is uncertain. A report of FDI EMG during the ballistic movements would give more certainty that greater acceleration with training has a contribution from greater FDI activity and hence, that the effects of PCMS are consistent with the authors' interpretation.

3. Experiment III has a within-subject cross-over design in which individual participants underwent three different interventions at one week intervals. As Experiment II showed that the learning effect of repeated movements lasted for at least one week, the rationale for the cross-over design with one week washout is unclear. The design seems likely to reduce possible differences between interventions given that increases in movement acceleration must eventually reach a plateau. Nonetheless, differences between interventions did occur. However, discussion of the rationale and/or possible influence of research design is warranted.

Minor comments

There seem to be no stimulation intensities given for either TMS or peripheral nerve stimulation when they are delivered as PCMS.

“+” symbol appears before some values e.g. lines 70, 138, 157, 223

All Figure legends. No descriptions of what the error bars represent.

Figure 2B. Problem with the y-axis title – missing letters at the start of acceleration.

Line 157. Not clear what 120.58% and 115.56% are percentages of.

Line 162 and elsewhere. “Despite of” should be “Despite”

Lines 224-239. A strong conclusion “the priming effects of PCMS on ballistic motor learning are bidirectional depending on the spike-time” is drawn but without discussion of the result that the decrease of motor learning might only last through one block of practice.

Lines 232 & 262 and elsewhere. The use of “inhibitory effect” to describe the effect of PCMS- should be carefully reviewed as the plastic change postulated is a down regulation of facilitatory synaptic input rather than a change in inhibition.

Line 267. Maybe “CM IAI” should be “CM ISI”. Also line 272.

Line 290. Supplementary information says that TMS at 0.1 Hz was at 120% RMT. That is not really “high intensity”

Supplementary information

PCMS protocols section does not mention stimulus intensities

Effects of non-paired rPNS or rTMS section does not mention rPNS intensity.

Responders/ Non-responders section indicates that TMS was given at 150% RMT during PCMS. This means that the control testing for MEP increases with non-paired rTMS should have occurred with 150% RMT. This is not obvious from the description of the control experiment.

Reviewer #2 (Remarks to the Author):

Experiments reported in this manuscript examined whether paired corticomotoneuronal stimulation (PCMS) has a priming effect on motor learning. The question is an interesting one and relevant to novel neuromodulation strategies that might be used as part of therapies seeking to restore functions compromised by neurological injury. There are multiple considerations that need to be addressed and other aspects of experimental design and statistical reporting that require clarification to inform a decision regarding the soundness of this work.

1. The preponderance of studies show that the PCMS changes the efficacy of synapses between upper and lower motor neurons in the spinal cord. The title seems to suggest that learning is mediated in the spinal cord. Is strengthening of corticospinal-motoneuronal synapses via PCMS priming learning or driving it altogether?

2. Related to the notion of Hebbian plasticity, there is some reason to question findings from Experiment III used to support the interpretation of bidirectional changes in learning. The coupled-control condition is trending in a way that is different in the Rest group (Experiment I) and Sham group (Experiment II). Stated another way, it appears that there is some facilitatory effect of the coupled-control condition, creating uncertainty about whether it is an effective comparison for inferring a suppressive effect on performance. In addition, the interpretation statement beginning on line 53 indicates that PCMS- 'delayed acquisition,' yet there is only a difference relative to this coupled-control condition and not PCMS+. If changes related to the PCMS+ condition are evident from the beginning to end of practice, then would it also be reasonable to expect the same for PCMS-? Can the transient difference noted only in initial trials of a single session of practice objectively be interpreted as Hebbian plasticity? The finding that PCMS- did not decrease but, rather, increased corticospinal excitability further calls into question this interpretation.

3. Statistical reporting is unclear in multiple instances. The statistical models include TIME as a factor, and it is stated that there were 4 levels of this factor. Based on information contained in the text and scaling shown in the figures, it appears that all time points were normalized as a percentage of baseline performance. Can the authors clarify what values were entered into models for the baseline time point? Relatedly, why is the change in performance taken from the final block relative to the first block rather than relative to baseline? If performance was normalized to baseline for both groups in Experiment I, then which models account for the reported lack of baseline difference between groups? Figure 1C seems to suggest that baseline performance was not stable in the Rest group. The meaning of the statement on lines 119-121 that appears intended to account for this finding is a bit unclear. What was average difference in this group and what observations support that this difference was due to a decrease in vigilance? The implication seems to be that PCMS might be having some differential effect on arousal/alertness. Please clarify.

4. The effectiveness of this motor learning paradigm/task requires further justification. Given the lack of a temporal or spatial accuracy requirement, it is unclear whether changes in peak acceleration can be ascribed to learning vs. enhanced motor output shown in prior PCMS studies. What instruction was provided to the subject with regard to the performance goal for producing these finger movements? Was EMG recorded during practice blocks and was there any normalized change between conditions in any of the experiments? It also would be helpful to see 1-s recordings of the acceleration overlaid by block, particularly for Experiment III.

5. What do the results of Experiment I add given that Experiment II is largely a replication of Experiment I with sham stimulation, despite a smaller sample size? Can the authors provide further rationale as to why they chose between-subjects designs for Experiments I and II and a within-subject design for Experiment III? It was noted that order effects in Experiment III were accounted for by adding a term for testing day in mixed effects models, but there does not appear to be reporting to this end. Does the counterbalancing not effectively deal with order effects regardless of whether the baselines were different?

Miscellaneous:

6. Statement beginning on line 17 is quite lengthy and seems to suggest that spinal motor neurons are activated via direct electrical stimulation by way of PCMS. Please revise.

7. Statement beginning on line 21 is unclear. What do the authors mean by 'malleability' of neural circuits? Changes in transmission shown by prior PCMS studies might suggest that the CM synapse is malleable, no?

8. There is a lack of precision with regard to the hypothesis. What mechanisms of motor learning are primed by PCMS? The final common pathway is clearly integral to motor control, but why is it particularly important for learning?

9. It is unclear why/how rapid finger movements rely on spinal 'neuroplasticity' as stated on lines 28-29. Similarly, it is unclear how acquisition of ballistic index finger movements are contingent on spinal 'plasticity' as stated on lines 45-46. Perhaps, spinal mechanisms are the relevant feature?

10. How does one-week retention in neurologically-normal young adults expand 'clinical relevance' as stated on lines 48-51?

11. It would appear that the vertical dashed lines in Figures 1B, 2B, and 3B correspond to practice blocks, but it is unclear why the ticks do not align.

12. Were the accelerations produced into index finger abduction or flexion? Index finger flexion is specified throughout the manuscript, but this does not appear consistent with the prior work cited and also seems inconsistent with the depiction shown in Figure 1A.

13. Please enter units for what appear to be normalized MEP values reported on line 90.

14. Does increasing peak acceleration of a single finger movement reflect a change in 'dexterous motor functions' as stated on lines 99-102?

15. It is unclear what 'this' is referring to in the statement beginning on line 190. Results reported in this paragraph are a bit unclear.

16. Reporting on changes in normalized MEPs in Experiment III is also unclear. It seems that there was a baseline difference between conditions, but the coupled-control condition also showed a facilitation? Did the M-max remain constant? Reporting in the supplementary indicates that only 6/18 subjects showed decreased corticospinal excitability following the PCMS- protocol, but a significant within-group increase is reported. The traces in Figure 3C do not appear representative.

17. Note that a previous study by Taylor's group showed high-intensity, low-frequency TMS increased corticospinal excitability (D'Amico et al., 2020, J Neurophys).

Point-by-point Responses to Referees comments

Color coding:

- Responses to referees are color-coded **blue**
- Text included in 'Manuscript' and 'Supplementary Information' are color-coded **red**

Reviewer #1 (Remarks to the Author):

This manuscript reports an interesting series of studies that support the idea that priming the human motor pathway through a paired-stimulation plasticity protocol (PCMS) can influence later activity-dependent changes in motor output. The studies provide good evidence that a non-invasive protocol designed to elicit facilitatory spike-timing-dependent plasticity at corticospinal-motoneuronal synapses can increase the rate of improvement during subsequent practice of a fast movement. The evidence for a reduction of the rate of improvement with PCMS designed to depress synaptic efficacy is less compelling. Previously, PCMS applied in people with incomplete spinal cord injury has been shown to improve motor learning of a task requiring dexterity (nine-hole peg test; Bunday & Perez 2012) and PCMS in able-bodied people can immediately alter motor output in submaximal tasks (Taylor & Martin 2009; Bunday & Perez 2012). The current manuscript is novel in showing effects of PCMS on learning of rapid movements and also that the performance benefits from practice after facilitatory PCMS last for at least 7 days. As the practised movement engages other muscles besides those targeted with the paired-stimulation plasticity protocol, reporting of changes in first dorsal interosseous EMG that are consistent with the changes in finger acceleration would strengthen the argument that the protocols interact to alter motor learning with regard to this muscle in particular.

Authors' response: We thank the reviewer for the interest and the valuable suggestions and comments. Below you will find our point-by-point response.

We would like to respectfully highlight that Bunday & Perez (2012) investigated the effect on 9HPT performance and not motor learning. That said, when scrutinizing their contribution, it descriptively appears that the improvement occurring from repeated testing have slightly different slopes perhaps indicating a serendipitous effect of learning that was not explicitly assessed or investigated. Indeed, the study by Bunday & Perez (2012) did not investigate meta-plastic effects and effects on learning, but rather on performance. This means that the primary novelty of the present manuscript is that PCMS-induced effects interact with effects of motor practice to promote long-term motor learning.

We agree with the reviewer that it strengthens the manuscript to incorporate reporting of FDI EMG results, this has now been incorporated in the revised manuscript and the supplementary information. We would like to thank the reviewer for the constructive suggestion. Please find more details regarding the analysis of the EMG data in the specific question (Q2) below.

Major comments

1. The manuscript lacks an Introduction. Therefore, understanding the rationale requires considerable prior familiarity with paired corticospinal motoneuronal stimulation, activity-dependent plasticity, spike-timing-dependent plasticity, control of human movement and human neurophysiology.

Authors' response: Thank you for your comments on the Introduction. We agree that the lack of an Introduction required prior familiarity within the research field. We have now added an Introduction in line with journal formatting guidelines that introduces and elaborates on important concepts and the rationale for the experiments included in the manuscript:

Introduction

Across organisms, motor learning is governed by experience-dependent plasticity in relevant neural circuits along the neuroaxis dependent on the task demands (Shmuelof & Krakauer, 2011). These intrinsic processes of learning may be modulated extrinsically by non-invasive neuromodulation techniques. Paired corticomotoneuronal stimulation (PCMS) is an example of such technique used in humans (Taylor & Martin, 2009). By repeatedly pairing transcranial magnetic stimulation of the primary motor cortex (M1) and motoneuronal electrical stimulation timed to arrive at the corticomotoneuronal (CM) synapses in close temporal proximity, spike-timing-dependent, bidirectional changes in CM transmission can be induced (Caporale & Dan, 2008; Markram et al., 1997). PCMS-induced increases in CM transmission have been demonstrated to transiently improve motor control in patients with spinal cord injury (Bunday & Perez, 2012a; Urbin et al., 2017). However, it remains unknown whether PCMS-protocols can promote motor learning and how PCMS protocols interacts with mechanisms of experience-dependent plasticity remain to be explored.

In a series of three experiments, we investigated the effects of PCMS-induced plasticity on subsequent spinal motor learning in humans. We hypothesized that PCMS could prime mechanisms of subsequent motor learning exclusively when directed to the neural circuitry underpinning the motor behavior. Ballistic performance requires a high voluntary output and thereby a high firing rate in as many motor units as possible. Increased efficiency of CM-synapses may lead to a more effective input to the motor neurons, resulting in increased peak acceleration. Indeed, changes in neural transmission at the spinal cord level may particularly contribute to improvements during ballistic motor learning (Giesebrecht et al., 2012). In Experiment I, we used a PCMS protocol timed to induce plasticity of the CM-synapse with the aim of enhancing subsequent ballistic motor learning. In Experiment II, we expanded the relevance by replicating the findings from Experiment I in a double-blinded, sham-controlled experiment and investigated the behavioral effects of PCMS on ballistic motor learning in a retention test seven days after motor practice. Lastly, in Experiment III, we investigated whether the effects of PCMS on ballistic learning were bidirectional and circuit-specific, and whether only paired stimulations timed to facilitate corticomotoneuronal transmission increased learning compared to paired stimulations timed to depress corticomotoneuronal synaptic transmission.

We show that non-invasive paired neuromodulation targeting the spinal cord at the level of corticomotoneuronal synapses interacts with subsequent learning-induced neuroplasticity and leads to improved ballistic motor learning and coinciding increases in muscle activation and corticospinal excitability. The positive priming effects of PCMS on spinal motor learning persist seven days after motor

practice. This observation highlights the behavioral relevance of the findings. The observed priming effects on spinal motor learning are in line with Hebbian learning rules. (page 1-2)

2. The choice of index finger flexion rather than abduction complicates the interpretation of these studies. The long finger flexors are prime movers for index flexion but PCMS with peripheral nerve stimulation delivered at the wrist was designed to act on the ulnar innervated intrinsic muscles of the hand. Specifically, it was aimed at first dorsal interosseous (FDI). FDI does contribute to index flexion, as does the first palmar interosseous muscle which is also ulnar innervated and should also be influenced by the PCMS. However, the relative contribution of these ulnar-innervated intrinsic muscles versus the other muscles contributing to index flexion is uncertain. A report of FDI EMG during the ballistic movements would give more certainty that greater acceleration with training has a contribution from greater FDI activity and hence, that the effects of PCMS are consistent with the authors' interpretation.

Authors' response: We naturally agree with the reviewer, that not only the FDI muscle but also the long finger flexor muscles act as prime movers for index finger flexion. FDI does however act both as abductor and flexor. In pilot experiment, we tested ballistic performance and effects of ballistic motor practice for both abduction and flexion. We found that isolated abduction movements of the index finger were accompanied by more pronounced development of fatigue during repeated movements compared to flexions. This was reported by the participants but also evident from the participants' motor performance that reached a plateau. Based on this, we chose to focus on index finger flexion, since this experimental setup displayed expected learning curves and participants reported less fatigue. Furthermore, both index finger flexion and abduction rely on multiple muscles, meaning that one movement is not 'purer' than the other. We have now provided more information on the choice of index finger flexion for the ballistic task in the discussion of the manuscript:

"FDI is not the only muscle involved in index finger flexions, but we chose FDI because the ulnar nerve primarily supplies the motor innervation of the index finger. Thereby we can be fairly certain about the mechanistic route compared to what is previously seen with lower limb muscles, i.e., m. tibialis anterior (Urbin et al., 2017), namely that the timing of α -motoneuron activity is corrupted by early arrival of Ia afferent inputs or similar." (Page 10, lines 241-246)

We also agree with the reviewer, that it most certainly strengthens the manuscript and the interpretation of the results to present EMG analyses and investigate changes in FDI EMG with ballistic motor practice. We thank the reviewer for these valuable and constructive inputs, and now provide an analysis of the FDI EMG results during the execution of the ballistic movements throughout the experiment. First, details on how the EMG was analyzed have been added to the **Methods – Data analysis:**

"FDI EMG data obtained during the ballistic task was analyzed post hoc for Experiment I and II to investigate if EMG during ballistic contractions changed with motor practice. Rectified EMG data was filtered using a moving root-mean-square filter with a time

constant of 50 ms (similar to Aagaard et al., 2002). We calculated EMG root-mean square (EMG_{RMS}) amplitude in a time window from EMG onset to 70 ms after EMG onset. Additionally, rate of EMG rise was calculated from EMG onset to 30 ms. Both measures were normalized to the individual M_{max} amplitude, to allow comparison between sessions (see Supplementary Fig. 2, for EMG results).” (page 18, lines 502-510).

The EMG results are now briefly presented in the main text of the manuscript, with a reference to the supplementary material, where the results of the EMG analyses are presented in further details:

... Main text:

“Since the behavioral data was similar in Experiment I and II, we chose to pool data for acceleration and electromyographic (EMG) recordings from the two experiments, to investigate practice-induced changes in FDI muscle activity and whether changes in peak acceleration were correlated with changes in EMG. The EMG analyses demonstrated a significant main effect of TIME, and significant GROUP and TIME interactions for both EMG amplitudes and rate of EMG rise. Pairwise comparisons showed that PCMS led to larger increases in EMG during ballistic motor practice compared to controls. Furthermore, we found a significant positive correlation between relative change in acceleration and the relative change in rate of EMG rise (see Supplementary Notes and Supplementary Fig. 2 for details).” (page 7, lines 160-169).

And in further detail in the Supplementary Information:

Changes in muscle activity during ballistic motor practice

During the ballistic index finger flexions, EMG was recorded from m. FDI. Raw traces of acceleration and rectified EMG are shown in Supplementary Figure 2a. Since acceleration data showed similar trends in Experiment I and II, we chose to pool data for EMG analyses from the two experiments post hoc, allowing us to investigate effects of motor practice on muscle activity and to compare PCMS against Control (Rest & Sham). In line with previous studies using ballistic motor learning tasks, we chose to analyze EMG root-mean square amplitude (Giesebrecht et al., 2012; Rogasch et al., 2009), as well as the rate of EMG rise (an indirect measure of efferent neural drive)(Aagaard et al., 2002) for ballistic trials during baseline measurements and during block 1-3 of motor practice. For EMG_{RMS} amplitude (Supplementary Fig. 2b), the linear mixed model showed a significant main effect of TIME ($F_{(4,7989.1)}=10.21, p<0.001$), and a GROUP x TIME interaction ($F_{(4,7989.1)}=5.52, p<0.001$). Within TIME, comparisons showed a significant increase from baseline to practice Block 2 ($4.9\% \pm 1.8, p<0.001$) and from baseline to practice Block 3 ($9.0\% \pm 1.82, p<0.001$). Pairwise comparisons showed that PCMS led to a significantly larger increase in EMG_{RMS} amplitude from baseline to Block 2 (PCMS: $12.1\% \pm 3.4$ vs. Control: $2.6\% \pm 3.7, p<0.01$). For EMG measurements obtained at Day 7, both groups had maintained their EMG relative to baseline with no significant between-group difference (PCMS: $9.8\% \pm 3.7$ vs. SHAM: $10.6\% \pm 4.1, p=0.89$). For Rate of EMG rise (Supplementary Fig. 3d), the linear mixed model also showed a significant main effect of TIME ($F_{(4,7988.7)}=9.56, p<0.001$) and a

significant *GROUP x TIME* interaction ($F_{(4,7988.7)}=12.4$, $p<0.001$). Within *TIME*, comparisons showed a significant increase from baseline to practice Block 2 ($4.6\%\pm 2.9$, $p=0.02$). Pairwise comparisons showed that PCMS led to a significantly larger increase in rate of EMG rise from baseline to Block 3 (PCMS: $10.1\%\pm 5.8$ vs. Control: $-10.3\%\pm 6.3$, $p<0.01$). At Day 7 we found no significant between-group difference (PCMS: $12.2\%\pm 6.9$ vs. SHAM: $15\%\pm 6.3$, $p=0.6$). Note that day 7 results only include data from Experiment II. Finally, we investigated whether practice-induced changes in EMG measures correlated with practice-induced changes in peak acceleration. Pearson correlation tests showed a non-significant correlation between relative change in peak acceleration and relative change in EMG_{RMS} amplitude ($r=0.21$, $p=0.16$) and a significant positive correlation between the relative change in rate of EMG rise ($r=0.42$, $p<0.01$) and relative change in acceleration (Supplementary Fig. 2e). Collectively, these analyses demonstrate a positive effect of ballistic motor practice on FDI muscle activity and larger increases in EMG for PCMS compared to controls. Finally, the results demonstrate that the change in rate of EMG rise for FDI was related to behavioral changes in ballistic motor performance. (Supplementary Information, page 2-3)

Overall, the data demonstrates that FDI is involved during ballistic index finger flexions and that FDI EMG changes with motor practice. The results demonstrated that FDI EMG root-mean square amplitude increased during practice relative to baseline. As described, main effects of *TIME* and *GROUP x TIME* interactions were found. We did not observe a significant correlation between relative change in root-mean square EMG and change in acceleration, but we did observe a correlation between the relative changes in rate of EMG rise and relative change in acceleration, suggesting that changes in FDI EMG is behaviorally relevant for the task used in this experiment. The results thereby confirm the involvement of FDI in the task and that changes in FDI muscle activity is relevant for improvements in performance with motor practice. However, given that these correlations do not explain all of the variance associated with performance improvements, our results also suggest that other factors affected peak acceleration than FDI EMG. This is not a surprise, since FDI is not the only muscle involved in index finger flexions. In this context, it is also important to note that the changes in estimated EMG measures over time were generally variable across subjects. That said, we think that inclusion of the EMG data analyses as suggested by the reviewer has strengthened the manuscript. Thank you for your constructive inputs.

Supplementary Figure 2. Change in EMG during rapid index finger flexions. **a)** Exemplary single subject data, raw traces of acceleration and EMG (rectified) during the rapid finger flexion. **b)** Mean EMG_{RMS} amplitude (% baseline), for both condition PCMS (blue) and Control (grey), each data point represent individual data at a given time point. **c)** Pearson correlation between change in EMG_{RMS} amplitude and the change in peak acceleration from the last 10 trials of practice relative to baseline. **d)** Mean rate of EMG rise (% baseline), for both condition PCMS (blue) and Control (grey). **e)** Pearson correlation between change in rate of EMG rise and the change in peak acceleration from the last 10 trials of practice relative to baseline. Error bars represent standard deviation.

3. Experiment III has a within-subject cross-over design in which individual participants underwent three different interventions at one week intervals. As Experiment II showed that the learning effect of repeated movements lasted for at least one week, the rationale for the cross-over design with one week washout is unclear. The design seems likely to reduce possible differences between interventions given that increases in movement acceleration must eventually reach a plateau. Nonetheless, differences between interventions did occur. However, discussion of the rationale and/or possible influence of research design is warranted.

Authors' response: This is indeed an important point, and we have now elaborated on the rationale for using this experimental design the *Methods* section.

We chose a within-subject design for Experiment III, to try to limit inter-individual variability between groups. Based on previous literature (Rogasch et al., 2009; Rosenkranz et al., 2007) we rationalized that our participants would not reach a plateau in motor performance after just three motor practice sessions (with 150 trials in each) spaced with one week. (page 14, lines 391-395).

In brief, Experiment II indeed showed learning after one week. Based on this, we also expected that retention of learning would be present after one week in Experiment III. However, based on findings from Rosenkranz et al., (2007), where ballistic motor learning was also investigated, with subjects practicing over 5 sessions across 5 consecutive days (Monday to Friday), with continued performance gains after each day, we also expected that performance would not plateau. Moreover, previous studies

quantifying peak acceleration during ballistic motor practice, i.e. Rogasch et al., (2009), used twice as many practice trials as we did within a session (300 practice trials, vs. the 150 trials in our Experiments). They did not show a plateau, thereby not showing signs of saturation after 300 trials within one session.

Based on these findings, we rationalized that our participants would not reach a plateau after just 3 sessions (with 150 trials in each) of motor practice spaced with one week. When revisiting our motor learning data, the learning rate was, however, in fact steeper at day 1 compared to the following 2 sessions. Importantly, however, and as mentioned by the reviewer, we were still able to detect differences between our experimental manipulations. Indeed, it is plausible that our within-subject design did mask or reduce some of the effect. Below, we have shown a figure illustrating data from the participants' first session. The effect size seemed greater when only comparing data from individuals' first session. Notably, this seemed to be the case for the facilitatory effects of PCMS+ compared to PCMS_{coupled-control}, but also for the impediment of performance for PCMS- compared to PCMS_{coupled-control}. However, please note that the figures below only contain data from 6 participants per group, since Experiment III involved 18 participants and thus had 6 participants for each intervention at Day 1.

Based on the constructive comment from the reviewer, we are now considering waiting more time between sessions in future studies to allow for larger washout between sessions and avoid diminishing effect sizes.

Minor comments

There seem to be no stimulation intensities given for either TMS or peripheral nerve stimulation when they are delivered as PCMS.

Authors' response: Thank you for noticing this. Information on stimulation intensities have now been added to the *Methods*:

“The PCMS protocols consisted of 100 paired TMS (intensity at 150% rMT) and PNS (intensity at 130% of Mmax stimulation) with a frequency of 0.1 Hz” (page 17, lines 483-488).

“+” symbol appears before some values e.g. lines 70, 138, 157, 223

Authors' response: The + was intended to symbolize an increase. For clarity, we have chosen to remove all “+” symbols when reporting results to avoid potential misunderstandings.

All Figure legends. No descriptions of what the error bars represent.

Authors' response: Error bars in all Figures represent Standard Deviations. We apologize for the omission of this detail. This has now been added to the Figure Legends. Thank you for bringing to our attention.

Figure 2B. Problem with the y-axis title – missing letters at the start of acceleration.

Authors' response: We have now corrected the y-axis title. Thank you for noticing this.

Line 157. Not clear what 120.58% and 115.56% are percentages of.

Authors' response: This has now been clarified. 100 = baseline. This has now been clarified in the manuscript and in the figures to avoid misunderstandings.

Line 162 and elsewhere. “Despite of” should be “Despite”

Authors' response: We have now corrected this to “Despite”. Thank you.

Lines 224-239. A strong conclusion “the priming effects of PCMS on ballistic motor learning are bidirectional depending on the spike-time” is drawn but without discussion of the result that the decrease of motor learning might only last through one block of practice.

Authors' response: Regarding our statement of bidirectional plasticity, we acknowledge that both reviewers find this statement too strong given the behavioral data supporting it. We argue that the difference noted in practice block 1 in PCMS- compared to the other conditions can be interpreted as an effect of Hebbian plasticity. We do however also agree that more evidence is needed to solidify the finding.

We found a statistically significant difference in performance between PCMS- and the two other protocols (PCMS+ and PCMS_{coupled-control}) throughout the first practice block (Figure 3b, bar chart showing Block 1 (B1)). That is, in B1, PCMS- performed poorer compared to both PCMS+ and PCMS_{coupled-control}. We argue that the difference noted

in initial trials of practice B1 in PCMS- compared to the other conditions can be interpreted as Hebbian plasticity. The priming effect of the PCMS+ became evident in B2 and B3. This is the first combination of a down-conditioning protocol targeting spinal spike-timing dependent like plasticity with ‘spinal learning’ so it is fairly uncharted territory in regards of the temporal extent of the effects. Previous research on operant learning involving the spinal circuitries have found that the behavioral manifestations of protocols thought to induce detrimental changes in the circuitry are rare (J.R. Wolpaw, negotiated equilibrium (Wolpaw, 2018)). Specifically, down-conditioning the soleus stretch and H-reflex improves gait function in SCI by reducing co-contractions impacting the ankle angle during the swing phase. Reversely, up-conditioning the same reflects does not compromise walking. Instead other joints compensate. In our set-up, it can be speculated that increases in CM connectivity to motoneurons innervating the digit flexors over the course of practice could have compensated for the effects of PCMS- on synaptic malleability of the FDI CM connections. See also response to Comment 3, regarding Experiment III and masking of within-subject design. In Brief, we want to remind that the effect size seemed greater when only comparing data from individuals’ first session. Notably, this seemed to be the case for the facilitatory effects of PCMS+ compared to PCMS_{coupled-control}, but also for the impediment of performance for PCMS- compared to PCMS_{coupled-control} (However, please note that the above figure only contain data from 6 participants per group, since Experiment III involved 18 participants and thus had 6 participants for each intervention at Day 1). Based on the constructive comment from the reviewer, we are now considering waiting more time between sessions in future studies to allow for larger washout between sessions and avoid diminishing effect sizes.

In summary, we agree with the reviewer that our initial interpretation of Hebbian bidirectional plasticity was a bit too strong based on our data. We have now modified these statements in a broader discussion rather than a conclusive statement.

“It should be noted that this negative effect on learning of PCMS- was only observed during early learning, which indicates a smaller or less consistent effect of PCMS- on ballistic motor learning compared to PCMS+. The smaller effect of PCMS- may also in part be influenced by effect sizes being diluted by the within-subject design and carry-over effects at the seven-day retention test. In summary, the effects observed in Experiment III conform to Hebbian learning rules in that they are contingent on temporal proximity of stimulation effects and governed by order of spike-timing.” (page 11, lines 276-283).

Lines 232 & 262 and elsewhere. The use of “inhibitory effect” to describe the effect of PCMS- should be carefully reviewed as the plastic change postulated is a down regulation of facilitatory synaptic input rather than a change in inhibition.

Authors’ response: Thank you, we have now elaborated on the use of “inhibitory effect” throughout the manuscripts and changed the wording to avoid misunderstanding.

Line 267. Maybe “CM IAI” should be “CM ISI”. Also line 272.

Authors’ response: Here, we actually think that IAI (interarrival interval) and not ISI (interstimulus interval) is more appropriate. Importantly, IAIs reflect the differences in the arrival of evoked volleys at the synaptic-network level whereas ISIs reflect the time between the delivery of pulses, which do not necessarily correspond to each other in PCMS studies due to differences in transmission times. For example, while the interarrival time of volleys at the CM synapse is -2 ms, +15 ms and +100 ms in PCMS+, PCMS- and PCMScoupled-control, the ISIs are 5.5 ms, 22.5 ms and 107.2 ms on average, respectively. Therefore, for the instances referenced above, we argue that CM IAI is the correct term because we refer to the arrival of volleys at the spinal level that are mediators of the protocols’ effects.

Line 290. Supplementary information says that TMS at 0.1 Hz was at 120% RMT. That is not really “high intensity”

Authors’ response: Thank you for bringing this to our attention. We acknowledge that the initial wording might have caused confusion. The 120% rMT mentioned in the Supplementary Information refers to the intensity used to measure MEPs after the stimulation protocol. The stimulation intensity used during rTMS (0.1Hz) was 150% rMT. We have now clarified this in the Supplementary information.

Supplementary information

PCMS protocols section does not mention stimulus intensities Effects of non-paired rPNS or rTMS section does not mention rPNS intensity. Responders/ Non-responders section indicates that TMS was given at 150% RMT during PCMS. This means that the control testing for MEP increases with non-paired rTMS should have occurred with 150% RMT. This is not obvious from the description of the control experiment.

Authors’ response: Thank you for bringing this to our attention. We have now added information about stimulation intensities throughout method section. TMS (intensity at 150% rMT) or PNS (intensity at 130% of M_{max} stimulation intensity).

Reviewer #2 (Remarks to the Author):

Experiments reported in this manuscript examined whether paired corticomotoneuronal stimulation (PCMS) has a priming effect on motor learning. The question is an interesting one and relevant to novel neuromodulation strategies that might be used as part of therapies seeking to restore functions compromised by neurological injury. There are multiple considerations that need to be addressed and other aspects of experimental design and statistical reporting that require clarification to inform a decision regarding the soundness of this work.

Authors’ response: We thank the reviewer for the interest and the constructive and valuable suggestions and comments. Below you will find our point-by-point response.

1. The preponderance of studies show that the PCMS changes the efficacy of synapses between upper and lower motor neurons in the spinal cord. The title seems to suggest that learning is mediated in the spinal cord. Is strengthening of corticospinal-motoneuronal synapses via PCMS priming learning or driving it altogether?

Authors' response: We thank the reviewer for the opportunity to address this question in depth and we hope that we have understood the question correctly. Indeed, previous studies have shown that practicing a ballistic motor task similar to the one used in the present study is accompanied by changes in CMEPs (Giesebrecht et al., 2012). This suggests that short-term changes in transmission across corticomotoneuronal synapses likely provide an important neural substrate for the improvement in the recruitment of motoneurons that results in increased acceleration of the fingers in a fixed direction. As we do not observe changes in ballistic performance immediately after PCMS+ and as subjects learn (improve ballistic performance with practice) also after the sham protocol or rest, we argue that PCMS+ primes this form of ballistic learning. In essence, we believe that ballistic motor learning involves plasticity at the level of the corticomotoneuronal synapses for all groups, but since PCMS+ induced plasticity prior to ballistic motor practice. That is, PCMS+ primed subsequent learning (Karabanov et al., 2015). This provides evidence for an interaction between stimulation-induced and experience-dependent plasticity.

We agree that several studies have (indirectly) shown that PCMS changes the efficacy of corticomotoneuronal synapses in the spinal cord (Bunday & Perez, 2012b; Fitzpatrick et al., 2016; Taylor & Martin, 2009). Many of these studies show that PCMS can have a positive effect on motor *performance* indexed as 'voluntary motor output' (Bunday & Perez, 2012b; Taylor & Martin, 2009; Urbin et al., 2017) and performance in a 9HPT test (Bunday & Perez 2012) We wanted to test the interaction of PCMS effects and subsequent motor *practice* to improve motor *learning*. To do this, we intentionally chose a motor learning paradigm also relying on changes at the segmental level (most likely corticomotoneuronal synapses) in the spinal cord (Giesebrecht et al., 2012). Because of this, we argue that in our Experiments, a large part of the learning is likely mediated by short-term modulation of networks in the spinal cord. In all three Experiments, PCMS+ improves learning to a greater extent compared to the control protocols (Experiment I: Rest; Experiment II: Sham; Experiment III: PCMS_{coupled-control}). Importantly, all these control protocols show significant within-group improvements in performance (increased acceleration of index finger flexions after motor practice). We argue that these findings indicate a priming effect of PCMS+, rather than driving learning all together (since control protocols also improved learning significantly compared to the baseline performance). This shows the importance of the interaction of PCMS and motor practice on motor learning and supports our use of 'Priming' in the title.

Regarding the spinal contribution to motor learning, we would like to add, that we now elaborate on this in the **Discussion**:

“Without any directional movement constraints, peak acceleration in a specific direction can be increased by optimizing movement direction (likely a cortical phenomenon) as well as acceleration per se by improving fast, coordinated activation of agonist spinal motoneurons. In line with previous experiments, we confined ballistic index finger movements to one plane to emphasize the spinal contribution to learning (Giesebrecht et al., 2012). Consequently, the observed increase in peak acceleration in the present study can be assumed to largely reflect improved efficacy of direct activation of spinal motoneurons.” (page 9, lines 229-236)

2. Related to the notion of Hebbian plasticity, there is some reason to question findings from Experiment III used to support the interpretation of bidirectional changes in learning. The coupled-control condition is trending in a way that is different in the Rest group (Experiment I) and Sham group (Experiment II). Stated another way, it appears that there is some facilitatory effect of the coupled-control condition, creating uncertainty about whether it is an effective comparison for inferring a suppressive effect on performance. In addition, the interpretation statement beginning on line 53 indicates that PCMS- 'delayed acquisition,' yet there is only a difference relative to this coupled-control condition and not PCMS+. If changes related to the PCMS+ condition are evident from the beginning to end of practice, then would it also be reasonable to expect the same for PCMS-? Can the transient difference noted only in initial trials of a single session of practice objectively be interpreted as Hebbian plasticity? The finding that PCMS- did not decrease but, rather, increased corticospinal excitability further calls into question this interpretation.

Authors' response: Thank you for your important reflections. We will address these one at the time.

First, given that the experimental design is different between experiments (between-subject vs. within-subject), we argue that it is difficult to compare the trajectories of changes in motor performance between Experiment I, II and III. As Experiment III is designed as a within-subject, cross-over experiment, the B1 performance does not reflect 'naïve performance levels' to the task for 2/3 of the subjects, due to potential carry-over effects from previous experimental sessions. That said, statistically there are no significant differences in learning from B1 to B3 between 'PCMS_{coupled-control}', 'Rest' and 'Sham' conditions (Rest: 10.93%±1.0; SHAM: 11.75%±0.8; PCMS_{coupled-control} 8.26%±0.8) (Figure 1C, 2C, 3C – and see below Figure for easier comparison). This suggests that all three conditions are comparable in terms of changes in performance from B1 to B3, suggesting that the PCMS_{coupled-control} condition is indeed a representative 'control' condition in terms of motor performance for Experiment III.

Second, regarding whether PCMS delayed acquisition, we would like to respectfully note that there was in fact a statistically significant difference in performance between PCMS- and the two other protocols (PCMS+ and PCMS_{coupled-control}) throughout the first practice block (Figure3b, bar chart showing Block 1 (B1)). That is, in B1, PCMS- performed poorer compared to both PCMS+ and PCMS_{coupled-control}. We argue that the difference noted in initial trials of practice B1 in PCMS- compared to the other conditions can be interpreted as Hebbian plasticity.

Collectively, we think that the arguments provided above speaks in favor of a bidirectional, Hebbian effect on motor performance / motor learning measures. We do however agree with the reviewer that more evidence is needed to further solidify the finding on bidirectionality. We also acknowledge that both reviewers find our statement too strong given the available behavioral data. We have therefore now modified our statements in a broader discussion rather than a conclusive statement:

“It should be noted that this negative effect on learning of PCMS- was only observed during early learning, which indicates a smaller or less consistent effect of PCMS- on ballistic motor learning compared to PCMS+. The smaller effect of PCMS- may also in part be influenced by effect sizes being diluted by the within-subject design and carry-over effects at the seven-day retention test. In summary, the effects observed in Experiment III conform to Hebbian learning rules in that they are contingent on temporal proximity of stimulation effects and governed by order of spike-timing.” (page 11, lines 276-283).

Finally, we agree with the reviewer that the facilitatory effect on corticospinal excitability after PCMS- was unexpected compared to previous literature (Bunday & Perez, 2012b). However, as we discuss in the main text of the manuscript, the use of MEPs to quantify corticospinal excitability could potentially mask a suppressive effect at the

spinal level. Indeed, based on inter-stimulus intervals of 22.5 ms on average, we speculate that it is changes at the cortical level in networks also involved in mediating effects of Paired Associative Stimulation (PAS) protocols that drive the effects observed on MEPs (given that MEPs are sensitive to changes across different levels of the corticomotor system). It is also important to note in this regard, that there might be different temporal trajectories for the effects of 'facilitatory' and 'inhibitory' PCMS protocols on excitability. Indeed, Taylor & Martin, (2009) showed different timelines for facilitatory and inhibitory PCMS protocols. Facilitatory PCMS protocol displayed increases in excitability emerging just after PCMS and lasting ~30 minutes. In contrast, inhibitory PCMS protocols showed decreased excitability effects from 30 to 60 minutes after stimulations. Importantly, Taylor & Martin targeted a different muscle and used a different outcome measure (MEPs in our study compared to CMEPs in the Taylor & Martin (2009) study). Direct comparisons should therefore be made with caution. However, their findings suggest that a PCMS+ and PCMS- protocol do not necessarily lead to similar time courses of effects on excitability, which could also be the case for the current data.

3. Statistical reporting is unclear in multiple instances. The statistical models include TIME as a factor, and it is stated that there were 4 levels of this factor. Based on information contained in the text and scaling shown in the figures, it appears that all time points were normalized as a percentage of baseline performance. Can the authors clarify what values were entered into models for the baseline time point? Relatedly, why is the change in performance taken from the final block relative to the first block rather than relative to baseline? If performance was normalized to baseline for both groups in Experiment I, then which models account for the reported lack of baseline difference between groups? Figure 1C seems to suggest that baseline performance was not stable in the Rest group. The meaning of the statement on lines 119-121 that appears intended to account for this finding is a bit unclear. What was average difference in this group and what observations support that this difference was due to a decrease in vigilance? The implication seems to be that PCMS might be having some differential effect on arousal/alertness. Please clarify.

Authors' response: We agree with the reviewer and we have now clarified the abovementioned instances and incorporated more thorough explanations. We chose to visualize the behavioral data in bins of 10 trials to provide a better representation of the learning curve. The baseline test was only thought as providing a measurement to normalize to i.e. for comparison. The statistical analysis was based on the comparison of the entire blocks. We included three levels of the TIME factor in Experiment I (Baseline, Block 1-3). Each subject then had a baseline value of 100. Indeed this approach reduce variance between individuals. If any main effects including interaction were significant, we proceeded to pairwise comparisons. As mentioned in the "Statistical Analysis" section we computed contrasts to test specific hypotheses. As shown in the inserted bar chart in Figure 1B, we chose specifically to test the relative change in acceleration for each practice block. This allowed us to compare the effect of PCMS on the practice blocks, which included augmented feedback after each trial.

Effectively, this means that the baseline block was included to allow us to normalize performance measures, and it allowed us to assess acute effects of PCMS on motor performance. Importantly, during baseline blocks, no augmented feedback was provided.

For the sake of clarity, we have now re-run our statistical analysis with baseline removed from the model. Also in this case, the results still show significant main effects including interactions as presented in the Main text. We have presented the interaction effects for models with removed baseline below:

Experiment I: Linear mixed effects model, GROUP (2 levels: PCMS and Rest) and TIME (3 levels: B1-B3) showing a significant GROUP x TIME interaction ($F_{(2,3799)}=35.1$, $p<0.001$) on peak index finger acceleration.

Experiment II: Linear mixed effects model, GROUP (2 levels: PCMS and Rest) and TIME (4 levels: B1-B3, Day 7) showing a significant GROUP x TIME interaction ($F_{(3,3327)}=18.7$, $p<0.001$) on peak index finger acceleration.

Experiment II: Linear mixed effects model, PROTOCOL (3 levels: PCMS+, PCMS-, PCMScoupled-control) and TIME (3 levels: B1-B3) and DAY (3 levels: Day 1-3), showing a significant GROUP x TIME interaction ($F_{(6,8477)}=12.0$, $p<0.001$) on peak index finger acceleration.

Since we use a relatively novel motor learning paradigm, we wanted to report the individual trajectories for the behavioral effect (Figure 1,2,3C). We agree with the Reviewer that the reporting of this analysis was insufficient in the “Statistical Analysis”, and this has now been added: *“We also tested in a new model the relative change from the first 10 trials of Block 1 to the last 10 trials of Block 3 Figure 1C)”* (page 18, line 524-525). Regarding Figure 1C, the Reviewer suggests that baseline performance was not stable in the Rest group. We want to clarify that there is variability between individuals in baseline performance, we average to the individual baseline performance as a stable ‘set-point’. We chose a baseline normalization in this task, first, because this is what the participants received augmented feedback on during motor practice (i.e. scores expressed as a % of baseline performance), and secondly, because there is inter-individual variability in absolute acceleration.

In relation to the statement on lines 119-121, we agree with the Reviewer and have now changed the lines:

“Furthermore, we cannot rule out that the observed between-group difference in average performance in the first part of block 1 of motor practice could be due to a placebo effect since the design of Experiment I was not blinded nor placebo or sham-controlled” (page 5, lines 112-115).

This argues for an optimized testing protocol and control condition, and therefore, we proceeded to Experiment II, where we included a sham control to better match conditions:

“To investigate these questions, we introduced a sham protocol that mimicked the perceptual experience of PCMS+ in Experiment II. This experiment aimed to replicate our results from Experiment I in a double-blinded, sham-controlled experimental design. Additionally, Experiment II included a 1-week retention test to assess the long-term effects of Hebbian priming on spinal motor learning.” (page 5, lines 115-120).

4. The effectiveness of this motor learning paradigm/task requires further justification. Given the lack of a temporal or spatial accuracy requirement, it is unclear whether changes in peak acceleration can be ascribed to learning vs. enhanced motor output shown in prior PCMS studies. What instruction was provided to the subject with regard to the performance goal for producing these finger movements? Was EMG recorded during practice blocks and was there any normalized change between conditions in any of the experiments? It also would be helpful to see 1-s recordings of the acceleration overlaid by block, particularly for Experiment III.

Authors' response: Thank you for your comments. We will address your questions one at a time.

Regarding choice of motor learning paradigm. We argue that the performance increases with practice in our model, are similar to the other groups who used a restrictive splint, and that these changes can be classified as learning (Giesebrecht et al., 2012; Rogasch et al., 2009). This type of motor learning seems to be mediated by increases in corticomotoneuronal excitability (Giesebrecht et al., 2012). In our study, performance improved gradually over the course of motor practice. That is, motor performance improved gradually from B1 to B3. This indicates a learning effect and displays the properties constituting prototypical performance/learning curves (Kantak & Winstein, 2012; Krakauer et al., 2019).

We consider it unlikely that effects can be ascribed solely to enhanced motor output, because (1) effects persist and increase over time, i.e., differences between primed PCMS groups and Rest or Sham increase over time (Figure 1B and 2B in Main Text), while (2) previous studies demonstrating effects of PCMS on motor output appeared immediately after PCMS (Taylor & Martin, 2009).

Furthermore, although we cannot conclusively exclude the possibility that the effect of PCMS could in part relate to a delayed effect on voluntary motor output, the finding from Experiment II that the beneficial effect persists 7 days later strengthens further the argumentation for a learning effect (Kantak & Winstein, 2012).

Regarding instructions to participants. More detail on the instructions have been added to the *Methods* section:

“Participants were instructed to perform the movements as fast as possible, and to keep improving their score during the practice blocks” (page 15, lines 426-427).

Regarding the 'temporal aspect': Participants were visually *cued* by a vertical line and told to try to perform the movement *around* that. Importantly, however, no feedback was provided on this aspect of the task. The task focused *solely* on producing the largest possible acceleration and this was the only objective. That is, there is in fact a temporal cueing, but this is not the important part of this task.

Considering analysis of EMG: We thank the reviewer for the constructive input on including EMG analyses. We agree that these analyses can indeed strengthen the manuscript and underline the relevance of investigating FDI effects. We have now added a report of FDI EMG during the ballistic movements. Overall, the results of these analyses show that FDI is involved during the ballistic index finger flexions. As described in detail below, the results demonstrated that FDI EMG root-mean square amplitude and the rate of EMG rise increased during practice relative to baseline. As described, the analysis demonstrated main effects of TIME, and GROUP x TIME interactions for both measures. Furthermore, we observed a correlation between the relative changes in rate of EMG rise and relative change in acceleration, but not between EMG_{RMS} and changes in acceleration. In general, although we also observed high inter-individual variability in EMG responses, these results show the involvement of FDI in the task and behavioral changes with motor practice, but also suggest that other factors affected peak acceleration than FDI EMG. This is not a surprise, since FDI is not the only muscle involved in index finger flexions. In what follows below, we describe the changes made in the new version of the manuscript:

First, details on the EMG analysis has been added to the **Methods – Data analysis**:
“FDI EMG data obtained during the ballistic task was analyzed post hoc for Experiment I and II to investigate if EMG during ballistic contractions changed with motor practice. Rectified EMG data was filtered using a moving root-mean-square filter with a time constant of 50 ms (similar to Aagaard et al., 2002). We calculated EMG root-mean square (EMG_{RMS}) amplitude in a time window from EMG onset to 70 ms after EMG onset. Additionally, rate of EMG rise was calculated from EMG onset to 30 ms. Both measures were normalized to the individual M_{max} amplitude, to allow comparison between sessions (see Supplementary Fig. 2, for EMG results).“
(page 17, lines 502-510).

The EMG results are now briefly presented in the main text of the manuscript, with a reference to the supplementary Information, in which the results are presented in detail and depicted:

... Main text:

“Since the behavioral data was similar in Experiment I and II, we chose to pool data for acceleration and electromyographic (EMG) recordings from the two experiments, to investigate practice-induced changes in FDI muscle activity and whether changes in peak acceleration were correlated with changes in EMG. The EMG analyses demonstrated a significant main effect of TIME, and significant GROUP and TIME interactions for both EMG amplitudes and rate of EMG rise. Pairwise comparisons showed that PCMS led to larger increases in EMG during ballistic motor practice

compared to controls. Furthermore, we found a significant positive correlation between relative change in acceleration and the relative change in rate of EMG rise (see Supplementary Notes and Supplementary Fig. 2 for details).” (page 7, lines 164-173).

And in more detail in the Supplementary Information:

“Changes in muscle activity during ballistic motor practice

During the ballistic index finger flexions, EMG was recorded from *m. FDI*. Raw traces of acceleration and rectified EMG are shown in Supplementary Figure 2a. Since acceleration data showed similar trends in Experiment I and II, we chose to pool data for EMG analyses from the two experiments post hoc, allowing us to investigate effects of motor practice on muscle activity and to compare PCMS against Control (Rest & Sham). In line with previous studies using ballistic motor learning tasks, we chose to analyze EMG root-mean square amplitude (Giesebrecht et al., 2012; Rogasch et al., 2009), as well as the rate of EMG rise (an indirect measure of efferent neural drive)(Aagaard et al., 2002) for ballistic trials during baseline measurements and during block 1-3 of motor practice. For EMG_{RMS} amplitude (Supplementary Fig. 2b), the linear mixed model showed a significant main effect of TIME ($F_{(4,7989.1)}=10.21$, $p<0.001$), and a GROUP x TIME interaction ($F_{(4,7989.1)}=5.52$, $p<0.001$). Within TIME, comparisons showed a significant increase from baseline to practice Block 2 ($4.9\%\pm 1.8$, $p<0.001$) and from baseline to practice Block 3 ($9.0\%\pm 1.82$, $p<0.001$). Pairwise comparisons showed that PCMS led to a significantly larger increase in EMG_{RMS} amplitude from baseline to Block 2 (PCMS: $12.1\%\pm 3.4$ vs. Control: $2.6\%\pm 3.7$, $p<0.01$). For EMG measurements obtained at Day 7, both groups had maintained their EMG relative to baseline with no significant between-group difference (PCMS: $9.8\%\pm 3.7$ vs. SHAM: $10.6\%\pm 4.1$, $p=0.89$). For Rate of EMG rise (Supplementary Fig. 3d), the linear mixed model also showed a significant main effect of TIME ($F_{(4,7988.7)}=9.56$, $p<0.001$) and a significant GROUP x TIME interaction ($F_{(4,7988.7)}=12.4$, $p<0.001$). Within TIME, comparisons showed a significant increase from baseline to practice Block 2 ($4.6\%\pm 2.9$, $p=0.02$). Pairwise comparisons showed that PCMS led to a significantly larger increase in rate of EMG rise from baseline to Block 3 (PCMS: $10.1\%\pm 5.8$ vs. Control: $-10.3\%\pm 6.3$, $p<0.01$). At Day 7 we found no significant between-group difference (PCMS: $12.2\%\pm 6.9$ vs. SHAM: $15\%\pm 6.3$, $p=0.6$). Note that day 7 results only include data from Experiment II. Finally, we investigated whether practice-induced changes in EMG measures correlated with practice-induced changes in peak acceleration. Pearson correlation tests showed a non-significant correlation between relative change in peak acceleration and relative change in EMG_{RMS} amplitude ($r=0.21$, $p=0.16$) and a significant positive correlation between the relative change in rate of EMG rise ($r=0.42$, $p<0.01$) and relative change in acceleration (Supplementary Fig. 2e). Collectively, these analyses demonstrate a positive effect of ballistic motor practice on *FDI* muscle activity and larger increases in EMG for PCMS compared to controls. Finally, the results demonstrate that the change in rate of EMG rise for *FDI* was related to behavioral changes in ballistic motor performance.

”(Supplementary Information, page 2-4)

Supplementary Figure 2. Change in EMG during rapid index finger flexions. a) Exemplary single subject data, raw traces of acceleration and EMG (rectified) during the rapid finger flexion. b) Mean EMG_{RMS} amplitude (% baseline), for both condition PCMS (blue) and Control (grey), each data point represent individual data at a given time point. c) Pearson correlation between change in EMG_{RMS} amplitude and the change in peak acceleration from the last 10 trials of practice relative to baseline. d) Mean rate of EMG rise (% baseline), for both condition PCMS (blue) and Control (grey). e) Pearson correlation between change in rate of EMG rise and the change in peak acceleration from the last 10 trials of practice relative to baseline. Error bars represent standard deviation.

We believe that inclusion of the EMG data analysis as suggested by the reviewer has strengthened the manuscript and would like to thank you for your inputs.

The reviewer asked to see acceleration traces recorded during the ballistic task. As requested, the figure inserted below shows single subject acceleration traces from each protocol from each block. The figure is not included in the manuscript but provides the reviewer with information on these recordings and examples of traces.

5. What do the results of Experiment I add given that Experiment II is largely a replication of Experiment I with sham stimulation, despite a smaller sample size? Can the authors provide further rationale as to why they chose between-subjects designs for Experiments I and II and a within-subject design for Experiment III? It was noted that order effects in Experiment III were accounted for by adding a term for testing day in mixed effects models, but there does not appear to be reporting to this end. Does the counterbalancing not effectively deal with order effects regardless of whether the baselines were different?

Authors' response: Experiment I provided the initial evidence of a positive effect of PCMS on motor learning. Due to our novel motor learning paradigm, we wanted to compare the effects of PCMS before motor practice to motor practice without prior stimulations. A resting/non-stimulation control protocol was therefore an obvious first step of the research project. We want to emphasize that the primary aim of the double-blinded, sham-controlled Experiment II was indeed to replicate findings from Experiment I, but in a double-blinded sham-controlled experiment. Since Experiment I was performed prior to Experiment II, we have included Experiment I in the manuscript to transparently present all collected data. We further believe that showing replication of our behavioral results in an independent sample strengthens the interpretation of our effects.

We chose a within-subject design for Experiment III, to try to limit inter-individual variability between groups. Since we wanted to test effects of three different stimulation protocols, the decision also had a pragmatic element to it, since a within-subject design

required fewer participants i.e. 18 instead of 54. The main argument for this decision was however to limit interindividual variability.

We have now added further information on results from main effect of 'day' from the statistical models (see also earlier response to Reviewer 1). As expected, statistical analysis showed a main effect of DAY ($F_{(2,8477)}=641$, $p<0.001$). Indeed the counterbalancing effectively dealt with order effects, and we still see the significant interaction effect in the performed analysis and reported results. Furthermore, we still observed increases in acceleration with practice in session 2 and 3, indicating that learning was not saturated.

Experiment II indeed showed learning after one week. Based on this, we also expected that retention of learning would be present after one week in Experiment III. However, based on findings from Rosenkranz et al., (2007), where ballistic motor learning was also investigated, with subjects practicing over 5 sessions across 5 consecutive days (Monday to Friday), with continued performance gains after each day, we also expected that performance would not plateau. Moreover, previous studies quantifying peak acceleration during ballistic motor practice, i.e. Rogasch et al., (2009), used twice as many practice trials as we did within a session (300 practice trials, vs. the 150 trials in our Experiments). They did not show a plateau, thereby not showing signs of saturation after 300 trials within one session.

Based on these findings, we rationalized that our participants would not reach a plateau after just 3 sessions (with 150 trials in each) of motor practice spaced with one week. This was also what we observed when revisiting our motor learning data: although the learning rate was steeper at day 1 compared to the following 2 sessions, we still observed improvements in performance across all test sessions. Furthermore, and importantly, we were still able to detect differences between our experimental manipulations despite of these carry over effects on performance. Indeed, it is plausible that our within-subject design did mask some of the effect. Below, we have shown a figure illustrating data from the participants' first session parsed by group. The effect sizes of our experimental manipulations (PCMS+, PCMS-) seemed greater when only comparing data from individuals' first session. Notably, this seemed to be the case for the facilitatory effects of PCMS+ compared to PCMS_{coupled-control}, but also for the impediment of performance for PCMS- compared to PCMS_{coupled-control}. However, please note that the figures below only contain data from 6 participants per group, since Experiment III involved 18 participants and thus had 6 participants for each intervention at Day 1.

Experiment III: Session 1 & baseline on Session 2 (~Day 7 retention)
 Note: Only 6 participants in each group

Miscellaneous:

6. Statement beginning on line 17 is quite lengthy and seems to suggest that spinal motor neurons are activated via direct electrical stimulation by way of PCMS. Please revise.

Authors' response: Thank you for this input, this statement has now been revised: *“Paired corticomotoneuronal stimulation (PCMS) is an example of such technique used in humans(Taylor & Martin, 2009). By repeatedly pairing transcranial magnetic stimulation of the primary motor cortex (M1) and motoneuronal electrical stimulation timed to arrive at the corticomotoneuronal (CM) synapses in close temporal proximity, spike-timing-dependent (Caporale & Dan, 2008; Markram et al., 1997) bidirectional changes in CM transmission can be induced”* (page 1, line 31).

7. Statement beginning on line 21 is unclear. What do the authors mean by 'malleability' of neural circuits? Changes in transmission shown by prior PCMS studies might suggest that the CM synapse is malleable, no?

Authors' response: Thank you for this input. We agree with the reviewer and have revised this statement accordingly.

8. There is a lack of precision with regard to the hypothesis. What mechanisms of motor learning are primed by PCMS? The final common pathway is clearly integral to motor control, but why is it particularly important for learning?

Authors' response: With the newly added *Introduction*, more precision has been added to clarify and substantiate the hypothesis:

“We hypothesized that PCMS could prime mechanisms of subsequent motor learning exclusively when directed to the neural circuitry underpinning the motor behavior. Ballistic performance requires a high voluntary output and thereby a high firing rate in as many motor units as possible. Increased efficiency of CM-synapses may lead to a more effective input to the motor neurons, resulting in increased peak acceleration. Indeed, changes in neural transmission at the spinal cord level may particularly contribute to improvements during ballistic motor learning(Giesebrecht et al., 2012). In Experiment I, we used a PCMS protocol timed to induce plasticity of the CM-synapse with the aim of enhancing subsequent ballistic motor learning. In Experiment II, we expanded the relevance by replicating the findings from Experiment I in a double-blinded, sham-controlled experiment and investigated the behavioral effects of PCMS on ballistic motor learning in a retention test seven days after motor practice. Lastly, in Experiment III, we investigated whether the effects of PCMS on ballistic learning were bidirectional and circuit-specific, and whether only paired stimulations timed to facilitate corticomotoneuronal transmission increased learning compared to paired stimulations timed to depress corticomotoneuronal synaptic transmission.” (page 2, lines 42-58)):

9. It is unclear why/how rapid finger movements rely on spinal 'neuroplasticity' as stated on lines 28-29. Similarly, it is unclear how acquisition of ballistic index finger movements are contingent on spinal 'plasticity' as stated on lines 45-46. Perhaps, spinal mechanisms are the relevant feature?

Authors' response: Thank you for your input. We agree with the reviewer and this particular sentence has been deleted in the new version of the *Introduction*. The expanded *Introduction* elaborates more on the relationship between PCMS and Ballistic index finger movements, and their connection to spinal mechanisms (*corticomotoneuronal synapses*):

“...Ballistic performance requires a high voluntary output and thereby a high firing rate in as many motor units as possible. Increased efficiency of CM-synapses may lead to a more effective input to the motor neurons, resulting in increased peak acceleration. Indeed, changes in neural transmission at the spinal cord level may particularly contribute to improvements during ballistic motor learning(Giesebrecht et al., 2012)”. (page 2, lines 44-49)

10. How does one-week retention in neurologically-normal young adults expand 'clinical relevance' as stated on lines 48-51?

Authors' response: We agree with the Reviewer. We have deleted the word 'clinical', and now focus more on the value of replication, and the seven-day retention test:

“In Experiment II, we expanded the relevance by replicating the findings from Experiment I in a double-blinded, sham-controlled experiment and investigated the behavioral effects of PCMS on ballistic motor learning in a retention test seven days after motor practice.” (page 2, lines 53-56).

11. It would appear that the vertical dashed lines in Figures 1B, 2B, and 3B correspond to practice blocks, but it is unclear why the ticks do not align.

Authors' response: The vertical dashed lines indicate the breaks between blocks, and are therefore not aligned with the ticks. To avoid confusion, we deleted the vertical dashed lines, and inserted space, to indicate the break between practice blocks.

12. Were the accelerations produced into index finger abduction or flexion? Index finger flexion is specified throughout the manuscript, but this does not appear consistent with the prior work cited and also seems inconsistent with the depiction shown in Figure 1A.

Authors' response: Thank you for bringing this to our attention. We have now clarified that accelerations produced index finger flexions, also added an extra label "index finger flexion" to Figure 1a.

We now provide more detail on the choice of index finger flexion rather than abduction. In general, we chose flexion over abduction since pilot experiments demonstrated that participants perceived less fatigue and performance improvement for ballistic flexions followed expected learning curves. Indeed, FDI is not the only muscle involved in index finger flexions. The long flexor muscles also contribute to the movement, but isolated abduction is also likely to involve cocontraction of other muscles. The choice of index finger flexion as a behavioral model has also been addressed *in the Discussion*:

"FDI is not the only muscle involved in index finger flexions, but we chose FDI because the ulnar nerve primarily supplies the motor innervation of the index finger. Thereby we can be fairly certain about the mechanistic route compared to what is previously seen with lower limb muscles, i.e., m. tibialis anterior (Urbin et al., 2017), namely that the timing of α -motoneuron activity is corrupted by early arrival of Ia afferent inputs or similar". (page 10, lines 240-245)

13. Please enter units for what appear to be normalized MEP values reported on line 90.

Authors' response: Thank you, this has now been added.

14. Does increasing peak acceleration of a single finger movement reflect a change in 'dexterous motor functions' as stated on lines 99-102?

Authors' response: The intention of this statement was to refer to previous findings, and the reference has now been added (Bunday & Perez, 2012b). We do not think peak acceleration of a single finger movement reflect a change in dexterous motor function. To avoid confusion, we decided to delete dexterous, and now only write "motor functions". Please also note that the paragraph with the sentence is now moved to the discussion (page 9, line 227).

15. It is unclear what 'this' is referring to in the statement beginning on line 190. Results reported in this paragraph are a bit unclear.

Authors' response: Thank you, we have now revised this paragraph.

16. Reporting on changes in normalized MEPs in Experiment III is also unclear. It seems that there was a baseline difference between conditions, but the coupled-control condition also showed a facilitation? Did the M-max remain constant? Reporting in the supplementary indicates that only 6/18 subjects showed decreased corticospinal excitability following the PCMS- protocol, but a significant within-group increase is reported. The traces in Figure 3C do not appear representative.

Authors' response: As mentioned in the manuscript, the three protocols in Experiment III were not entirely matched at baseline. Despite this baseline difference, PCMS+ induced relatively larger increases in MEP from baseline to post stimulation compared to PCMS-, but not compared to PCMS_{coupled-control}. Indeed, the facilitatory effect of PCMS_{coupled-control} on corticospinal excitability was unexpected (as we discuss on page 12). We have now added information about M-max (see also Supplementary Fig. 3), which did not change the paragraph. Regarding the reporting in the *Supplementary Information*, we have now added further information to the legend of Figure 3C, to make the Figure more understandable. We have also included reporting of all data on M and F-wave amplitude and characteristics in the revised version of Supplementary Notes to provide these data in full to the readers. We added the following text in the **Result section**:

“Analyses of M_{max} and F-wave amplitudes demonstrated no significant main effects (see Supplementary Fig. 3.), suggesting that normalization to M_{max} likely cannot explain the observed group differences in corticospinal excitability.” (page 5, lines 107-109)

17. Note that a previous study by Taylor's group showed high-intensity, low-frequency TMS increased corticospinal excitability (D'Amico et al., 2020, J Neurophys).

Authors' response: Thank you for bringing this to our attention. The mentioned paper has now been added to the **Discussion**:

“Another alternative is that the observed after-effects were not caused by the coupling per se, but rather by individual effects of PNS or TMS alone. To investigate this possibility, we conducted a set of control experiments as part of the present study. These control experiments showed that neither unpaired high intensity PNS or unpaired high intensity TMS at a frequency of 0.1 Hz resulted in changes in MEP amplitudes (see Supplementary Fig. 4). These findings are consistent with results from previous studies (Fitzpatrick et al., 2016; Shulga et al., 2016). However, It has previously been reported that high intensity rTMS at 0.1 Hz can lead to increased MEP amplitudes in the adductor pollicis muscle, with effects lasting up to 35 min(D'Amico et

al., 2020). *Differences in stimulation parameters and the specific muscle targeted could potentially account for these conflicting findings. Nevertheless, our control experiment demonstrated that the stimulation parameters employed for TMS and PNS in the present study did not increase MEP amplitudes when administered alone. This demonstrates that the observed changes in corticospinal excitability were specifically mediated by the coupled stimulations, rather than their individual effects.*" (page 13, lines 340-354).

References used in Authors' responses:

- Aagaard, P., E.B. Simonsen, J.L. Andersen, P. Magnusson & P. Dyhre-Poulsen (2002): Increased rate of force development and neural drive of human skeletal muscle following resistance training. *Journal of Applied Physiology*, Vol. 93:4, pp. 1318–1326.
- Bunday, K.L. & M.A. Perez (2012a): Motor recovery after spinal cord injury enhanced by strengthening corticospinal synaptic transmission. *Current Biology*, Elsevier Ltd, Vol. 22:24, pp. 2355–2361.
- Bunday, K.L. & M.A. Perez (2012b): Motor recovery after spinal cord injury enhanced by strengthening corticospinal synaptic transmission. *Current Biology*, Vol. 22:24, pp. 2355–2361.
- Caporale, N. & Y. Dan (2008): Spike Timing–Dependent Plasticity: A Hebbian Learning Rule. *Annual Review of Neuroscience*, Vol. 31:1, pp. 25–46.
- D'Amico, J.M., S.C. Dongés & J.L. Taylor (2020): High-intensity, low-frequency repetitive transcranial magnetic stimulation enhances excitability of the human corticospinal pathway. *Journal of Neurophysiology*.
- Fitzpatrick, S.C., B.L. Luu, J.E. Butler & J.L. Taylor (2016): More conditioning stimuli enhance synaptic plasticity in the human spinal cord. *Clinical Neurophysiology*, International Federation of Clinical Neurophysiology, Vol. 127:1, pp. 724–731.
- Giesebrecht, S., H. Van Duinen, G. Todd, S.C. Gandevia & J.L. Taylor (2012): Training in a ballistic task but not a visuomotor task increases responses to stimulation of human corticospinal axons. *Journal of Neurophysiology*, Vol. 107:, pp. 2485–2492.
- Kantak, S.S. & C.J. Winstein (2012): Learning-performance distinction and memory processes for motor skills: A focused review and perspective. *Behavioural Brain Research*, Elsevier B.V., Vol. 228:1, pp. 219–231.
- Karabanov, A., U. Ziemann, M. Hamada, M.S. George, A. Quartarone et al. (2015): Brain Stimulation Consensus Paper: Probing Homeostatic Plasticity of Human Cortex With Non-invasive Transcranial Brain Stimulation. *Brain Stimulation*, Elsevier, Vol. 8:5, pp. 993–1006.
- Krakauer, J.W., A.M. Hadjiosif, J. Xu, A.L. Wang & A.M. Haith (2019): Motor Learning. *Comprehensive Physiology*, Vol. 9:April, pp. 613–663.
- Markram, H., J. Lu, M. Frotscher & B. Sakmann (1997): Regulation of Synaptic

Efficacy by Coincidence of Postsynaptic APs and EPSPs. *Science*, Vol. 275:January, pp. 213–216.

Rogasch, N.C., T.J. Dartnall, J. Cirillo, M.A. Nordstrom & J.G. Semmler (2009): Corticomotor plasticity and learning of a ballistic thumb training task are diminished in older adults. *Journal of Applied Physiology*, Vol. 107:6, pp. 1874–1883.

Rosenkranz, K., A. Kacar & J.C. Rothwell (2007): Differential modulation of motor cortical plasticity and excitability in early and late phases of human motor learning. *Journal of Neuroscience*, Soc Neuroscience, Vol. 27:44, pp. 12058–12066.

Shmuelof, L. & J.W. Krakauer (2011): Are we ready for a natural history of motor learning? *Neuron*, Elsevier Inc., Vol. 72:3, pp. 469–476.

Shulga, A., A. Zubareva, P. Lioumis & J.P. Mäkelä (2016): Paired Associative Stimulation with High-Frequency Peripheral Component Leads to Enhancement of Corticospinal Transmission at Wide Range of Interstimulus Intervals. *Frontiers in Human Neuroscience*, Vol. 10:September, pp. 1–6.

Taylor, J.L. & P.G. Martin (2009): Voluntary motor output is altered by spike-timing-dependent changes in the human corticospinal pathway. *Journal of Neuroscience*, Soc Neuroscience, Vol. 29:37, pp. 11708–11716.

Urbin, M.A., R.A. Ozdemir, T. Tazoe & M.A. Perez (2017): Spike-timing-dependent plasticity in lower-limb motoneurons after human spinal cord injury. *Journal of Neurophysiology*, Vol. 118:4, pp. 2171–2180.

Wolpaw, J.R. (2018): The negotiated equilibrium model of spinal cord function. *Journal of Physiology*, Vol. 16:, pp. 3469–3491.

Reviewers' comments:

Reviewer #1 (Remarks to the Author):

The manuscript has been improved by revision. In particular, analysis of FDI EMG supports the interpretation that differences in FDI activity contribute to the differences in motor performance after the PCMS intervention. Additional comments have arisen from the revisions.

Specific comments follow

Line 39. "Interacts" should be "interact"

Line 42-44. The hypothesis is not clearly stated. It is not clear what "exclusively" refers to.

Lines 160-169. Please add n (sample size) for these analyses.

Line 179. Delete "effects"

Figure 3 legend. As this is a cross over design there are no groups to have between group comparisons. In 3f, it is not clear what the asterisks mean, i.e. are these comparisons to baseline or to both other time points? More specifically, what difference does the asterisk for Post practice PCMS- represent?

Line 218. "learning-induced neuroplasticity" - perhaps "experience-dependent" would be better here. It is difficult to conceptualise learning as causing neuroplasticity when they seem equivalent processes.

Line 221. The study did not provide evidence of whether effects of PCMS on motor learning persisted for seven days but rather showed that the better performance that ensued from the effect of PCMS on motor learning persisted.

Line 239. "the PCMS prime" should be "priming by PCMS".

Line 242-246. I do not follow the argument here about choice of muscle and the influence of Ia afferents. It is unclear where the afferents are thought to arrive to alter motoneuron activity. If it is at the motoneurons then there seems no reason that Ia afferent volleys from ulnar nerve stimulation would not arrive around the same time as antidromic volleys in the motor axons, or why this should be different in the legs.

Line 280 "carry over effects at the seven-day..." to "carry over effects as seen at the seven-day"

Lines 473-479. How many stimuli were delivered and at which time points? This stimulation is described as if it were a set up procedure but Mmax and F waves values are now reported for baseline, post stimulation and post practice.

Supplementary section on Mmax amplitudes etc

No degrees of freedom are given for F values in this section

Reviewer #2 (Remarks to the Author):

The authors have made notable attempts to address comments on the original submission, but some issues remain. In general, there is still a need to improve clarity. Lines 32-36, for example, still imply direct electrical stimulation of spinal motor neurons. Similarly, “efficiency” and “efficacy” of synapses are used interchangeably. These issues are relatively minor but should be resolved. In addition, it is understood that enhanced transmission at the spinal level is helping to facilitate learning, but the title still seems to suggest that learning is taking place in the spinal cord. The word “spinal” is used as a modifier of (motor) learning, but this seems misleading given that spinal and supraspinal circuitries are strongly integrated and that learning any voluntary movement is in some way reliant on the latter. These considerations aside, more substantive issues related to reporting and methodology are grouped by experiment as follows.

Experiment I:

Language on lines 79-81 indicates that PCMS led to significantly larger improvements from the first to the last practice block, but statistical analyses appear to have shown that group differences existed at block 2 and block 3. More direct reporting is needed here.

Relatedly, lines 81-84 specify that there was no group difference at baseline, but there are no values reported nor any depiction of the data (e.g., Fig 1B). It is understood that accelerations were normalized to baseline, but there should be some reporting to help the reader gauge the comparison.

Experiment II:

The authors state that Experiment II was double blinded. How was this possible given that the experimenter controlled coil orientation and stim amplitude for peripheral nerve stimulation?

Similar to reporting in Experiment 1, lines 128-131 indicate that PCMS led to significantly larger improvements from the first to the last practice block, but a group difference was noted at block 3. Please be more direct.

Related to EMG analyses, why did the authors use baseline for comparison rather than block 1 as was done for acceleration and corticospinal excitability? It seems that this complicates interpretation of correlations in the relative change between acceleration and EMG.

How was the relative change in acceleration and EMG computed?

It seems odd that both EMG RMS and EMG rise increased to a greater extent over baseline in the PCMS group relative to the Sham group, but this change was not retained on Day 7, yet the gains in acceleration were retained.

How were data entered into the model given that only the Sham group had Day 7 retention but data from both Rest (Experiment I) and Sham (Experiment II) were pooled in the analysis?

Experiment III:

Part of the confusion surrounding comment 2 stems from the statement on lines 192-193 in the original manuscript which stated, "Additionally, we found PCMS- to slow motor learning evident as a lower performance in Block 1 compared to PCMSCoupled-control." This statement is no longer present in the resubmission, but some concerns remain regarding the interpretation. Part of the issue is that it seems as though there really is not an effective comparator since both PCMS+ and PCMSCoupled-control are trending above baseline. A true sham might have offered a better comparison. It is also problematic that corticospinal excitability is increased following all three protocols (Fig 3F). How do the authors reconcile that excitability along the pathway mediating behavior is increased yet motor output is suppressed? The explanation that cortical changes may be masking spinal changes seems to conflict with where along the neuraxis the effects of PCMS are thought to be mediated. PCMS has not been shown to have an effect on cortical circuits. Finally, acceleration traces from the single subject provided in the rebuttal do not appear to support a suppressive effect. All of these issues call into question whether there are bidirectional effects regardless of the small difference between conditions in the first block of practice.

Revision-Round-2: Point-by-point Responses to Referees comments

We would like to thank both reviewers for their constructive feedback and suggestions. We have tried to accommodate most of them in the revised version of our manuscript. Specifically, we have made several changes to the manuscript, including: The addition of a new control experiment (Experiment 4), and new analyses to address the inputs from the reviewers.

We believe that this has improved the manuscript substantially. Please find the revisions made in a point-by-point manner to your comments in what follows below.

Color coding:

- Responses to referees are color-coded **blue**
- Text included in 'Manuscript' and 'Supplementary Information' are color-coded **red**, and **green** text are specific changes made in Revision-Round-2.

Reviewer #1 (Remarks to the Author):

The manuscript has been improved by revision. In particular, analysis of FDI EMG supports the interpretation that differences in FDI activity contribute to the differences in motor performance after the PCMS intervention. Additional comments have arisen from the revisions.

Authors' response: We thank the reviewer for the positive feedback. Below you will find our responses to the additional comments.

Specific comments follow

Line 39. "Interacts" should be "interact"

Authors' response: Thank you, we have now corrected this.

Line 42-44. The hypothesis is not clearly stated. It is not clear what "exclusively" refers to.

Authors' response: We agree with the reviewer, 'exclusively' was not intuitive, and we have now removed 'exclusively'.

The intention of the use of '**Exclusively**' refers to the **PCMS+ protocol**. Since PCMS+ was the only protocol, where the TMS volley and PNS volley was timed to enhance the neural transmission at the CM-synapses, the neural circuitry underpinning the ballistic

performance. This was not the case in any of the control protocols (Rest in Experiment 1, Sham in Experiment 2, or PCMS- or PCMS_{coupled-control} in Experiment 3).

Lines 160-169. Please add n (sample size) for these analyses.

Authors' response: Thank you for noticing. We have now added sample size to the paragraph: N=46 participants.

Line 179. Delete “effects”

Authors' response: Thank you, “Effects” has been deleted.

Figure 3 legend. As this is a cross over design there are no groups to have between group comparisons. In 3f, it is not clear what the asterisks mean, i.e. are these comparisons to baseline or to both other time points? More specifically, what difference does the asterisk for Post practice PCMS- represent?

Authors' response: Thank you for noticing this. We have now changed the wording to between-protocol and within-protocol comparisons. As mentioned in the Figure 3 caption: The asterisks (*) indicates significant within-protocol comparisons $p < 0.05$ between time-points. For example, the asterisks in Figure 3F for Post practice PCMS+ represents significantly higher MEP amplitudes post practice compared to PCMS+ at baseline.

Line 218. “learning-induced neuroplasticity” – perhaps “experience-dependent” would be better here. It is difficult to conceptualise learning as causing neuroplasticity when they seem equivalent processes.

Authors' response: We agree with the reviewer, and we have updated the term accordingly. Also at lines 66-67.

Line 221. The study did not provide evidence of whether effects of PCMS on motor learning persisted for seven days but rather showed that the better performance that ensued from the effect of PCMS on motor learning persisted.

Authors' response: We agree and have now clarified the statement accordingly: *“The improved performance that ensued from the effect of PCMS on ballistic motor learning persisted seven days after motor practice (Experiment II)”*. (Lines 307-309)

Line 239. “the PCMS prime” should be “priming by PCMS”.

Authors' response: Thank you, this has been corrected.

Line 242-246. I do not follow the argument here about choice of muscle and the influence of Ia afferents. It is unclear where the afferents are thought to arrive to alter motoneuron activity. If it is at the motoneurons then there seems no reason that Ia afferent volleys from ulnar nerve stimulation would not arrive around the same time as antidromic volleys in the motor axons, or why this should be different in the legs.

Authors' response: Thank you for your comment. We now elaborate more on our argument to clarify for the reviewer:

FDI is not the only muscle involved in index finger flexions, but we chose FDI because the ulnar nerve primarily supplies the motor innervation of the index finger. The deep afferents from the index finger and thumb run in the median nerve, whereas the α -motoneurons innervating FDI run in the ulnar nerve (Amoiridis, 1992; Meals & Shaner, 1983). We therefore argue that FDI is a good target when we aim to target corticomotoneuronal transmission through antidromic invasion of the soma and dendrites specifically. Notwithstanding, we fully acknowledge that the supramaximal stimulation intensity likely causes current to spread to the median nerve, so a contribution from afferent activation of α -motoneurons cannot be excluded. On the other hand, we used a relative short pulse duration for the ulnar, which lowers activation threshold for nodal excitation of the motoneurons as compared to sensory afferents (Bostock & Rothwell, 1997; Panizza et al., 1994). Thereby by choosing FDI as our target muscle, we have a more controlled model compared to what is previously seen with lower limb muscles, i.e., m. tibialis anterior (Urbin et al., 2017), namely that the timing of α -motoneuron activity is corrupted by early arrival of Ia afferent inputs or similar.

Note, that after careful considerations, we have chosen not to include above paragraph in the manuscript. This choice was primarily made to prevent a wide-ranging diversion from the main focus in the discussion.

Line 280 “carry over effects at the seven-day...” to “carry over effects as seen at the seven-day”

Authors' response: Thank you, this has been corrected accordingly.

Lines 473-479. How many stimuli were delivered and at which time points? This stimulation is described as if it were a set up procedure but M_{max} and F waves values are now reported for baseline, post stimulation and post practice.

Authors' response: Thank you for bringing this to our attention. We apologize for not providing this information. It has now been added to the paragraph:

“Electrical stimulation with high voltage electrical current (200 μ s pulse duration, DS7A; Digitimer) was delivered to the ulnar nerve at the wrist (Bar Stimulating Electrode, Digitimer) to measure the maximal compound muscle action potential (M_{max}) and F-waves. The latencies of M_{max} and F-waves were used to calculate the peripheral conduction time used to individualize the paired stimulation protocols (Supplementary Fig. 1b-d). F-wave latency was defined as the F-wave with the earliest onset (Christiansen et al., 2018). At baseline, post stimulation and post practice M_{max} amplitudes and F-wave amplitudes were measured to indirectly quantify α -motoneuron excitability (Mcneil et al., 2013). We delivered 60 stimulations at supramaximal (130% of M_{max}) intensity delivered at 1 Hz (Lin & Floeter, 2004).” (Lines 611-620).

Supplementary section on Mmax amplitudes etc

No degrees of freedom are given for F values in this section

Authors' response: Thank you for bringing this to our attention, we have now added degrees of freedom for F values:

“...For F-wave persistence, we found a main effect of TIME in all three experiments (Experiment I, $F_{(2,28)}=3.83$, $p=0.03$; Experiment II, $F_{(2,39)}=23.9$, $p<0.001$; Experiment III, $F_{(2,144)}=19.9$, $p<0.001$).” (Supplementary)

Reviewer #2 (Remarks to the Author):

The authors have made notable attempts to address comments on the original submission, but some issues remain. In general, there is still a need to improve clarity. Lines 32-36, for example, still imply direct electrical stimulation of spinal motor neurons. Similarly, “efficiency” and “efficacy” of synapses are used interchangeably. These issues are relatively minor but should be resolved. In addition, it is understood that enhanced transmission at the spinal level is helping to facilitate learning, but the title still seems to suggest that learning is taking place in the spinal cord. The word “spinal” is used as a modifier of (motor) learning, but this seems misleading given that spinal and supraspinal circuitries are strongly integrated and that learning any voluntary movement is in some way reliant on the latter. These considerations aside, more substantive issues related to reporting and methodology are grouped by experiment as follows.

Authors' response: Thank you for the constructive feedback and the acknowledgment of our revisions. Below you will find our point-by-point response to your comments and suggestions. We wish to highlight that we have included a new set of relevant control experiments (40 additional experiments) in the revised version of the manuscript in response to some of the concerns raised by the reviewer. These are now presented as Experiment IV in the revised manuscript to accommodate relevant critical points and to strengthen the manuscript. Please find more details in what follows.

Line 32-36 has been clarified:

“By repeatedly pairing transcranial magnetic stimulation of the primary motor cortex (M1) and electrical stimulation of peripheral nerves timed to arrive at the corticomotoneuronal (CM) synapses in close temporal proximity, spike-timing-dependent, bidirectional changes in CM transmission can be induced” (Lines 35-39)

The two instances where “efficiency” and “efficacy” has been streamlined:

“...Increased efficiency of CM-synapses may lead to a more effective input to the motor neurons, resulting in increased peak acceleration.” (Lines 48-49)

“Consequently, the observed increase in peak acceleration in the present study can be assumed to largely reflect improved efficiency of direct activation of spinal motoneurons.” (Lines 325-327)

Regarding, “...it is understood that enhanced transmission at the spinal level is helping to facilitate learning, but the title still seems to suggest that learning is taking place in the spinal cord. The word “spinal” is used as a modifier of (motor) learning, but this seems misleading given that spinal and supraspinal circuitries are strongly integrated and that learning any voluntary movement is in some way reliant on the latter”

We acknowledge the reviewers comment, and we of course want to avoid potential misunderstandings regarding the title of the manuscript. We have therefore decided to change the title to: “**Hebbian priming of human motor learning**”.

Experiment I: Language on lines 79-81 indicates that PCMS led to significantly larger improvements from the first to the last practice block, but statistical analyses appear to have shown that group differences existed at block 2 and block 3. More direct reporting is needed here.

Authors’ response: Thank you for bringing this to our attention. We have now clarified the sentences to avoid confusion.

Compared to Rest, the PCMS group performed better at the second practice block (B2) (PCMS+: $39.8\% \pm 1.8$ vs. Rest $15.7\% \pm 1.8$, $p=0.01$) and the third practice block (B3) (PCMS+: $48.2\% \pm 1.8$ vs. Rest $20.5\% \pm 1.8$, $p<0.01$) (Figure 1B). Importantly, there was no difference at the beginning of practice (B1), suggesting that the effects of PCMS gradually interacted with motor practice. This was corroborated by comparisons of changes in performance from the beginning of practice (B1) to the end of practice (B3) (PCMS+: $22.28\% \pm 1.0$ vs. Rest: $10.93\% \pm 1.0$, $p<0.001$, Figure 1c). (Lines 86-94).

Relatedly, lines 81-84 specify that there was no group difference at baseline, but there are no values reported nor any depiction of the data (e.g., Fig 1B). It is understood that accelerations were normalized to baseline, but there should be some reporting to help the reader gauge the comparison.

Authors’ response: We have now added information of raw baseline peak acceleration values (g). This has also been done for Experiment II (Fig 2B) and Experiment III (Fig 3B) i.e. consistently for all experiments. Thank you for your constructive input.

“...Performance did not differ between groups at baseline in raw values (PCMS+: $M=1.9$, $SD=0.44$ vs. Rest: $M=2.14$ g, $SD=0.46$ $p=0.727$), suggesting that the differences that emerged during practice were not due to general differences in task proficiency but likely could be ascribed to PCMS+ interacting with the effects of motor practice “(Lines 93-96)

Because of the above, we have also added the following to the ‘Statistical Analysis’ section: “Mean (M) and standard deviation (SD) are shown when raw data is presented.” (line 682)

Experiment II: The authors state that Experiment II was double blinded. How was this possible given that the experimenter controlled coil orientation and stim amplitude for peripheral nerve stimulation?

Authors' response: This is a very important point, and we agree that this should have been described in greater details. We thank the reviewer for giving us the opportunity to elaborate. The section below has been added to the 'Experimental design' section, and provides more detail to how Experiment II was double-blinded.

In Experiment II, Investigator A was aware of allocation (whether PCMS or Sham were administered) while Investigator B was blinded to the allocation. During the intervention, Investigator B physically left the laboratory during the PCMS/Sham protocol. Investigator B returned to the laboratory after the PCMS/sham intervention and oversaw the motor practice session. Importantly, before commencing motor practice, Investigator A (who was aware of the group allocation) left the laboratory. Effectively, this means that Investigator A could not influence how participants performed in the motor practice session. Group allocation was not revealed until all data points were collected and analyzed. This also means that Investigator B performed the data analysis blinded. With this setup, Investigator A, who managed the TMS coil (position and orientation) during PCMS or Sham was unable to affect motor performance and the subsequent data analysis. (Lines 537-546)

Similar to reporting in Experiment 1, lines 128-131 indicate that PCMS led to significantly larger improvements from the first to the last practice block, but a group difference was noted at block 3. Please be more direct.

Authors' response: Thank you for bringing this to our attention. We have now clarified the sentence to avoid confusion.

Compared to Sham, the PCMS group performed better at the third practice block (B3) (PCMS+: $28.6\% \pm 1.4$ vs. Sham $19.3\% \pm 1.4$, $p < 0.01$) (Figure 2B). There was no difference at practice block 1 (B1) nor from practice block 2 (B2). Similar to Experiment I, a difference was observed in comparisons of changes in performance from the beginning of practice (B1) to the end of practice (B3) (PCMS+: $17.4\% \pm 0.8$ vs. Sham: $11.8\% \pm 0.8$, $p < 0.001$, Figure 2c). Lines 141-146).

Related to EMG analyses, why did the authors use baseline for comparison rather than block 1 as was done for acceleration and corticospinal excitability? It seems that this complicates interpretation of correlations in the relative change between acceleration and EMG.

Authors' response:

We have been consistent in that that all variables were normalized to their respective baseline values. In that way, we can correlate the relative change from baseline to end of motor practice between acceleration and EMG measures. We agree with the reviewer that EMG (%baseline) contrasts relative to Block 1 are in line with our analysis on acceleration data, this has now been changed accordingly:

“Within TIME, comparisons showed a significant increase from practice Block 1 to practice Block 2 ($2.7\% \pm 0.9$, $p = 0.03$) and from practice Block 1 to practice Block 3 ($4.1\% \pm 0.9$, $p < 0.01$). Pairwise comparisons showed that PCMS led to a significantly larger increase in EMG_{RMS} amplitude from practice Block 1 to practice Block 2 (PCMS: $4.2\% \pm 1.3$ vs. Control: $1.1\% \pm 1.4$, $p = 0.048$). (Supplementary)”

While acceleration and EMG measures were obtained in parallel, this was not the case for measures of corticospinal excitability, which were obtained in separate measurements. Note that for corticospinal excitability (MEP's) we chose baseline comparison (since only motor performance was measured at Block 1-3).

How was the relative change in acceleration and EMG computed?

Authors' response: We now present this more clearly.

All of our variables were presented relative to their respective baseline values. The correlation was made between EMG (%baseline) and acceleration (%baseline) at end of practice.

Pearson correlation tests showed a non-significant correlation between peak acceleration (%baseline) and EMG_{RMS} amplitude (%baseline) ($r=0.21$, $p=0.16$) and a significant positive correlation between the rate of EMG rise (%baseline) ($r=0.42$, $p<0.01$) and acceleration (%baseline) (Supplementary Fig. 2e).(Supplementary)

It seems odd that both EMG RMS and EMG rise increased to a greater extent over baseline in the PCMS group relative to the Sham group, but this change was not retained on Day 7, yet the gains in acceleration were retained.

Authors' response: We agree with the Reviewer that it could be expected that the difference in EMG change would also be present at day 7 given that the gains in acceleration were retained. However, we would also like to highlight that the day 7 results only include data from Experiment II (N=20). Furthermore, it is important to note that day 7 results stem from another EMG recording. Although we were pertinent in ensuring that EMG electrodes were placed in a comparable manner at both sessions, and used golden standard approaches to normalize our EMG measures within-session to the M_{max} to account for potential recording conditions, we cannot rule out that this may have affected our results.

How were data entered into the model given that only the Sham group had Day 7 retention but data from both Rest (Experiment I) and Sham (Experiment II) were pooled in the analysis?

Authors' response: We thank the reviewer for this important point. We have now chosen to separate the analysis into two models: Model 1 (within day 1 changes with pooled data) and Model 2 (only experiment II, including day 7 data). In that way Model 2 only takes into account the comparison between PCMS and Sham from experiment II. This was done due to unequal data between day 1 and day 7 after the pooling. This are now described in more detail in the EMG analysis in the supplementary, and the model estimates and p-values have been corrected based on the results from the two new models.

Experiment III: Part of the confusion surrounding comment 2 stems from the statement on lines 192-193 in the original manuscript which stated, "Additionally, we found PCMS- to slow motor learning evident as a lower performance in Block 1 compared to PCMSCoupled-

control.” This statement is no longer present in the resubmission, but some concerns remain regarding the interpretation. Part of the issue is that it seems as though there really is not an effective comparator since both PCMS+ and PCMS_{coupled-control} are trending above baseline. A true sham might have offered a better comparison. It is also problematic that corticospinal excitability is increased following all three protocols (Fig 3F). How do the authors reconcile that excitability along the pathway mediating behavior is increased yet motor output is suppressed? The explanation that cortical changes may be masking spinal changes seems to conflict with where along the neuraxis the effects of PCMS are thought to be mediated. PCMS has not been shown to have an effect on cortical circuits. Finally, acceleration traces from the single subject provided in the rebuttal do not appear to support a suppressive effect. All of these issues call into question whether there are bidirectional effects regardless of the small difference between conditions in the first block of practice.

Authors’ response: Thank you for your thorough inputs to Experiment III. In what follows, we will address your comments one by one.

Regarding “Part of the confusion surrounding comment 2 stems from the statement on lines 192-193 in the original manuscript which stated, “Additionally, we found PCMS- to slow motor learning evident as a lower performance in Block 1 compared to PCMS_{coupled-control}.” This statement is no longer present in the resubmission, but some concerns remain regarding the interpretation.”

As mentioned by the Reviewer, we have now modified our interpretation of bidirectional changes in learning in the first revision of the manuscript. We agree with the Reviewer that more evidence is needed to further solidify the finding on bidirectionality. Moreover, we now avoid the term ‘bidirectionality’ in the Abstract, and at the end of the Introduction, and in the beginning of the Discussion. Nevertheless, we still argue based on our results, that there are bidirectional between-group differences based on our findings. Please see below.

Regarding: “...Part of the issue is that it seems as though there really is not an effective comparator since both PCMS+ and PCMS_{coupled-control} are trending above baseline.”

We wish to clarify that the observed upward trend in all groups at practice block 1, block 2 and block 3 compared to baseline motor performance is indeed an expected outcome, given the nature of motor training. Since all groups of participants practiced a motor task, improvements are expected – also within the first block of practice. We argue that the interesting part here is the comparison between protocols across the practice blocks, namely that PCMS+ leads to superior motor performance at the end of motor practice (B3) compared to both PCMS- and PCMS_{coupled-control} and the lower performance in PCMS- compared to PCMS+ and PCMS_{coupled-control} in the first block of practice (B1). The former result exactly underlines the conclusion that stimulation protocols prime learning effects.

Regarding: “A True Sham might have offered a better comparison”.

First and foremost, we wish to clarify that our study does indeed incorporate a double-blinded **sham** control protocol in Experiment 2. In Experiment 2, we asked participants after completion whether they believed, they received real or sham stimulation. No clear trend in reporting was observed. This validates that this protocol did indeed act as a proper control. We therefore argue that the comparison of PCMS+ and a traditional sham had already been addressed in Experiment 2 with an appropriate design and that it therefore does not add additional scientific value to also include the same sham control in Experiment 3, especially given the aims of this experiment.

Experiment 3 was designed to compare the effects observed in Experiments 1 and 2 with high-end active comparative interventions with different proposed physiological effects at the CM-level. Hence, we compared these effects with a protocol previously shown to reduce effective corticomotoneuronal transmission (PCMS-) and a control protocol that applied the paired stimulations outside the window of spinal Hebbian interactions (PCMS_{coupled-control}). We therefore argue, that the latter protocol provides a thorough and appropriate control. Specifically, this allowed us to expand the findings from Experiment 1 and 2 by investigating whether the PCMS effects were contingent on temporal proximity of stimulation and governed by order of spike-timing. We hypothesized that PCMS+ would improve learning as compared to PCMS_{coupled-control} and PCMS- and that PCMS- would reduce learning as compared to PCMS_{coupled-control} and PCMS+.

Notably, in Experiment III, we replicate the positive priming effects of PCMS+ by showing greater ballistic motor learning compared to both of these protocols. We further show that PCMS effects indeed conform to the order and interval of spike-timing, evidenced by the lower performance in PCMS- compared to PCMS_{coupled-control} in the first block of practice. However, we agree with the Reviewer that the effects of the PCMS- protocol are not as prominent as the ones observed for the PCMS+ protocol and only short-lived. We acknowledge this point explicitly in the manuscript and discuss possible reasons for this on lines 362-371 in the main body of the manuscript.

“It should be noted that this negative effect on learning of PCMS- was only observed during early learning, which indicates a smaller or less consistent effect of PCMS- on ballistic motor learning compared to PCMS+. The smaller effect of PCMS- may also in part be influenced by effect sizes being diluted by the within-subject design and the ensuing carry-over effects between experimental days. Nevertheless, the results of Experiment III demonstrate that the priming effect of PCMS on motor learning are circuit and timing-specific meaning that in summary, the effects observed conform to Hebbian learning rules in that they are contingent on temporal proximity of stimulation effects and governed by order of spike-timing.” (Lines 362-371)

To further address the Reviewers concerns we conducted a new set of control experiments (Experiment IV) to better understand the effects on corticospinal excitability (CSE) after all three PCMS protocols. We found this to be a valuable addition to the manuscript. We have therefore added the new control experiment (together with the unpaired rTMS/rPNS control

experiment previously presented in the supplementary results) to the main manuscript as Figure 4 (the Figure is also presented below).

Experiment IV now includes the ten subjects participated in two sessions with either unpaired rTMS or rPNS (previously presented in the supplementary material). The results of this experiment are presented in Figure 4a and demonstrate that unpaired stimulations do not lead to changes in CSE. Eight new subjects completed all three PCMS protocol from Experiment III in different sessions separated by a week (Figure 4b), as well as a non PCMS session (rest) and a motor practice session (Figure 4c). We investigate the effect of PCMS+, PCMS- and PCMS_{coupled-control} on corticospinal excitability for an hour without a following motor practice session. We further tested the effects of motor practice alone.

These data highlight the specificity of the timing between stimulations for the evoked effects, since PCMS+ is followed by the biggest increase in CSE. This also supports the PCMS_{coupled-control} as an active control protocol. Figure 4c further shows an increase in corticospinal excitability in the time after motor practice. This practice-induced effect on corticospinal excitability also confound the findings from the post practice block in Experiment III, since ballistic motor practice in itself (as expected) increase corticospinal excitability.

Below text and figure has been added as Figure 4 of the manuscript (lines 251-303).

Experiment IV: Effects of stimulation protocols and motor practice on corticospinal excitability

To investigate whether the observed effects of PCMS on corticospinal excitability observed in Experiment III were contingent on pairing of peripheral and central stimulations, we tested effects of unpaired rTMS and rPNS on corticospinal excitability in 10 naïve subjects. MEP amplitudes were recorded before the intervention and at 0, 15, 30 and 45 min after the intervention. The results depicted in Figure 4a showed that unpaired cortical or peripheral nerve stimulations did not influence corticospinal excitability since neither of the stimulation protocols led to changes in normalized MEP amplitudes, evident from the non-significant interaction effect Group and Time ($F_{(3,1328)}=0.54$, $p=0.65$) (see Figure 4a).

In the second part of Experiment IV, we tested effects of PCMS protocols without motor practice and effects of motor practice vs. rest on corticospinal excitability in eight naïve individuals. Eight participants completed the PCMS protocols used in Experiment III (PCMS+, PCMS- and PCMS_{coupled-control}), a motor practice session (without any PCMS) and a rest session (no PCMS + no motor practice). Session order was randomized and counterbalanced, and sessions were spaced by one week. Repeated measures of corticospinal excitability were obtained before the intervention and at 0, 15, 30, 45 and 60

min after the interventions (PCMS or motor practice). In line with the results presented in Experiment III, all groups displayed an increase in MEP amplitudes (Figure 4b). However, the magnitude of increases in MEP amplitudes was different for the different PCMS protocols (Figure 4b). This was supported statistically by the LMM showing a significant interaction effect of Protocol and Time ($F_{(8,1694)}=2.05$, $p=0.03$). Post-hoc comparisons showed higher MEP across the entire time course after PCMS+ compared to PCMS- ($85\% \pm 10.9$, $p < 0.001$) and PCMS_{coupled-control} ($+49.5 \pm 11.0$, $p < 0.001$), and PCMS- had smaller MEPs compared to PCMS_{coupled-control} (-36.3 ± 11.0 , $p < 0.01$). These results replicate our findings from Experiment III by demonstrating that PCMS can indeed lead to increases corticospinal excitability irrespective of spike-timing intervals, but also show that the effect on corticospinal excitability is greater in magnitude for the PCMS+ protocol (Figure 4b).

The final part of Experiment IV investigated effects of motor practice alone vs. rest on corticospinal excitability. The results demonstrated that ballistic motor practice increased corticospinal excitability compared to rest (Figure 4c). Across the time period after motor practice, the motor practice group had significantly greater MEP amplitudes (%baseline) compared to the rest condition, evident from the significant main effect of Group ($F_{(1,1056)}=64.7$, $p < 0.001$).

Collectively, Experiment IV demonstrates that ballistic motor learning is accompanied by increased corticospinal excitability. Furthermore, it shows that while unpaired stimulations do not lead to changes in corticospinal excitability, paired stimulations as part of PCMS do, but effects are greater in magnitude when the timing of the paired stimulations are set to match principles of Hebbian plasticity.

Figure 4. Experiment IV: Control Experiments investigating effects of unpaired rPNS or rTMS, and the specificity of PCMS effects on corticospinal excitability. a) Within-subject design (N=10), with participants completing both rPNS (purple) or rTMS (light-purple) separated by a week. To test the effects of non-paired low frequency stimulations on corticospinal excitability. b) Within-subject design

(N=8), with participants competing three test days, to test three PCMS stimulations protocols effect on corticospinal excitability (1: PCMS+, blue; 2: PCMS-, purple, and 3: PCMS_{coupled-control}, orange). Note similar PCMS protocols was used as in Experiment III, but now with an expanded time course, and without interfering effects of motor practice. c) The same 8 participants completed two test days, the effects of motor practice without prior PCMS stimulations was compared against a non-practicing group (Rest) that neither received prior PCMS stimulations. All three plots show MEP amplitude in percentage of baseline from each time point. Error bars indicate standard deviation. Note the same scaling on the y-axis on the three plots.

In relation to Experiment IV, the following has been added to the discussion section:

“Experiment IV additionally demonstrated, that all PCMS protocols increased corticospinal excitability compared to rest. Notably, the PCMS+ led to greater increases in corticospinal excitability compared to PCMS- conditioning and the PCMS_{coupled-control}. These results demonstrate again – but now for changes in corticospinal excitability – that facilitatory effects required paired stimulation and were modulated by specific timing aimed at the CM synapse. While motor practice is accompanied by increased corticospinal excitability, a similar but very clear effect can be observed specifically following PCMS+ but not following unpaired stimulation protocols and lower following PCMS at other intervals. This finding is relevant, since it demonstrates the specificity of the effect and substantiates the role of principles governing Hebbian plasticity including effects of pre- and postsynaptic activation and the role of timing of this activation” (Lines 459-470).

Regarding: *” It is also problematic that corticospinal excitability is increased following all three protocols (Fig 3F). How do the authors reconcile that excitability along the pathway mediating behavior is increased yet motor output is suppressed?*

We acknowledge the reviewer’s comment and we aim at discussing these findings thoroughly in the manuscript. We acknowledge that these points may not have been clear in the previous version of manuscript, and we would therefore like to elaborate on these points in what follows:

First, the finding from PCMS- that ballistic motor output is suppressed, and excitability is increased, is not an uncommon phenomenon. In fact, this phenomenon is clearly highlighted in the literature (Bestmann & Krakauer, 2015). Several studies have shown that the relationship between changes in MEPs and motor behavior (with motor training) is not straightforward (Bagce et al., 2012; Gelli et al., 2007; McDonnell & Ridding, 2006). This point is also addressed in the manuscript:

“Importantly, we demonstrate that increasing corticospinal excitability does not improve motor learning per se; only when the targeted plasticity is directed to the neural circuitry that underpins the specific motor behavior – i.e., PCMS+ aimed at the CM level - then subsequent ballistic learning is enhanced. This shows that the relationship between changes in corticospinal excitability and motor performance is not straightforward, which is not an uncommon phenomenon in the literature (Bagce et al., 2012; Gelli et al., 2007; McDonnell & Ridding, 2006)” (lines 400-406).

The physiological complexity of the motor evoked potential (MEP) as a compound potential might be part of the reason for the ambiguous relationships between motor behavior and corticospinal excitability, as discussed in the manuscript and in greater detail below.

Regarding: “The explanation that cortical changes may be masking spinal changes seems to conflict with where along the neuro-axis the effects of PCMS are thought to be mediated. PCMS has not been shown to have an effect on cortical circuits.”

Here, we want to point out the complexity of the interactions between cortical and peripheral nerve stimulations in the applied PCMS protocols. We found that all three protocols led to increase corticospinal excitability. We have clearly acknowledged and discussed potential physiological explanations for these findings in detail throughout the manuscript. Importantly, however, this discussion was indeed not intended to convey the point that PCMS *per se* works through cortical circuits, but rather that the specific interstimulus interval (ISI) between PNS and TMS (i.e., ~ 22.5 ms) that allows an inter-arrival time (IAI) that should reduce transmission at the CM synapses in PCMS protocols **also** leads to coinciding activity at the cortical level via co-stimulation of sensory afferents at latencies and ISIs classically used in studies using paired-associative stimulation (~ 22.5ms). See the illustration below.

We acknowledge that this point may not have been clear. We have therefore modified these sections:

- **PCMS-:** “Furthermore, we did not find a suppressive effect of PCMS- on corticospinal excitability. Indeed, significant facilitation of MEP amplitudes were observed after PCMS-. At first glance, this result is at odds with previous reports, but in contrast to Taylor & Martin (Taylor & Martin, 2009) we evaluated corticospinal excitability with TMS evoked MEPs and not CMEPs, a measure of subcortical excitability. Interestingly, when targeting FDI, the average interstimulus interval (ISI) between the delivery of PNS and TMS needed to ensure the inter-arrival interval of 15ms at CM level was ~22.5ms. This closely resembles the ISI used in facilitatory PAS, demonstrated to induce STDP-like plasticity on the cortical level, leading to increases in MEP amplitudes (Stefan et al., 2000). This means that although PCMS- was designed to reduce effective transmission at the CM-level, it is possible that the ISI needed to facilitate such timing-specific interactions, also led to co-occurring plastic changes in cortical circuits.” (Lines 424-434)
- **PCMScoupled-control:** “...Nevertheless, the observation that PCMS_{coupled-control} led to increased MEP amplitudes provides the first evidence that repeated pairing of high intensity PNS (130% M_{max}) and TMS (150% rMT) at this interval increases corticospinal excitability. In conventional paired associative stimulation (PAS)(Stefan et al., 2000), paired stimulations are timed with an interstimulus interval (ISI) of ~25ms to target the sensorimotor cortical

circuitry mediating short latency afferent inhibition. However, it is less well studied how peripheral nerve stimulation with an ISI of ~107ms (CM IAI of 100ms) before TMS modulates the size of the MEP (Turco et al., 2018). The one study investigating this ISI found an inhibition of the MEP (Kotb et al., 2005), ascribing the effect to networks associated with long latency afferent inhibition (see e.g., Turco et al. (Turco et al., 2018) for discussion). It should be acknowledged that the peripheral stimulation at supramaximal intensities may have elicited sensory afferent activation, which at longer – but not short - latencies may have influenced cortical processing and potentially influenced the TMS pulse at the cortical level. Although caution is warranted when concluding on the mechanisms underlying the effects of PCMS_{coupled-control} (IAI 100ms) on corticospinal excitability, we suggest changes in the cortical sensorimotor network mediating LAI to account for the observed facilitation..” (lines 406-422)

Again, we want to underline that we do not argue that PCMS works through cortical circuits, but emphasize that the timing of PNS and TMS in the PCMS protocol makes it possible that the afferent volleys co-evoked by PNS of the ulnar nerve (and from skin receptors etc.) temporally coincide in cortical circuits with the delivery of the TMS pulse, as known from conventional studies on short-latency and long-latency sensorimotor integration using paired peripheral nerve stimulation and TMS (Turco et al., 2018). Other classical studies have established that repeated pairing of PNS and TMS at this interval (~22.5 ms), known as paired-associative stimulation (PAS), leads to increases in corticospinal excitability, likely due to increases in excitability at the cortical level (Stefan et al., 2000; Suppa et al., 2017).

As mentioned above, we acknowledge that this was not clear in the submitted version of the manuscript, and we have therefore modified these sections in the discussion as shown above to accommodate this. Nevertheless, the findings of the experiments in the manuscript are quite clear in the sense, that they demonstrate a positive priming effect on ballistic motor learning by targeting the level of the CM synapses, that effects on motor learning are preserved seven days later, that the effects are specific to the order and timing of stimulation in accordance with principles of Hebbian plasticity and this is further evidenced by the finding of an admittedly smaller but nevertheless significant negative effect by PCMS- on motor learning. Experiment IV provides further evidence that the effects of PCMS on corticospinal excitability are specifically related to the pairing of stimulations and to the specific timing of stimulations.

Regarding: *“Finally, acceleration traces from the single subject provided in the rebuttal do not appear to support a suppressive effect.”*

As mentioned in the *First Round of Revisions*: The inserted figure showed single subject acceleration traces from each protocol from each block. The figure is not included in the manuscript but provides the reviewer with information on these recordings and examples of traces. Importantly, a direct comparison of peak acceleration in absolute values cannot be justified due to the cross-over design. But we want to point out, that the chosen individual

traces represent the overall trend in the data, if the respective traces in the figure were normalized to their baseline value (PCMS+: B1=113.6%, B2=120.5%, B3=127.3%, PCMS-: B1=104.3%, B2=112.6%, B3=113.4%, PCMS_{coupled-control}: B1=109.1%, B2=115.9%, B3=116.3%).

Normalizing to baseline improves within-subject comparisons but is influenced by the test-retest effects (or rather learning-relearning). Only when averaging over the cohort we get a meaningful picture. These data demonstrate that across block 1 the subjects perform worse after the PCMS- protocol when compared to the coupled control protocol. This effect is no longer present in block two, whereas the performance is better following PCMS+. We argue that most likely learning is suppressed for PCMS- during block1 - or slightly delayed. This is also supported by the additional analysis provided here:

To address the Reviewers concerns, and support our findings we have now performed additional analyses to investigate these effects by analyzing the average of the first 10 trials of block 1 from the three protocols. If performance is similar in the beginning of Block 1, it would support the argument of delayed learning. Note, this analysis is only included in the Response to Reviewer comment. When average data is set up in a frequentist repeated measures ANOVA we get $F=1.025$, $p=0.369$, meaning no significant difference. Using Bayesian statistics (JASP Team 2023, version 0.17.3), and asking what the evidence is that groups are different at the first 10 trials of Block 1 (prior=0.5) we get a Bayes Factor = 0.332. If the hypothesis is turned around, we ask, what is the evidence of the groups being equal (prior 0.5) we get a Bayes Factor = 3.012 (see below for model details). Based on the work by van Doorn et al., (2021), this can be understood the way, that there is 'moderate' (Bayes Factor from 3-10) evidence for performance during the first 10 trials of block 1 being equal between PCMS conditions, and only 'weak' evidence of them being different (Bayes Factor around 1). This may be interpreted as no difference in motor performance between groups during the start of the first practice block, meaning that the observed performance difference during Block1 in Experiment III must be an effect on early learning. The finding that performance is not different between groups following stimulation but different during the first block of motor practice indicates slowed or lower initial learning for PCMS- during block 1, and this results provides further support to bidirectional behavioral effects and interpretations of Hebbian priming.

Model Comparison

Models	P(M)	P(M data)	BF _M	BF ₀₁	error %
Null model (incl. subject)	0.500	0.752	3.031	1.000	
TREATMENT (first 10 of Block1)	0.500	0.248	0.330	3.031	1.066

Note. All models include subject

Regarding: “All of these issues call into question whether there are bidirectional effects regardless of the small difference between conditions in the first block of practice.”

We agree with the reviewer that the initial submission had too conclusive statements regarding bidirectionality, and since the effects of PCMS- were smaller compared to the main focus of the manuscript on PCMS+ we modified the wording on bidirectionality since the first submission to balance conclusions. Nevertheless, we still argue that our data support that PCMS can exert timing-specific and bidirectional effects on motor learning dependent on timing between stimulations. We do acknowledge in the manuscript – and we agree with the reviewer that the behavioral effects of PCMS- are less pronounced compared to the effects of PCMS+, which are the main focus in the manuscript. For this reason, we have carefully rephrased the wording of the bidirectionality of the effects of PCMS in the revised manuscript.

Please find specific changes in relation to the presentation and interpretation of the findings in Experiment III in what follows:

In the end of the Introduction (and the start of the discussion) we focus on the positive, priming effects of PCMS (i.e. PCMS+), and we do not conclude on bidirectionality or the PCMS- effects:

- Abstract: We have removed the following sentence from the abstract: “At an inhibitory interval, subsequent learning was transiently impeded”

- Introduction: “*The positive priming effects of PCMS on spinal motor learning persist seven days after motor practice. This observation highlights the behavioral relevance of the findings. The observed priming effects on ballistic motor learning are in line with Hebbian learning rules demonstrating the importance of paired stimulations and of spike-timing*” (lines 68-72)
- Discussion: “*Finally, this study observed that priming effects on motor learning are in line with Hebbian learning rules (Experiment III).*”(lines 309-311)

In conclusion, we agree with the reviewer that the effects of PCMS- are smaller compared to PCMS+ which are indeed the main focus of the manuscript. Nevertheless, the findings do indeed support the interpretations on Hebbian priming effects since we observe timing-specificity throughout the included experiments, and the finding of – admittedly smaller – bidirectional effects. That said, we certainly agree with the Reviewer that more future studies are warranted in order to investigate mechanisms underlying these findings, and we have modified the discussion accordingly.

References used in Authors’ responses:

- Amoiridis, G. (1992): Median–ulnar nerve communications and anomalous innervation of the intrinsic hand muscles: An electrophysiological study. *Muscle & Nerve*, Vol. 15:5, pp. 576–579.
- Bagce, H.F., S. Saleh, S. V. Adamovich, J.W. Krakauer & E. Tunik (2012): Corticospinal excitability is enhanced after visuomotor adaptation and depends on learning rather than performance or error. *Journal of Neurophysiology*, American Physiological Society Bethesda, MD, Vol. 109:4, pp. 1097–1106.
- Bestmann, S. & J.W. Krakauer (2015): The uses and interpretations of the motor-evoked potential for understanding behaviour. *Experimental Brain Research*, Vol. 233:3, pp. 679–689.
- Bostock, H. & J.C. Rothwell (1997): Latent addition in motor and sensory fibres of human peripheral nerve. *Journal of Physiology*, Vol. 498:1, pp. 277–294.
- Christiansen, L., M.A. Urbin, G.S. Mitchell & M.A. Perez (2018): Acute intermittent hypoxia enhances corticospinal synaptic plasticity in humans. *ELife*, Vol. 7:, pp. 1–17.
- van Doorn, J., D. van den Bergh, U. Böhm, F. Dablander, K. Derks et al. (2021): The JASP guidelines for conducting and reporting a Bayesian analysis. *Psychonomic Bulletin and Review*, Vol. 28:3, pp. 813–826.
- Gelli, F., F. Del Santo, T. Popa, R. Mazzocchio & A. Rossi (2007): Factors influencing the relation between corticospinal output and muscle force during voluntary contractions. *European Journal of Neuroscience*, Vol. 25:11, pp. 3469–3475.
- Kotb, M.A., T. Mima, Y. Ueki, T. Begum, A.T. Khafagi et al. (2005): Effect of spatial attention on human sensorimotor integration studied by transcranial magnetic stimulation. *Clinical Neurophysiology*, Vol. 116:5, pp. 1195–1200.
- Lin, J.Z. & M.K.A.Y. Floeter (2004): Do F-wave measurements detect changes in motor neuron excitability. *Muscle and Nerve*, September, pp. 289–294.
- McDonnell, M.N. & M.C. Ridding (2006): Transient motor evoked potential suppression following a complex sensorimotor task. *Clinical Neurophysiology*, Vol. 117:6, pp. 1266–1272.
- Mcneil, C.J., J.E. Butler, J.L. Taylor & S.C. Gandevia (2013): Testing the excitability of human motoneurons. *Frontiers in Human Neuroscience*, Vol. 7:April, pp. 1–9.

- Meals, R.A. & M. Shaner (1983): Variations in digital sensory patterns: A study of the ulnar nerve—median nerve palmar communicating branch. *Journal of Hand Surgery, American Society for Surgery of the Hand*, Vol. 8:4, pp. 411–414.
- Panizza, M., J. Nilsson, B.J. Roth, J. Rothwell & M. Hallett (1994): The time constants of motor and sensory peripheral nerve fibers measured with the method of latent addition. *Electroencephalography and Clinical Neurophysiology/ Evoked Potentials*, Vol. 93:2, pp. 147–154.
- Stefan, K., E. Kunesch, L.G. Cohen, R. Benecke & J. Classen (2000): Induction of plasticity in the human motor cortex by paired associative stimulation. *Brain*, Vol. 123:3, pp. 572–584.
- Suppa, A., A. Quartarone, H. Siebner, R. Chen, V. Di Lazzaro et al. (2017): The associative brain at work: Evidence from paired associative stimulation studies in humans. *Clinical Neurophysiology, International Federation of Clinical Neurophysiology*, Vol. 128:11, pp. 2140–2164.
- Taylor, J.L. & P.G. Martin (2009): Voluntary motor output is altered by spike-timing-dependent changes in the human corticospinal pathway. *Journal of Neuroscience, Soc Neuroscience*, Vol. 29:37, pp. 11708–11716.
- Turco, C. V., J. El-Sayes, M.J. Savoie, H.J. Fassett, M.B. Locke et al. (2018): Short- and long-latency afferent inhibition; uses, mechanisms and influencing factors. *Brain Stimulation, Elsevier Ltd*, Vol. 11:1, pp. 59–74.
- Urbin, M.A., R.A. Ozdemir, T. Tazoe & M.A. Perez (2017): Spike-timing-dependent plasticity in lower-limb motoneurons after human spinal cord injury. *Journal of Neurophysiology*, Vol. 118:4, pp. 2171–2180.

REVIEWER COMMENTS

Reviewer #1 (Remarks to the Author):

Revision has improved the manuscript and I have no new comments.

This previous comment has not been satisfactorily addressed. “Figure 3 legend..... In 3f, it is not clear what the asterisks mean, i.e. are these comparisons to baseline or to both other time points? More specifically, what difference does the asterisk for Post practice PCMS- represent?”

It remains unclear what the asterisks mean when there are 3 time points within a protocol. The mean for Post-practice PCMS- does not appear different from baseline. Is it different from Post stimulation? By contrast, I assume that for both PCMS+ and PCMScontrol the corresponding asterisks indicate difference from baseline.

Reviewer #2 (Remarks to the Author):

The manuscript contains revisions as well as data from additional experiments intended to address issues noted in the previous reviews. These revisions and data pertain mostly to results from Experiment III, which are central to the authors’ interpretation. The PCMS- protocol does not suppress corticospinal excitability but, rather, facilitates it. The authors’ explanation about indirect relationships between excitability and learning are acknowledged. However, the same trend persists in the new data presented in the revised manuscript and is independent of practice. The finding that PCMS- confers a facilitatory effect on corticospinal excitability but a suppressive effect on early performance is still difficult to appreciate. Overall, the authors need to take greater care interpreting their data.

Learning is inferred from performance which, in this case, is the peak acceleration of index finger flexions. The small (~4%) suppressive effect of the PCMS- protocol on peak acceleration in only the first practice block and not in later blocks also is central to the authors’ interpretation. What does it mean if the theorized interaction with learning does not hold at the later time points? Use of this finding to support time specificity of the effect becomes even more difficult to appreciate given that the PCMS- protocol, independent of practice, increases excitability along the pathway mediating these finger movements. The comparison of the first 10 trials from the first block between

conditions is understood, but it is not entirely clear how it alleviates these issues nor how statements such as the following are fully supportable:

Lines 367-70: “Nevertheless, the results of Experiment III demonstrate that the priming effect of PCMS on motor learning are circuit and timing specific. Specifically, the effects observed conform to Hebbian learning rules in that they were contingent on temporal proximity of stimulation effects and governed by order of spike-timing.”

Lines 308-10: “Finally, this study observed that priming effects on motor learning are in line with Hebbian learning rules (Experiment III).”

Lines 289-90: “...effects are greater in magnitude when the timing of the paired stimulations are set to match principles of Hebbian plasticity.”

Lines 69-71: “The observed priming effects on ballistic motor learning are in line with Hebbian learning rules demonstrating the importance of paired stimulations and of spike-time.”

Although the additional experiments and explanation provided in the response are well thought, these issues still need to be reconciled and assume a more central place in this manuscript. It seems that content placed at the end of the supplementary methods is relevant and can help the reader understand the nuanced findings. To be clear, simply because the PCMS- protocol yielded less facilitation on MEP size than the PCMS+ protocol does not account for the fact that it did not exert its intended effect in most subjects.

Certain content also needs to be condensed as originally suggested to streamline presentation and get the reader to more impactful aspects of this work. Experiment I does not add much given the purpose and methods used in Experiment II. As a thought experiment, how would a human subject respond to receiving the treatment versus not receiving the treatment given any advanced knowledge as to the purpose of the experiments from consenting procedures? It seems that there is reasonable potential for a placebo effect in Experiment I. Regardless, results from Experiment I should be more concisely reported and serve as a lead into Experiment II where a sham was administered.

Other comments/questions:

-The abstract does not appear updated to account for the new data.

-Lines 60-63: This sentence is unclear.

-Line 133-34: Change “long-lasting” to “lasting” so as to not overstate results.

-Lines 191-94: Refine/condense and move these statements into Experiment III.

-Lines 240-47: These statements are confusing. That authors state that “...at post practice MEPs remained increased in the PCMS+ group, but decreased in both PCMS- and PCMS coupled-control, all non-significant within-group differences...” In Figure 3F, there are asterisks over post stimulation and post practice time points for all three protocols. Reviewer 1 inquired about the meaning of the asterisks in this figure, and the authors’ response indicates that it is a “within-protocol” comparison and explain “asterisks in Figure 3F for Post practice PCMS+ represents significantly higher MEP amplitudes post practice compared to PCMS+ at baseline.” Relatedly, the PCMS+ trend observed at the group level (3F) does appear reflected in the traces selected for an individual subject (3D), but the same groups trends from PCMS- and PCMS coupled-control protocols do not appear reflected in the individual subject traces. It is acknowledged that these are intended to be representative traces from the same subject, but how representative are they if they only reflect certain group-level trends and not others?

-Findings reported for Experiment II show that peak accelerations recorded 7 days after practice were higher at the group level than after the final practice block. It appears that this trend is observed in several subjects. Is it common to see performance on a novel task continue to improve further after a retention period/no further repetition on the task?

-There are limited details pertaining to procedures at baseline. How much practice accumulated prior to conditioning stimulation?

-Figure 4 does not contain any indication of significant effects/interactions noted on lines 268-76.

Revision-Round-3: Point-by-point Responses to Referees comments

We would like to thank both reviewers for their constructive feedback and suggestions. We have now carefully considered the inputs from the reviewers and implemented changes accordingly, and we believe that this has improved the quality of the manuscript. In the revised version of our manuscript, we have condensed the presentation of Experiment I as suggested, providing a faster transition to Experiment II, where a proper sham control is introduced in a double-blinded protocol. The restructured manuscript naturally provides more space to discuss another central aspect of the manuscript, namely the results from Experiment III and IV. Furthermore, we have included another sub-analysis in the Supplementary Discussion on Responders/Non-responders to the PCMS- protocol.

Please find our point-by-point revisions to your comments in what follows below.

Color coding:

- Responses to referees are color-coded **blue**
- Text included in 'Manuscript' and 'Supplementary Information' is color-coded **red**, and **green** text represents specific changes made in Revision-Round-3.

Reviewer #1 (Remarks to the Author):

Revision has improved the manuscript and I have no new comments.

Authors' response: We thank the reviewer for all previous comments and the acknowledgement of the revised manuscript.

This previous comment has not been satisfactorily addressed. "Figure 3 legend.....
In 3f, it is not clear what the asterisks mean, i.e. are these comparisons to baseline or to both other time points? More specifically, what difference does the asterisk for Post practice PCMS- represent?"

It remains unclear what the asterisks mean when there are 3 time points within a protocol. The mean for Post-practice PCMS- does not appear different from baseline. Is it different from Post stimulation? By contrast, I assume that for both PCMS+ and PCMS_{control} the corresponding asterisks indicate difference from baseline.

Authors' response: We apologize for not providing a satisfactory correction in the last round, and we thank the reviewer for noticing this inconsistency. We have now changed the Figures and figure legends so the asterisks reflect significant within-protocol comparisons $p < 0.05$ between time-points relative to baseline.

"The asterisk () indicates significant within-protocol comparisons $p < 0.05$ between time-points relative to baseline". (lines: 232-233).*

The asterisk at Post practice PCMS- have thus been removed, since MEPs at post practice is not significantly different compared to baseline. By contrast, MEPs at post practice are significantly different compared to baseline in both PCMS+ and PCMS_{coupled-control} as the Reviewer rightfully assumed.

Reviewer #2 (Remarks to the Author):

The manuscript contains revisions as well as data from additional experiments intended to address issues noted in the previous reviews. These revisions and data pertain mostly to results from Experiment III, which are central to the authors' interpretation. The PCMS- protocol does not suppress corticospinal excitability but, rather, facilitates it. The authors' explanation about indirect relationships between excitability and learning are acknowledged. However, the same trend persists in the new data presented in the revised manuscript and

is independent of practice. The finding that PCMS- confers a facilitatory effect on corticospinal excitability but a suppressive effect on early performance is still difficult to appreciate. Overall, the authors need to take greater care interpreting their data.

Authors' response: We want to thank the reviewer for the constructive feedback and the acknowledgment of our revisions.

Indeed, we were also surprised that all PCMS protocols in Experiment III led to an increase in MEP amplitudes. This led us to conduct Experiment IV, to investigate whether these findings could be replicated, and whether the effects were dependent on motor practice. As highlighted by the reviewer, the overall findings of increased corticospinal excitability following all three PCMS protocols persisted, and this was independent of motor practice. We still argue that our conceptual framework and interpretation of our results is valid. However, we agree with the reviewer that more clarification, precision, and nuances are needed to in certain parts of our interpretation of results.

Below we start with a summary of our overall response to Reviewer 2's comments, which is followed by a point-by-point response to each question/concern.

Learning is inferred from performance which, in this case, is the peak acceleration of index finger flexions. The small (~4%) suppressive effect of the PCMS- protocol on peak acceleration in only the first practice block and not in later blocks also is central to the authors' interpretation. What does it mean if the theorized interaction with learning does not hold at the later time points? Use of this finding to support time specificity of the effect becomes even more difficult to appreciate given that the PCMS- protocol, independent of practice, increases excitability along the pathway mediating these finger movements. The comparison of the first 10 trials from the first block between conditions is understood, but it is not entirely clear how it alleviates these issues nor how statements such as the following are fully supportable:

Lines 367-70: "Nevertheless, the results of Experiment III demonstrate that the priming effect of PCMS on motor learning are circuit and timing specific. Specifically, the effects observed conform to Hebbian learning rules in that they were contingent on temporal proximity of stimulation effects and governed by order of spike-timing."

Lines 308-10: “Finally, this study observed that priming effects on motor learning are in line with Hebbian learning rules (Experiment III).”

Lines 289-90: “...effects are greater in magnitude when the timing of the paired stimulations are set to match principles of Hebbian plasticity.”

Lines 69-71: “The observed priming effects on ballistic motor learning are in line with Hebbian learning rules demonstrating the importance of paired stimulations and of spike-time.”

Overall authors’ response: We share the reviewer’s concerns regarding the relationship between the behavioral and electrophysiological findings, and we agree that the short-lived inhibitory behavioral effects after PCMS-, although significant, call for a nuanced discussion on the bidirectionality of the priming effects.

First, it is relevant to underline that the small effects of PCMS- on early ballistic learning (block 1) are only slightly smaller than the inhibitory after-effects (9%) on ‘voluntary motor output’ reported by Taylor & Martin (2009). The remaining studies employing the PCMS- protocol only assessed effects in MEPs evoked with TMS and/or TES and therefore did not evaluate behavioral effects of PCMS- (Bunday & Perez, 2012a; Urbin et al., 2017). Furthermore, it is not straight forward to compare the magnitude of priming effects on ballistic *learning* observed in this study with after-effect on submaximal ballistic contractions reported in the study by Taylor & Martin (2009). Finally, the small effect of 4% is likely an underestimation of the actual inhibitory effect of the protocol, which is caused by the cross-over design of Experiment III. Based on previous constructive comments from the Reviewer, we have addressed and acknowledged this explicitly in the discussion of our manuscript (Lines 367-372).

Below we propose two mechanisms that both could explain the coinciding findings of impaired early learning and increased MEP amplitudes after PCMS-. Finally, we propose revised versions of the statements highlighted by the reviewer.

First, as argued in our previous response to reviewer, and stated in the Discussion section of the manuscript, the lack of MEP suppression after PCMS- could reflect opposing effects in supraspinal circuits that are also activated during the PCMS protocol. The PCMS- protocol had an average interstimulus interval (ISI) of 22.5 ms spanning 21-24ms, which is aligned

to the timing needed for facilitatory interaction at the cortical level (i.e. cortical PAS). Although the protocol may have had inhibitory aftereffects at the level of the CM-synapses, it is also likely, that the protocol could have led to facilitatory effects at a cortical level. Importantly, MEPs evoked by cortical stimulation with TMS represent overall corticospinal excitability and would thus reflect the net change at both cortical and spinal levels. As the reviewer points out, such cortical facilitation could potentially also influence the motor behavior. We acknowledge this, and cannot exclude the possibility that a potential cortical facilitation may have affected the behavioral results. The short-lasting negative effects on learning could in fact reflect a small difference in the temporal evolution of opposing priming processes at the cortical and spinal levels, respectively. On the other hand, the STDP-like plasticity occurring within the primary motor cortex (M1) when sensory volleys arrive from the primary somatosensory cortex (S1) time-locked to TMS may be confined to a system that supports a transcortical sensorimotor loop and can be excited by low intensity TMS. While we thus acknowledge the limitations, we nevertheless observed a significant negative behavioral effect of PCMS- on early ballistic motor learning. This finding is consistent with the hypothesis, and the result should therefore also be reported. In acknowledgment of the Reviewer, we have clarified the interpretation of the results in the discussion.

Secondly, inspired by the Reviewers comments we have considered another potential explanation for the contrasting finding of decreased early learning and increased MEP amplitudes after PCMS-. After careful consideration we have chosen to include the proposed alternative explanation presented below in the Supplementary Discussion. This follows up on the PCMS- responder/non responder discussion on corticospinal excitability. We do not include it in the main discussion, to keep main focus on the positive priming effects of the PCMS+ protocol in the manuscript. The following paragraph has been included in the Supplementary Discussion:

An alternative mechanism for the contrasting finding that PCMS- decrease early learning and increase MEP amplitudes could be the result of single-pulse TMS and PCMS not targeting the same parts of the motoneuronal pool. The single-pulse TMS with intensity of 120% rMT was used to record MEP's before and after PCMS. It can be assumed that an MEP evoked with this intensity involves activation of approximately half of the motoneuronal pool that can be excited by TMS during rest (Devanne et al., 1997). In PCMS-, the high-intensity TMS pulse (150% rMT) is closer to the maximum level and can therefore be

expected to reflect a substantially larger activation of the motoneuronal pool. The timely paired PNS is supramaximal and antidromic volleys can be expected in all fibers of the stimulated nerve. However, the antidromic activation of the motoneurons favor the larger motoneurons due to collision between H-waves and antidromic activity in the smaller, which further decouples the affected motoneuronal pool from the pool excited by single pulse TMS at 120%. In contrast, movement at maximal speed or with maximal rate of force development depend on both high rate of motor unit recruitment and motoneuronal firing rates (Del Vecchio et al., 2019). In this light, it is possible that the ability to increase recruitment rates with ballistic motor practice was hampered due to the inhibitory priming that effected corticomotoneuronal synapses on motoneurons that are recruited during maximal ballistic contractions but less so from descending activity evoked when stimulating M1 120 % of rMT.

As we observed a facilitation of corticospinal excitability after PCMS- and a facilitation of both MEPs and learning after the PCMS+ protocol, we propose that the mechanisms outlined above contributed to the observed behavioral and electrophysiological effects.

Clarification of used terminology and revised statements

We agree with the reviewer that more clarification and precision is needed to certain statements in the Manuscript. Below we have defined key terms, and streamlined the terminology throughout the manuscript.

Temporal proximity – This term refers to how close in time stimulations will arrive at the CM-synapse. Stimulations are timed so interarrival intervals at the CM-synapse are closely related to each other in time (PCMS+, interarrival interval of 2 ms) compared to longer time delays at the CM-synapse (PCMS-, interarrival interval of 15 ms; and PCMScoupled-control, interarrival interval of 100 ms). We now use the term temporal proximity more consistently in the manuscript, opposed to the term ‘timing’, as we acknowledge that ‘timing’ could refer to other aspects of the PCMS protocols.

Order of spike timing- This term refers to the order by which the CM-synapse is activated by our paired stimulations. I.e., when the pre-synapse is activated before the post-synapse at the CM-synapses (PCMS+) compared to when the post-synapse is activated before the pre-synapse at the CM-synapses (PCMS- and PCMScoupled-control).

Below, we present revised versions of the statements highlighted by the reviewer. With this in mind, we still argue that our behavioral findings conform to Hebbian learning rules, while we do acknowledge that the effects of PCMS- are small, as pointed out in the discussion.

Regarding: Lines 367-70: “Nevertheless, the results of Experiment III demonstrate that the priming effect of PCMS on motor learning are circuit and timing specific. Specifically, the effects observed conform to Hebbian learning rules in that they were contingent on temporal proximity of stimulation effects and governed by order of spike-timing.”

We agree with the reviewer, and have now made the statement more specific in relation to our results.

“Nevertheless, the results of Experiment III, provide proof-of-principle that the priming effects of PCMS on ballistic learning conform to Hebbian learning rules, in that they are contingent on temporal proximity of paired stimulations and in that they are governed by order of spike-timing.” (*Lines 380-383*)

We now avoid any conclusive statements about circuit-specificity, since all of our protocols were intended to target CM-synapses (even though the ISI from PCMS- and PCMS_{coupled-control} could have had an effect on cortical levels as addressed in the discussion). Furthermore, we specify “that priming effects of PCMS on **ballistic learning...**” opposed to the previous, more general wording “...the priming effect of PCMS...”. We believe that this helps the interpretation of our results, as it provides a clearer focus on the actual results being discussed: here, the behavioral findings on ballistic learning rather than electrophysiological findings on changes in the corticospinal excitability. In the revised section-snippet highlighted above:

Temporal proximity – refers to the greater effect on ballistic learning when the stimulations at the CM-synapse are closer to each other in time (PCMS+, interarrival interval of 2 ms) compared to longer time delays at the CM-synapse (PCMS-, interarrival interval of 15 ms; and PCMS_{coupled-control}, interarrival interval of 100 ms).

Order of spike timing- refers to the observed greater effect on ballistic learning when the pre-synapse is activated before the post-synapse at the CM-synapses (PCMS+) compared to when the post-synapse is activated before the pre-synapse at the CM-synapses (PCMS- and PCMS_{coupled-control}).

Regarding: Lines 308-10: “Finally, this study observed that priming effects on motor learning are in line with Hebbian learning rules (Experiment III).”

This statement is placed in the beginning of the discussion (before the statement discussed above Line 367-70). We believe that the explanatory sentence following this statement is important to fully understand the argumentation – and we have revised the following sentence, to be streamlined with terminology (as explained above):

“Finally, this study observed that priming effects on motor learning are in line with Hebbian learning rules (Experiment III): PCMS with a short inter-arrival interval of pre-synaptic activation just before post-synaptic activation at the level of the CM-synapse improved motor learning compared to PCMS protocols with longer timing intervals and opposite order of spike-timing.” (Lines: 320-324).

Regarding: Lines 289-90: “...effects are greater in magnitude when the timing of the paired stimulations are set to match principles of Hebbian plasticity.”

This sentence summarizes the findings from Experiment IV. It has been modified and moved in the revised version of the manuscript. It now reads:

“Furthermore, unpaired stimulations did not lead to changes in corticospinal excitability, showing that facilitation of corticospinal excitability in the present study was coupling-dependent and required paired stimulations. Finally, all PCMS protocols increased corticospinal excitability, with the largest effect observed after PCMS+ demonstrating the importance of stimulus timing also for changes in corticospinal excitability.” (Lines 292-297)

The following has been moved to the end of the discussion on corticospinal excitability:

“Collectively, the findings from Experiment III and IV show a facilitation of corticospinal excitability measured via motor evoked potentials following PCMS. However, the effects of PCMS on corticospinal excitability are larger in magnitude when temporal proximity between stimulations is closer and order of spike timing is set to match the principles of Hebbian plasticity.” (Lines 484-488)

Regarding: Lines 69-71: “The observed priming effects on ballistic motor learning are in line with Hebbian learning rules demonstrating the importance of paired stimulations and of spike-time.”

We have revised the sentence to bring more clarity:

“The observed priming effects of PCMS on ballistic motor learning are to a wide extent in line with Hebbian learning rules demonstrating the importance of close temporal proximity and order of spike timing.” (Lines 82-85)

The phrase “...to a wide extent...” avoids a sharp conclusive statement. Nevertheless, we still argue that the sentence is supported by our argumentation throughout the manuscript.

The following sections include more detailed point-by-point responses to each question/concern:

Regarding: *“What does it mean if the theorized interaction with learning does not hold at the later time points?”*

We agree with the reviewer, that from a behavioral point-of-view, the small effect of PCMS- during the first practice block is of less importance, since it does not hold through at the end of practice. The fact that the small PCMS- effect during the first practice block is not present during later practice can potentially be explained by a superior effect of motor practice. That is, undergoing motor practice in block 2 and 3 seem to have been sufficient to “overcome” the small effect of PCMS- on initial/early learning observed in block 1.

From a mechanistic perspective, we argue that the small – but nevertheless significant – behavioral effect of PCMS- is still interesting, since it supports a bidirectional modulation of ballistic learning. That said, we naturally agree with the previous and present comments from the Reviewer that caution is warranted when making interpretations about this bidirectionality. For this reason, we focus our interpretation of the findings more on temporal proximity and the order of spike timing aspect of Hebbian learning theory. This is discussed in more detail for the specific statements pointed out by the reviewer (see above).

Regarding: *“Use of this finding to support time specificity of the effect becomes even more difficult to appreciate given that the PCMS- protocol, independent of practice, increases excitability along the pathway mediating these finger movements”*

We thank the Reviewer for this comment. We acknowledge that the results on the increase in corticospinal excitability of PCMS- coupled with a decrease in early motor learning can seem difficult to reconcile and should be interpreted cautiously. We now accommodate the reviewer’s inputs, by bringing in new perspectives in the Supplementary Discussion:

Responders/Non-responders in PCMS- (presented above). Moreover, because of the complexity, we also dedicate a larger part of the manuscript to thoroughly discuss the findings with great care and detail as referenced below:

“In Experiment III, we also compared effects of PCMS+ to two other PCMS protocols designed to leave transmission at the CM synapse unchanged (PCMS_{coupled-control}) and designed to reduce effective CM transmission at the spinal level (PCMS-). We found again a positive priming effect of PCMS+ and a smaller and shortlived negative effect of PCMS- on ballistic motor learning. Interestingly, we found increases in MEP amplitudes in all three protocols. These effects on corticospinal excitability were replicated in another sample in Experiment IV. Importantly, our results suggest that increasing corticospinal excitability does not improve motor learning per se; only when the targeted plasticity is directed at the neural circuitry that underpins the specific motor behavior – i.e. PCMS+ aimed at the CM level – then subsequent ballistic motor learning is enhanced. This shows that the relationship between changes in corticospinal excitability and motor performance is not straightforward, and this is a commonly observed phenomenon in the literature (Bagce et al., 2012; Gelli et al., 2007; McDonnell & Ridding, 2006). (Lines 407-419)

.... And again at the following section: *“Furthermore, we did not find a suppressive effect of PCMS- on corticospinal excitability. Instead, a significant facilitation of MEP amplitudes was observed after PCMS-. At first glance, this result is at odds with previous reports, but in contrast to Taylor & Martin (2009) (Taylor & Martin, 2009) we evaluated corticospinal excitability with MEPs evoked by cortical TMS and not CMEPs, a measure of subcortical excitability. One study found that inhibitory PCMS was accompanied by suppressed corticospinal excitability, but only for individuals in whom, the afferent input reduced MEP size during the PCMS protocol (Urbin et al., 2017). The lack of a suppressive effect of inhibitory PCMS on corticospinal excitability in the present study could thus be related to electrophysiological non-responders. We did however not see any link between responders/non-responders and motor performance (this is elaborated further in the Supplementary Discussion and Supplementary Fig. 4). Interestingly, when targeting FDI, the average interstimulus interval (ISI) between the delivery of PNS and TMS needed to ensure the inter-arrival interval of 15ms at CM level was ~22.5ms. This closely resembles the ISI used in facilitatory PAS, demonstrated to induce STDP-like plasticity at the cortical level, leading to increased MEP amplitudes (Stefan et al., 2000). This means that although*

PCMS- was designed to reduce effective transmission at the CM-level, it is possible that the ISI needed to induce such timing-specific interactions, also led to co-occurring changes in cortical circuits. It is well-known that the MEP size reflects excitability along the corticomotor pathway as well as upstream of M1 (Bestmann & Krakauer, 2015). In this light, the MEP amplitude after PCMS- likely reflects the sum of cortical and spinal plasticity including potential opposing effects. However, as we did not evaluate changes in muscle response to electrical cervicomedullary stimulation, we are limited to speculating on the loci of PCMS induced changes in excitability.” (Lines: 436-459).

Admittedly, we were also initially puzzled by the findings on corticospinal excitability across protocols. This is also the reason why we performed – and included - a new experiment (Experiment IV) in the last round of revisions to further investigate the effects of these protocols on corticospinal excitability without any motor practice. Importantly, the increase in corticospinal excitability for all protocols was replicated in this follow-up experiment. This suggests that the effect is replicable.

Regarding: *“The comparison of the first 10 trials from the first block between conditions is understood, but it is not entirely clear how it alleviates these issues nor how statements such as the following are fully supportable.”*

Thank you for your acknowledgement of the additional analysis provided in last revision round. We agree with the reviewer that more clarity is needed to fully support our interpretation and argumentation regarding this analysis.

The analysis of the first 10 trials from the first block was used to investigate whether the paired stimulations had any acute effect on **ballistic motor performance as opposed to ballistic motor learning**. By acute effect on ballistic motor performance, we mean the average performance within the first 10 trials of practice. As shown in the analysis, we do not see any acute effect on motor performance. Instead, we observe the effect evolve during the first practice block. That is why we focus on the interaction effect of the paired stimulations and subsequent motor practice, in other words, we see an effect on **ballistic motor learning**. By ballistic motor learning, we refer to the entire practice block. This distinction between acute effects on motor performance and interaction effects with motor practice is a key aspect, and the learning perspective and interactions is what separates this article from previous findings. This is mentioned in the discussion:

“...Previous studies investigating the effects of PCMS protocols have found positive effects of PCMS on motor functions(Bunday & Perez, 2012b) but not on maximal voluntary contraction(D’Amico et al., 2018; Dongés et al., 2019). In the present study, we demonstrate a positive priming effect of PCMS compared to rest and sham on ballistic motor learning....”(Lines 343-346)

Although the additional experiments and explanation provided in the response are well thought, these issues still need to be reconciled and assume a more central place in this manuscript. It seems that content placed at the end of the supplementary methods is relevant and can help the reader understand the nuanced findings. To be clear, simply because the PCMS- protocol yielded less facilitation on MEP size than the PCMS+ protocol does not account for the fact that it did not exert its intended effect in most subjects.

Authors’ response: We thank the reviewer for appreciating the added experiment and analysis (Experiment IV). In line with the other comments from Reviewer 2 on the overall structure of the manuscript, we have condensed the presentation of Experiment I and instead now provide a thorough discussion of the results from Experiment III and Experiment IV placing it as a more central part of the manuscript.

Regarding: *“It seems that content placed at the end of the supplementary methods is relevant and can help the reader understand the nuanced findings.”*

We thank the reviewer for highlighting our analysis in the Supplementary Information (Supplementary Figure 4). In line with the first (and only) previous study reporting this observation, we find that MEP amplitudes *during* inhibitory PCMS (PCMS₋) are proposed as a marker for electrophysiological responders and non-responders (Urbin et al., 2017). Experiment III showed that PCMS₋ suppressed early learning, assessed as performance during practice block 1, compared PCMS₊ and PCMS_{coupled-control}. Whether this electrophysiological marker of PCMS₋ responders and non-responders also bears predictive value for the behavioral data is unknown. Based on the comments from Reviewer 2, we decided to scrutinize how the subgroups of electrophysiological “responders” and “non-responders” performed during practice block 1. Please find the results from this analysis enclosed in the updated Figure S4 below (Figure S4D). Visual inspection of motor performance during the first practice block indicates no difference between PCMS₋

responders and non-responders. The proposed electrophysiological marker of PCMS-responder/non-responder does not seem to predict motor performance in practice block 1.

Supplementary Figure 4. Responders vs Non-responders to PCMS-. **a)** MEP amplitudes post PCMS- (%baseline), unexpectedly, data show that 12 participants had increased MEP after PCMS- (here categorized as non-responders), and only 6 participants with decreased MEP (responders) **b)** MEP amplitudes during PCMS-, divided in whether participants responded as intended to the PCMS- (decreased MEP amplitude post PCMS-). Each datapoint is the individual mean of 10 stimulations. **c)** Group and individual means of MEP amplitudes during all stimulations delivered during PCMS- presented as % of MEP amplitudes at baseline. Please note that for the majority of individuals responding as intended to the PCMS-, the MEP amplitudes delivered during PCMS- (150% RMT) were smaller than MEP amplitudes delivered at baseline (120% RMT) showing an effective reduction of MEP amplitudes. **d)** Responders and non-responders showed similar motor performance, peak acceleration (% of baseline), during the first practice block, each datapoint is the individual mean of the 50 trials in block 1. Error bars indicate standard deviation.

It should be kept in mind that only 6 participants are in classified as “responders”. This likely limits the statistical power of this subgroup analysis. For this reason, we have decided to keep this analysis in the Supplementary Information to keep the main manuscript concise and focused on the main research questions.

However, we have now elaborated on the interpretation and discussion of these results in the manuscript:

The following has been added to the Supplementary Discussion: Responders/Non-Responders in PCMS-: *“Experiment III showed that PCMS- suppressed early learning, assessed as performance during practice block 1, compared to PCMS+ and PCMS_{coupled-control}. We speculated whether the electrophysiological marker of responsiveness to PCMS- could also explain differences in changes in ballistic motor performance during the early learning phase (practice block 1). However, this did not seem be the case (Supplementary Fig. 4d). This suggests that other mechanisms in addition to a ‘pure’ corticomotoneuronal STDP rule influence the electrophysiological after-effects of PCMS.”*

In the Discussion: *“Furthermore, we did not find a suppressive effect of PCMS- on corticospinal excitability. Instead, a significant facilitation of MEP amplitudes was observed after PCMS-. At first glance, this result is at odds with previous reports, but in contrast to Taylor & Martin (2009)(Taylor & Martin, 2009) we evaluated corticospinal excitability with MEPs evoked by cortical TMS and not CMEPs, a measure of subcortical excitability. One study found that inhibitory PCMS was accompanied by suppressed corticospinal excitability, but only for individuals in whom, the afferent input reduced MEP size during the PCMS protocol(Urbin et al., 2017). The lack of a suppressive effect of inhibitory PCMS on corticospinal excitability in the present study could thus be related to electrophysiological non-responders. We did however not see any link between responders/non-responders and motor performance (this is elaborated further in the Supplementary Discussion and Supplementary Fig. 4).”* (Lines 436-447)

Regarding: *“To be clear, simply because the PCMS- protocol yielded less facilitation on MEP size than the PCMS+ protocol does not account for the fact that it did not exert its intended effect in most subjects.”*

We completely agree with the Reviewer on this point. We would also like to point out that this has never been our intended explanation/interpretation and we apologize for any misunderstandings. We have now added the text below (in green) to the summarizing paragraph of Experiment IV in the discussion. We hope that this makes the interpretation of these results clearer.

“Experiment IV additionally demonstrated in another independent sample of individuals, that all PCMS protocols increased corticospinal excitability compared to rest. Notably, the PCMS+ led to larger increases in corticospinal excitability compared to PCMS- and PCMS_{coupled-control}. As was the case in Experiment III, the PCMS- protocol did not result in a suppression of MEP amplitudes.” (Lines 474-479).

We do however, want to highlight once again, that we discuss in detail a plausible explanation for the ‘unexpected’ (based on previous studies) facilitatory effect of PCMS- on MEP amplitudes:

“Furthermore, we did not find a suppressive effect of PCMS- on corticospinal excitability. Instead, a significant facilitation of MEP amplitudes was observed after PCMS-. At first glance, this result is at odds with previous reports, but in contrast to Taylor & Martin (2009)(Taylor & Martin, 2009) we evaluated corticospinal excitability with MEPs evoked by cortical TMS and not CMEPs, a measure of subcortical excitability. One study found that inhibitory PCMS was accompanied by suppressed corticospinal excitability, but only for individuals in whom, the afferent input reduced MEP size during the PCMS protocol(Urbin et al., 2017). The lack of a suppressive effect of inhibitory PCMS on corticospinal excitability in the present study could thus be related to electrophysiological non-responders. We did however not see any link between responders/non-responders and motor performance (this is elaborated further in the Supplementary Discussion and Supplementary Fig. 4). Interestingly, when targeting FDI, the average interstimulus interval (ISI) between the delivery of PNS and TMS needed to ensure the inter-arrival interval of 15ms at CM level was ~22.5ms. This closely resembles the ISI used in facilitatory PAS, demonstrated to induce STDP-like plasticity at the cortical level, leading to increased MEP amplitudes (Stefan et al., 2000). This means that although PCMS- was designed to reduce effective transmission at the CM-level, it is possible that the ISI needed to induce such timing-specific interactions, also led to co-occurring changes in cortical circuits. It is well-known that the MEP size reflects excitability along the corticomotor pathway as well as upstream of M1 (Bestmann & Krakauer, 2015). In this light, the MEP amplitude after PCMS- likely reflects the sum of cortical and spinal plasticity including potential opposing effects. However, as we did not evaluate changes in muscle response to electrical cervicomedullary stimulation, we are limited to speculating on the loci of PCMS induced changes in excitability.” (Lines: 436-459).

Certain content also needs to be condensed as originally suggested to streamline presentation and get the reader to more impactful aspects of this work. Experiment I does not add much given the purpose and methods used in Experiment II. As a thought experiment, how would a human subject respond to receiving the treatment versus not receiving the treatment given any advanced knowledge as to the purpose of the experiments from consenting procedures? It seems that there is reasonable potential for a placebo effect in Experiment I. Regardless, results from Experiment I should be more concisely reported and serve as a lead into Experiment II where a sham was administered.

Authors' response: We have now condensed several paragraphs in the manuscript, especially relating to Experiment I. We believe that the more condensed content and tight structure has increased the quality of the manuscript. Nevertheless, we sincerely believe that Experiment I deserves to be presented separately since this experiment preceded and formed the basis of Experiment II. We also want to highlight the important fact that the results from Experiment I are replicated in Experiment II.

Specifically, we have taken the following steps to condense the presentation of Experiment I:

- The previous beginning to the Experiment I paragraph in the Result section has been condensed and moved to the Introduction:
 - o *“In Experiment I, we individualized PCMS so that the descending corticospinal volleys elicited by TMS of the M1 hand area arrived at the corticomotoneuronal pre-synapse 2 ms before a peripherally triggered antidromic volley in the motor axons arrived at the post-synapse. This protocol (referred to as ‘PCMS+’) has consistently been shown to increase corticospinal excitability in humans (Bunday & Perez, 2012b; Shulga et al., 2015; Taylor & Martin, 2009)”.* (Lines 59-63)
- Concluding statements from Experiment I have been moved to Experiment II. In this way, we summarize the findings from Experiment I and II as a more cohesive story.
 - o *“Similar to the results in Experiment I, there was no difference at the beginning of practice (B1), indicating that the effects of PCMS+ in fact interacted with motor practice. Indeed, a significant difference was observed when comparing changes in performance from the beginning of practice (B1) to the end of practice (B3)”* (Lines 141-146)
- The paragraph on Mmax and F-waves has also been moved to Experiment II, and presented for Experiment I, II, III as a whole. This also bridges to the combined analysis on EMG from Experiment I and II.
 - o *“Analyses of M_{max} and F-wave amplitudes demonstrated no significant main effects across experiments (see Supplementary Fig. 3 for Experiment I, II & III), suggesting that α -motoneuronal excitability and peripheral recording conditions were stable, and that the normalization to M_{max} is unlikely to explain the observed group differences in corticospinal excitability..”* (Lines 173-177)

Regarding: *“As a thought experiment, how would a human subject respond to receiving the treatment versus not receiving the treatment given any advanced knowledge as to the purpose of the experiments from consenting procedures? It seems that there is reasonable potential for a placebo effect in Experiment I”:*

We agree with the reviewer, and this of course provides the motivation/need for performing Experiment II. Please note that we explicitly highlight the potential placebo effect as a limitation of Experiment I. This indeed provides a clear bridge to Experiment II:

“We cannot rule out that the observed between-group difference in average performance in the first part of block 1 of motor practice could be due to a placebo effect since the design of Experiment I was not blinded nor placebo or sham-controlled. To resolve these questions, we introduced a sham protocol that mimicked the perceptual experience of PCMS+ in the double-blinded Experiment II.” (Lines 125-129)

Other comments/questions:

-The abstract does not appear updated to account for the new data.

Authors' response: Thank you. Please see the updated version of the Abstract below, with the addition of the data Experiment IV:

“Motor learning is governed by experience-dependent plasticity in relevant neural circuits. In four experiments, we provide the first evidence and a double-blinded, sham-controlled replication (Experiment I-II) demonstrating that motor learning involving ballistic index finger movements is improved by preceding paired corticospinal-motoneuronal stimulation (PCMS), a human model for exogenous induction of spike-timing-dependent plasticity. The behavioral effects of PCMS targeting corticomotoneuronal (CM) synapses are order- and timing-specific and partially bidirectional (Experiment III). Specifically, when PCMS is timed to arrive at a facilitatory interval at CM-synapses, learning is improved. This protocol also led to larger increases in corticospinal excitability compared to control protocols, and unpaired stimulations did not increase corticospinal excitability (Experiment IV). Our findings demonstrate that non-invasively induced plasticity governed by Hebbian learning rules interacts positively with experience-dependent plasticity to promote motor learning. This offers a mechanistic rationale to enhance motor practice effects in neurorehabilitation by priming sensorimotor training with individualized PCMS.”. (Lines: 16-30)

-Lines 60-63: This sentence is unclear.

Authors' response: We agree with the Reviewer, the sentence has been rephrased:

“In Experiment IV, we further scrutinized the effects of PCMS on corticospinal excitability. Specifically, we explored whether effects of PCMS on corticospinal excitability required pairing of peripheral and cortical stimulations by assessing the effects of both paired and unpaired repetitive TMS and PNS on corticospinal excitability.” (Lines 68-72).

-Line 133-34: Change “long-lasting” to “lasting” so as to not overstate results.

Authors' response: This has now been corrected.

-Lines 191-94: Refine/condense and move these statements into Experiment III.

Authors' response: Thank you for this suggestion. The statements have been condensed and moved from the end of Experiment II and into Experiment III. The new introductory results paragraph for Experiment III is presented below:

“Experiment III: Timing-dependent and network-specific effects of PCMS

Cellular STDP is characterized by after-effects, which depend on both the temporal proximity and order of pre- and postsynaptic spiking and the effects can be bidirectional (Bi & Poo, 1998). To investigate if similar rules of order and timing-specificity apply to the network mediating the priming effects on ballistic motor learning shown in Experiment I and II, we conducted a within-subject cross-over experiment. In Experiment III, we investigate whether the effects of PCMS on motor learning were dependent on temporal proximity and order of spike timing at the level of the CM-synapses. Moreover, the experiment investigated whether PCMS can lead to bidirectional priming effects in humans and hence be capable of promoting or conversely decreasing ballistic learning depending on spike-timing.” (Lines 193-203).

-Lines 240-47: These statements are confusing. That authors state that “...at post practice MEPs remained increased in the PCMS+ group, but decreased in both PCMS- and PCMS coupled-control, all non-significant within-group differences...” In Figure 3F, there are asterisks over post stimulation and post practice time points for all three protocols. Reviewer 1 inquired about the meaning of the asterisks in this figure, and the authors' response

indicates that it is a “within-protocol” comparison and explain “asterisks in Figure 3F for Post practice PCMS+ represents significantly higher MEP amplitudes post practice compared to PCMS+ at baseline.” Relatedly, the PCMS+ trend observed at the group level (3F) does appear reflected in the traces selected for an individual subject (3D), but the same groups trends from PCMS- and PCMS coupled-control protocols do not appear reflected in the individual subject traces. It is acknowledged that these are intended to be representative traces from the same subject, but how representative are they if they only reflect certain group-level trends and not others?

Authors’ response: We apologize for the confusion. We have now corrected and clarified these statements (below in green):

Regarding the figure legend in Figure 3. We have now changed the wording and figures to avoid any misunderstandings. The asterisks reflect significant within-protocol comparisons $p < 0.05$ between time-points relative to baseline. The asterisk at Post practice PCMS- has thus been removed, since MEPs at post practice is not significantly different from baseline. By contrast, MEPs at post practice are significantly different compared to baseline in both PCMS+ and PCMS_{coupled-control}. The following has been added to Figure 1, 2, 3 legends:

“The asterisk (*) indicates significant within-protocol comparisons $p < 0.05$ between time-points relative to baseline”. (Lines: 232-233)

Regarding *“Relatedly, the PCMS+ trend observed at the group level (3F) does appear reflected in the traces selected for an individual subject (3D), but the same groups trends from PCMS- and PCMS coupled-control protocols do not appear reflected in the individual subject traces. It is acknowledged that these are intended to be representative traces from the same subject, but how representative are they if they only reflect certain group-level trends and not others?”*

Authors’ response: We acknowledge this input from the reviewer. We now show single subject MEP traces, different individuals from their first test day (thus aligned with the presentation in Figure 1d and 2d).

-Findings reported for Experiment II show that peak accelerations recorded 7 days after practice were higher at the group level than after the final practice block. It appears that

this trend is observed in several subjects. Is it common to see performance on a novel task continue to improve further after a retention period/no further repetition on the task?

Authors' response: We thank the reviewer for this question. Indeed this is an interesting observation. We want to clarify however, that from end of practice (last 10 trials of practice block 3) to the first 10 trials on day 7, we do not see an increase in performance in group mean performance (Figure 2c). For PCMS+, on group level we see maintained performance on day 7 compared to end of practice on main day, and indeed 5/10 participants show an increase. For Sham we see a drop in performance from end of practice on main day to day 7.

If the Reviewer refers to the improved performance from the entire practice block 3 (50 trials) to the entire block on day 7 (50 trials) (Figure 2B, inserted plot), we want to clarify/highlight that this difference also evolves through the practice block. This indicates that the introduction of more practice trials on day 7 leads to continued learning after one week. This is an important finding, since it shows that the participants' learning curve was not saturated and that they could still improve with additional motor practice.

Regarding the question: *“Is it common to see performance on a novel task continue to improve further after a retention period/no further repetition on the task?”*

In a classical motor behavior-memory framework, it would be common to see a slight performance drop when a given participant retrieves a motor memory at a delayed retention test (Kantak & Winstein, 2012). Whether performance would continue to improve during a retention period highly depends on the given task, and indeed, if no further repetition is introduced, there is likely to be a decay and conversely if more practice is introduced, continued learning will likely occur. Again, this depends on the given “novel” task. Multiple studies have found offline gains following motor practice, in particular for tasks involving motor sequence learning (see e.g. Krakauer et al., 2019).

More relevant to the present manuscript, we looked at previous studies using ballistic finger tasks:

Many of the studies using ballistic motor learning tasks comparable to the one used in the present study did not include a delayed retention test (>1 day) (Carroll et al., 2008; Cirillo et al., 2010; Classen et al., 1998; Giesebrecht et al., 2012; Lauber et al., 2013; Lee et al., 2010;

Rogasch et al., 2009). Earlier studies have used ballistic motor tasks to test interference with motor memory consolidation (Baraduc et al., 2004; Muellbacher et al., 2001, 2002). These studies often include an immediate retention test i.e. on the same experimental day. The control groups often show a small performance drop between practice blocks and at immediate retention (Baraduc et al., 2004; Muellbacher et al., 2002). The study by Muellbacher et al., (2001) introduced a 30 days follow-up test, without further repetition of the task, and here they report of maintained performance. The behavioral effects of motor practice observed in the present study both within- and between-session are thus fully in line with the observations reported in the literature.

-There are limited details pertaining to procedures at baseline. How much practice accumulated prior to conditioning stimulation?

Authors' response: As written in the Method section (see below), participants were allowed 5 familiarization trials, before the 10 baseline trials. Meaning, 15 trials in total before the PCMS protocol:

“Participants were allowed five familiarization trials before the baseline test on the first test day. Motor performance was quantified as peak acceleration and was measured in 10 trials at each time point (baseline, post-PCMS and post practice). At baseline and post-tests, no augmented feedback was provided on motor performance.”(lines 583-587)

-Figure 4 does not contain any indication of significant effects/interactions noted on lines 268-76.

Authors' response: Thank you for bringing this to our attention. We apologize for not providing this information in the previous version of the figure. Indications of significant differences have now been added to Figure 4, which is also presented below.

Figure 4. Experiment IV: Control Experiments investigating the effects of unpaired rPNS or rTMS, and the specificity of PCMS effects on corticospinal excitability. a) Within-subject design (N=10), with participants completing both rPNS (purple) or rTMS (light-

purple) separated by a week. To test the effects of non-paired low frequency stimulations on corticospinal excitability. **b)** Within-subject design (N=8), with participants completing three test days, to test the effects of three PCMS stimulation protocols on corticospinal excitability (1: PCMS+, blue; 2: PCMS-, purple, and 3: PCMS_{coupled-control}, orange). Note similar PCMS protocols was used as in Experiment III, but now with an expanded time course, and without interfering effects of motor practice. “##” indicates significantly larger MEPs after PCMS+ across the entire post stimulation period compared to both PCMS- and PCMS_{coupled-control}. “#” indicates significantly smaller MEPs after PCMS- across the entire post stimulation period compared to PCMS_{coupled-control}. **c)** The same 8 participants completed two test days, and the effects of motor practice without prior PCMS stimulations was compared against a non-practicing group (Rest) that neither received prior PCMS stimulations. “#” indicates significantly larger MEPs after Motor Practice across the entire post stimulation period compared to Rest. All three plots show MEP amplitude in percentage of baseline from each time point. Error bars indicate standard deviation. Note the identical scaling of the y-axis in the three plots. Significance level $p < 0.05$

References used in Authors’ responses:

- Bagce, H.F., S. Saleh, S. V. Adamovich, J.W. Krakauer & E. Tunik (2012): Corticospinal excitability is enhanced after visuomotor adaptation and depends on learning rather than performance or error. *Journal of Neurophysiology*, American Physiological Society Bethesda, MD, Vol. 109:4, pp. 1097–1106.
- Baraduc, P., N. Lang, J.C. Rothwell & D.M. Wolpert (2004): Consolidation of dynamic motor learning is not disrupted by rTMS of primary motor cortex. *Current Biology*, Elsevier, Vol. 14:3, pp. 252–256.
- Bestmann, S. & J.W. Krakauer (2015): The uses and interpretations of the motor-evoked potential for understanding behaviour. *Experimental Brain Research*, Vol. 233:3, pp. 679–689.
- Bi, G.Q. & M.M. Poo (1998): Synaptic modifications in cultured hippocampal neurons: Dependence on spike timing, synaptic strength, and postsynaptic cell type. *Journal of Neuroscience*, Vol. 18:24, pp. 10464–10472.
- Bunday, K.L. & M.A. Perez (2012a): Motor recovery after spinal cord injury enhanced by strengthening corticospinal synaptic transmission. *Current Biology*, Vol. 22:24, pp. 2355–2361.
- Bunday, K.L. & M.A. Perez (2012b): Motor recovery after spinal cord injury enhanced by strengthening corticospinal synaptic transmission. *Current Biology*, Elsevier Ltd, Vol. 22:24, pp. 2355–2361.
- Carroll, T.J., M. Lee, M. Hsu & J. Sayde (2008): Unilateral practice of a ballistic movement causes bilateral increases in performance and corticospinal excitability. *Journal of Applied Physiology*, Vol. 104:6, pp. 1656–1664.
- Cirillo, J., N.C. Rogasch & J.G. Semmler (2010): Hemispheric differences in use-dependent corticomotor plasticity in young and old adults. *Experimental Brain Research*, Vol. 205:1, pp. 57–68.
- Classen, J., J. Liepert, S.P. Wise, M. Hallett, L.G. Cohen et al. (1998): Rapid Plasticity of Human Cortical Movement Representation Induced by Practice. *Journal of Neurophysiology*, Vol. 79:, pp. 1117–1123.
- D’Amico, J.M., S.C. Dongés & J.L. Taylor (2018): Paired corticospinal-motoneuronal stimulation increases maximal voluntary activation of human adductor pollicis. *Journal of Neurophysiology*, Vol. 119:, pp. 369–376.
- Devanne, H., B.A. Lavoie & C. Capaday (1997): Input-output properties and gain changes in the human corticospinal pathway. *Experimental Brain Research*, Vol. 114:2, pp. 329–338.
- Dongés, S.C., C.L. Boswell-Ruys, J.E. Butler & J.L. Taylor (2019): The effect of paired corticospinal–motoneuronal stimulation on maximal voluntary elbow flexion in cervical spinal cord injury: an experimental study. *Spinal Cord*, Springer US, Vol. 57:9, pp. 796–804.
- Gelli, F., F. Del Santo, T. Popa, R. Mazzocchio & A. Rossi (2007): Factors influencing the relation between corticospinal output and muscle force during voluntary contractions. *European Journal of Neuroscience*, Vol. 25:11, pp. 3469–3475.
- Giesebrecht, S., H. Van Duinen, G. Todd, S.C. Gandevia & J.L. Taylor (2012): Training in a ballistic task but not a visuomotor task increases responses to stimulation of human corticospinal axons. *Journal of*

Neurophysiology, Vol. 107:, pp. 2485–2492.

- Kantak, S.S. & C.J. Winstein (2012): Learning-performance distinction and memory processes for motor skills: A focused review and perspective. *Behavioural Brain Research*, Elsevier B.V., Vol. 228:1, pp. 219–231.
- Krakauer, J.W., A.M. Hadjiosif, J. Xu, A.L. Wang & A.M. Haith (2019): Motor Learning. *Comprehensive Physiology*, Vol. 9:April, pp. 613–663.
- Lauber, B., J. Lundbye-Jensen, M. Keller, A. Gollhofer, W. Taube et al. (2013): Cross-limb interference during motor learning. *PLoS ONE*, Vol. 8:12, available at:<http://doi.org/10.1371/journal.pone.0081038>.
- Lee, M., M.R. Hinder, S.C. Gandevia & T.J. Carroll (2010): The ipsilateral motor cortex contributes to cross-limb transfer of performance.pdf. *The Journal of Physiology*, Vol. 588:1, pp. 201–212.
- McDonnell, M.N. & M.C. Ridding (2006): Transient motor evoked potential suppression following a complex sensorimotor task. *Clinical Neurophysiology*, Vol. 117:6, pp. 1266–1272.
- Muellbacher, W., U. Ziemann, B. Boroojerdi, L. Cohen & M. Hallett (2001): Role of the human motor cortex in rapid motor learning. *Experimental Brain Research*, Springer, Vol. 136:4, pp. 431–438.
- Muellbacher, W., U. Ziemann, J. Wissel, N. Dang, M. Kofler et al. (2002): Early consolidation in human primary motor cortex. *Nature*, Nature Publishing Group, Vol. 415:6872, pp. 640–643.
- Rogasch, N.C., T.J. Dartnall, J. Cirillo, M.A. Nordstrom & J.G. Semmler (2009): Corticomotor plasticity and learning of a ballistic thumb training task are diminished in older adults. *Journal of Applied Physiology*, Vol. 107:6, pp. 1874–1883.
- Shulga, A., P. Lioumis, E. Kirveskari, S. Savolainen, J.P. Mäkelä et al. (2015): The use of F-response in defining interstimulus intervals appropriate for LTP-like plasticity induction in lower limb spinal paired associative stimulation. *Journal of Neuroscience Methods*, Elsevier B.V., Vol. 242:, pp. 112–117.
- Stefan, K., E. Kunesch, L.G. Cohen, R. Benecke & J. Classen (2000): Induction of plasticity in the human motor cortex by paired associative stimulation. *Brain*, Vol. 123:3, pp. 572–584.
- Taylor, J.L. & P.G. Martin (2009): Voluntary motor output is altered by spike-timing-dependent changes in the human corticospinal pathway. *Journal of Neuroscience*, Soc Neuroscience, Vol. 29:37, pp. 11708–11716.
- Urbán, M.A., R.A. Ozdemir, T. Tazoe & M.A. Perez (2017): Spike-timing-dependent plasticity in lower-limb motoneurons after human spinal cord injury. *Journal of Neurophysiology*, Vol. 118:4, pp. 2171–2180.
- Del Vecchio, A., F. Negro, A. Holobar, A. Casolo, J.P. Folland et al. (2019): You are as fast as your motor neurons: speed of recruitment and maximal discharge of motor neurons determine the maximal rate of force development in humans. *Journal of Physiology*, Vol. 597:9, pp. 2445–2456.

REVIEWER COMMENTS

Reviewer #1 (Remarks to the Author):

I have no further comments on this revised manuscript.

Reviewer #2 (Remarks to the Author):

Although the manuscript is improved in some respects, the authors persist with a mechanistic explanation that is not fully supported by the data. As the authors indicate, most human studies have demonstrated Hebbian plasticity by way of facilitating or inhibiting corticospinal excitability, yet the latter is not observed in this study. Discussion of this specific result still does not have a more prominent place in the manuscript, as suggested, nor is it even reported in the Abstract. While using Hebbian principles to motivate the study is reasonable, asserting that the findings are in line with Hebbian learning rules “to a wide extent” (lines 82-85) is not an objective interpretation. Overall, this interpretation aggrandizes and, to some extent, distracts from the findings.

Other specific comments:

Abstract (lines 23-26): Specifically, when PCMS is timed to arrive at a facilitatory interval at CM-synapses, learning is improved. This protocol also led to larger increases in corticospinal excitability compared to control protocols, and unpaired stimulations did not increase corticospinal excitability (Experiment IV).

Use of the modifier “facilitatory” interval seems preemptive in this context. It also seems misleading to not disclose that an inhibitory effect on corticospinal excitability was not observed.

Abstract (lines 29-30): This offers a mechanistic rationale to enhance motor practice effects in neurorehabilitation by priming sensorimotor training with individualized PCMS.

Use of “this” is vague. Regardless, extrapolating these findings to neurorehabilitation seems to be a bit of a stretch based on the data obtained from young adults who are neurologically normal.

Introduction (lines 42-45): By repeatedly pairing transcranial magnetic stimulation of the primary motor cortex (M1) and electrical stimulation of peripheral nerves timed to arrive at the corticomotoneuronal (CM) synapses in close temporal proximity, spike-timing-dependent, bidirectional changes in CM transmission can be induced 2,5.

What is being paired? What is arriving? Perhaps wording can be borrowed from the new content on lines 59-62?

Introduction (lines 82-85): The observed priming effects of PCMS on ballistic motor learning are to a wide extent in line with Hebbian learning rules demonstrating the importance of close temporal proximity and order of spike timing.

It seems misleading to suggesting that the effects are in line with Hebbian learning rules “to a wide extent” considering reported effects on learning and corticospinal excitability.

Results (lines 292-297): Furthermore, unpaired stimulations did not lead to changes in corticospinal excitability, showing that facilitation of corticospinal excitability in the present study was coupling dependent and required paired stimulations. Finally, all PCMS protocols increased corticospinal excitability, with the largest effect observed after PCMS+ demonstrating the importance of stimulus timing also for changes in corticospinal excitability.

As noted previously, simply because the PCMS- protocol yielded less facilitation on MEP size than the PCMS+ protocol does not account for the fact that it did not exert its intended effect in most subjects. The authors addressed this comment elsewhere in the current manuscript, but the above statement is new and contradicts the authors’ intended interpretation noted in their response.

Discussion (lines 320-324): Finally, this study observed that priming effects on motor learning are in line with Hebbian learning rules (Experiment III): PCMS with a short inter-arrival interval of pre-synaptic activation just before post-synaptic activation at the level of the CM-synapse improved motor learning compared to PCMS protocols with longer timing intervals and opposite order of spike-timing.

While this statement is a more objective account of the findings, an alternative to “in line” is needed here. Perhaps, “approximates” Hebbian learning rules is more representative.

Discussion (lines 340-342): This suggests that PCMS+ results in long lasting effects on motor learning that may have therapeutical relevance.

Although “therapeutical” is not a word, suggesting that a finding in healthy, young adults seems misleading without qualifying statements nor data in support.

Discussion (lines 380-383): Nevertheless, the results of Experiment III, provide proof-of-principle that the priming effects of PCMS on ballistic learning conform to Hebbian learning rules, in that they are contingent on temporal proximity of paired stimulations and in that they are governed by order of spike-timing.

Same issue here as noted above. Use “approximate” rather than “conform to.” Instead of “are” contingent on/governed by, the authors should consider “may be” contingent on/governed by.

One final note is that some highly relevant work has been published while this manuscript has been under review and should be incorporated.

Journal of Neurophysiology:

<https://doi.org/10.1152/jn.00499.2022>

CNS Neuroscience & Therapeutics:

<https://doi.org/10.1111/cns.14561>

Revision-Round-4: Point-by-point Responses to Referees comments

We would like to thank both reviewers for their time and comments to the manuscript. We have addressed the remaining comments from Reviewer 2, and believe that this has improved the quality of the manuscript. The following general changes have been made in the revised version of our manuscript:

- (1) We now further emphasize that the behavioral findings are in line with Hebbian principles, while acknowledging that the effects on corticospinal excitability are not. This is stated already in the abstract and further explicitly acknowledged and elaborated upon in the discussion.
- (2) We have added clarity to several key sentences mentioned by the Reviewer.
- (3) We have added a paragraph in the discussion and incorporated the key papers which have been published while this manuscript has been under review as suggested by Reviewer 2.

These changes will be detailed in our point-by-point response to the comments below.

Color coding:

- Responses to referees are color-coded blue
- Green text in the Manuscript represents specific changes made in Revision-Round-4.

Reviewer #1 (Remarks to the Author):

I have no further comments on this revised manuscript.

Authors' response: We thank the reviewer for all previous comments and the acknowledgement of the revised manuscript.

Reviewer #2 (Remarks to the Author):

Although the manuscript is improved in some respects, the authors persist with a mechanistic explanation that is not fully supported by the data. As the authors indicate, most human studies have demonstrated Hebbian plasticity by way of facilitating or

inhibiting corticospinal excitability, yet the latter is not observed in this study. Discussion of this specific result still does not have a more prominent place in the manuscript, as suggested, nor is it even reported in the Abstract. While using Hebbian principles to motivate the study is reasonable, asserting that the findings are in line with Hebbian learning rules “to a wide extent” (lines 82-85) is not an objective interpretation. Overall, this interpretation aggrandizes and, to some extent, distracts from the findings.

Overall authors’ response: We want to thank the reviewer for the constructive feedback and for acknowledging our revisions. We start with our overall response, followed by responses to the ‘other specific comments’ mentioned by the reviewer.

As mentioned by the reviewer, using Hebbian principles to motivate the study is reasonable. We provide evidence that priming of ballistic motor learning promotes learning. It does so, when stimulations are specifically timed to facilitate synaptic transmission at the CM-level (PCMS+). Additionally, we also observe smaller bidirectional effects in agreement with Hebbian principles. That is, decreased early learning when PCMS- is delivered. Consequently, the observed behavioral effects on motor learning are in line with Hebbian principles of plasticity. This is the main focus of the manuscript.

Secondly, we agree with the reviewer that the results on corticospinal excitability do not support Hebbian principles on all accounts – in the sense that the results do not demonstrate bidirectional effects. The electrophysiological results demonstrate specificity in the sense, that PCMS+ is accompanied by larger increases in corticospinal excitability compared to other protocols. We thus recognize, that PCMS- does not lead to a decrease in overall corticospinal excitability. It is important to emphasize that this does not exclude the possibility of bidirectional effects at the targeted CM level, but our experimental setup do not allow teasing apart differential effects on different levels of the corticospinal system (cortical vs. spinal). We acknowledge this input from the reviewer and have now mentioned this explicitly in the abstract. This nuanced interpretation of our findings is now positioned at a more prominent place in the manuscript. We still also address this thoroughly in the discussion.

We have now also made it clearer in the relevant paragraphs that the behavioral findings approximate Hebbian principles while the effects on corticospinal excitability do not display bidirectional effects. This is now incorporated in relevant paragraphs of the abstract, in the end of the introduction section, and in the discussion. Please find the updated sections under ‘other specific comments’ below.

In summary, with the changes listed above, we believe that we thoroughly embrace and address the complexity of the findings on motor behavior and corticospinal excitability. Lastly, we want to highlight that the main focus of the manuscript is – and has been throughout - on the behavioral findings. We believe that the electrophysiological results can be reasonably explained, and we provide potential explanations in the discussion. However, we also agree that these findings necessitate a nuanced interpretation and a detailed discussion. We would therefore like to thank the reviewer for the constructive input.

Below we address the more specific comments by Reviewer 2.

Other specific comments:

Abstract (lines 23-26): “*Specifically, when PCMS is timed to arrive at a facilitatory interval*

at CM-synapses, learning is improved. This protocol also led to larger increases in corticospinal excitability compared to control protocols, and unpaired stimulations did not increase corticospinal excitability (Experiment IV). “

Use of the modifier “facilitatory” interval seems preemptive in this context. It also seems misleading to not disclose that an inhibitory effect on corticospinal excitability was not observed.

Authors’ response: We agree with the Reviewer and the sentence has now been rephrased. The modifier ‘facilitatory’ has been deleted, and we now more explicitly describe the protocol. We also mention that an intended inhibitory protocol did not lead to a changes in corticospinal excitability. Together with the overall comment from the Reviewer, the Abstract now also includes a sentence on the behavioral findings being in part aligned with Hebbian learning rules, while the effects on corticospinal excitability demonstrate timing-specificity but not bidirectionality.

Updated Abstract below:

“Motor learning relies on experience-dependent plasticity in relevant neural circuits. In four experiments, we provide initial evidence and a double-blinded, sham-controlled replication (Experiment I-II) demonstrating that motor learning involving ballistic index finger movements is improved by preceding paired corticospinal-motoneuronal stimulation (PCMS), a human model for exogenous induction of spike-timing-dependent plasticity. Behavioral effects of PCMS targeting corticomotoneuronal (CM) synapses are order- and timing-specific and partially bidirectional (Experiment III). PCMS with a 2ms inter-arrival interval at CM-synapses enhances learning and increases corticospinal excitability compared to control protocols. Unpaired stimulations did not increase corticospinal excitability (Experiment IV). Our findings demonstrate that non-invasively induced plasticity interacts positively with experience-dependent plasticity to promote motor learning. The effects of PCMS on motor learning approximate Hebbian learning rules, while the effects on corticospinal excitability demonstrate timing-specificity but not bidirectionality. These findings offer a mechanistic rationale to enhance motor practice effects by priming sensorimotor training with individualized PCMS.”(Lines 16-31)

We further would like to thank the reviewer for suggestions using the word ‘approximate’ to describe our effects on skill learning from a Hebbian point-of-view (see below). This has now also been incorporated in the abstract.

Abstract (lines 29-30): *“This offers a mechanistic rationale to enhance motor practice effects in neurorehabilitation by priming sensorimotor training with individualized PCMS.”*

Use of “this” is vague. Regardless, extrapolating these findings to neurorehabilitation seems to be a bit of a stretch based on the data obtained from young adults who are neurologically normal.

Authors’ response: We agree with the Reviewer, we have removed “neurorehabilitation”, from the abstract. Please find the updated abstract inserted above.

Introduction (lines 42-45): *“By repeatedly pairing transcranial magnetic stimulation of the primary motor cortex (M1) and electrical stimulation of peripheral nerves timed to arrive at the corticomotoneuronal (CM) synapses in close temporal proximity, spike-timing-dependent, bidirectional changes in CM transmission can be induced 2,5.”*

What is being paired? What is arriving? Perhaps wording can be borrowed from the new

content on lines 59-62?

Authors' response: We agree with the Reviewer, and have modified the sentence accordingly. Thank you for your suggestion. The sentence has been rephrased, and is now more specific:

“By repeatedly pairing **descending corticospinal volleys elicited by** transcranial magnetic stimulation (TMS) of the primary motor cortex (M1) **with an antidromic volley in the motor axons triggered by** electrical stimulation of peripheral nerves timed to arrive at the corticomotoneuronal (CM) synapses in close temporal proximity, it is possible to induce spike-timing-dependent, bidirectional changes in CM transmission” (Lines 42-47).

Introduction (lines 82-85): “*The observed priming effects of PCMS on ballistic motor learning are to a wide extent in line with Hebbian learning rules demonstrating the importance of close temporal proximity and order of spike timing.*”

It seems misleading to suggest that the effects are in line with Hebbian learning rules “to a wide extent” considering reported effects on learning and corticospinal excitability.

Authors' response: We agree with the reviewer and have rephrased the sentence. Importantly, this sentence only described the effects on ballistic motor learning and did not pertain to changes in corticospinal excitability. We have now added a sentence on the effects on corticospinal excitability.

“The observed priming effects of PCMS on ballistic motor learning **approximate** Hebbian learning rules demonstrating the importance of close temporal proximity and order of spike timing. The effects of PCMS on corticospinal excitability demonstrate timing-specificity but not bidirectionality.” (Line 84-88)

Results (lines 292-297): “*Furthermore, unpaired stimulations did not lead to changes in corticospinal excitability, showing that facilitation of corticospinal excitability in the present study was coupling dependent and required paired stimulations. Finally, all PCMS protocols increased corticospinal excitability, with the largest effect observed after PCMS+ demonstrating the importance of stimulus timing also for changes in corticospinal excitability.*”

As noted previously, simply because the PCMS- protocol yielded less facilitation on MEP size than the PCMS+ protocol does not account for the fact that it did not exert its intended effect in most subjects. The authors addressed this comment elsewhere in the current manuscript, but the above statement is new and contradicts the authors' intended interpretation noted in their response.

Authors' response: We thank the reviewer for this comment. As noted, the highlighted sentence is placed in the Results section. Consequently, we have rewritten the paragraph, that now summarize the findings of Experiment IV, without providing any interpretation of the specific results.

“Collectively, Experiment IV demonstrates that unpaired stimulations do not lead to changes in corticospinal excitability, showing that facilitation of corticospinal excitability in the present study was coupling-dependent and required paired stimulations. All PCMS protocols increased corticospinal excitability, with the largest effect observed after PCMS+. Finally, Experiment IV demonstrates that ballistic motor learning is accompanied by increases in corticospinal excitability.” (Lines 295-300)

As mentioned by the Reviewer, the interpretation of these results is placed elsewhere, namely in the discussion.

Discussion (lines 320-324): *“Finally, this study observed that priming effects on motor learning are in line with Hebbian learning rules (Experiment III): PCMS with a short inter-arrival interval of pre-synaptic activation just before post-synaptic activation at the level of the CM-synapse improved motor learning compared to PCMS protocols with longer timing intervals and opposite order of spike-timing.”*

While this statement is a more objective account of the findings, an alternative to “in line” is needed here. Perhaps, “approximates” Hebbian learning rules is more representative.

Authors’ response: We agree with the reviewer, we have changed the wording to “approximate”.

Discussion (lines 340-342): *“This suggests that PCMS+ results in long lasting effects on motor learning that may have therapeutical relevance.”*

Although “therapeutical” is not a word, suggesting that a finding in healthy, young adults seems misleading without qualifying statements nor data in support.

Authors’ response: We thank the reviewer, and have now rephrased the sentence, and added qualifying statements.

“This suggests that PCMS+ results in long lasting effects on motor learning that may have relevance in neurorehabilitation. Previous studies did not find an effect of PCMS on maximal voluntary contraction^{13,14}, but a recent study found a positive effect of PCMS on subsequent force control motor practice in electrophysiological responders¹⁵. Other studies have found positive effects of PCMS alone on motor functions in individuals with and without spinal cord injury^{6,7,12}. However, the priming effect of PCMS on motor learning demonstrated in the current study remains to be investigated in patient groups. (Lines 343-351).

Discussion (lines 380-383): *“Nevertheless, the results of Experiment III, provide proof-of-principle that the priming effects of PCMS on ballistic learning conform to Hebbian learning rules, in that they are contingent on temporal proximity of paired stimulations and in that they are governed by order of spike-timing.”*

Same issue here as noted above. Use “approximate” rather than “conform to.” Instead of “are” contingent on/governed by, the authors should consider “may be” contingent on/governed by.

Authors’ response: We thank the reviewer, and have now rephrased the sentence accordingly.

“Nevertheless, the results of Experiment III, provide proof-of-principle that the priming effects of PCMS on ballistic learning approximate Hebbian learning rules, in that they may be contingent on temporal proximity of paired stimulations and order of spike-timing.” (Lines 387-390).

One final note is that some highly relevant work has been published while this manuscript has been under review and should be incorporated.

Journal of Neurophysiology: <https://doi.org/10.1152/jn.00499.2022>

CNS Neuroscience & Therapeutics: <https://doi.org/10.1111/cns.14561>

Authors’ response: We thank the reviewer for bringing this to our attention.

We have incorporated the paper from Grover et al., (2023) in the discussion section. We use this reference as one of the studies demonstrating positive effects of PCMS on motor *performance* and corticospinal excitability in both individuals with and without spinal cord injury. We address that the effects of PCMS on motor learning remains to be investigated in patient groups. We have also added a reference to the Urbin et al., (2024) paper in the same part of the discussion: “a recent study found a positive effect of PCMS on subsequent force control motor practice in electrophysiological responders¹⁵” (Lines 346-348). We would like to note that Urbin et al. (2024) actually cite the *bioRxiv* Preprint version of the present manuscript in their discussion section. Urbin et al. write: “A recent mechanistic study in neurologically intact young adults also showed a priming effect of PCMS on subsequent learning of ballistic index finger movements”. We also note – as mentioned by the editor - that “When evaluating your revised manuscript, we will not consider any similar papers published independently in the meantime to compromise the novelty of your study”. It is nevertheless relevant to cite the mentioned papers, which have been submitted and published while this manuscript has been in review.

References used in Authors’ responses:

Grover, F.M., B. Chen & M.A. Perez (2023): Increased paired stimuli enhance corticospinal-motoneuronal plasticity in humans with spinal cord injury. *Journal of Neurophysiology*, Vol. 129:6, pp. 1414–1422.

Urbin, M.A., C.W. Lafe, M.E. Bautista, G.F. Wittenberg & T.W. Simpson (2024): Effects of noninvasive neuromodulation targeting the spinal cord on early learning of force control by the digits. *CNS Neuroscience and Therapeutics*, Vol. 30:2, pp. 1–13.